# The comparative role of physical system processes in Hudson Strait ice stream cycling: a comprehensive model-based test of Heinrich event hypotheses

Kevin Hank[1] and Lev Tarasov[1]

[1]Department of Physics and Physical Oceanography, Memorial University of Newfoundland, St. John's, NL, A1B 3X7,Canada
*khank@mun.ca

**Correspondence:** Kevin Hank (khank@mun.ca)

**Abstract.** Despite their recognized significance on global climate and extensive research efforts, the mechanism(s) driving Heinrich Events remain(s) a subject of debate. Here, we use the 3D thermo-mechanically coupled Glacial Systems Model (GSM) to examine Hudson Strait ice stream surge cycling as well as the role of 3 factors previously hypothesized to play a critical role in Heinrich events: ice shelves, glacial isostatic adjustment, and sub-surface ocean temperature forcings. In contrast to all previous modeling studies examining HEs, the GSM uses a transient last glacial cycle climate forcing, global visco-elastic glacial isostatic adjustment model, and sub-glacial hydrology model. The results presented here are based on a high-variance sub-ensemble retrieved from North American history matching for the last glacial cycle.

Over our comparatively wide sampling of the potential parameter space (52 ensemble parameters for climate forcing and process uncertainties), we find two modes of Hudson Strait ice streaming: classic binge-purge versus near continuous ice streaming with occasional shutdowns and subsequent surge onset overshoot. Our model results indicate that large ice shelves covering the Labrador Sea during the last glacial cycle only occur when extreme calving restrictions are applied. The otherwise minor ice shelves provide insignificant buttressing for the Hudson Strait ice stream. While sub-surface ocean temperature forcing leads to minor differences regarding surge characteristics, glacial isostatic adjustment does have a significant impact. Given input uncertainties, the strongest controls on ice stream surge cycling are the poorly constrained deep geothermal heat flux under Hudson Bay and Hudson Strait and the basal drag law. Decreasing the geothermal heat flux within available constraints and/or using a Coulomb sliding law instead of a Weertman-type power law leads to a shift from the near-continuous streaming mode to the binge-purge mode.

# 1 Introduction

Heinrich Events (HEs) offer a near unique opportunity to explore a coupled ice-climate-ocean instability that operates on the order of centuries to millennia (Heinrich, 1988). HEs are generally attributed to large armadas of icebergs drifting from Hudson Strait across the North Atlantic ($40°$ - $55°$N) while depositing sediment layers of Ice-Rafted Debris (IRD). At least six of these Heinrich Layers have been identified between $60$ kyr BP (before present) and $15$ kyr BP, but there is evidence for earlier events (Table 6.3 in Bradley, 2014, and references therein). Heinrich Layers are usually characterized by high detrital

carbonate concentration, low abundance of organic carbon and terrigenous lipids, and high magnetic susceptibility (Hemming, 2004). Mineralogical inferences from IRD records indicate a Hudson Strait/Hudson Bay provenance, but there is also evidence for Icelandic and European source regions for at least 2 HEs (Grousset et al., 1993; Gwiazda et al., 1996; Hemming, 2004). HEs generally coincide with the coldest phases of the Dansgaard-Oeschger (DO) cycles, followed by a sharp temperature increase (e.g., Hemming, 2004; Clement and Peterson, 2008; Hodell et al., 2010). Marine records indicate a weakening of the Atlantic

Meridional Overturning Circulation (AMOC) and an increase in sub-surface ocean temperatures leading up to HEs (Marcott et al., 2011, and references therein).

  Despite over $50$ years of research, the exact mechanism behind HEs remains unclear. The proposed hypotheses include an internally driven binge-purge model (MacAyeal, 1993; Payne, 1995; Calov et al., 2002), an atmospherically synchronized surge mechanism (Schannwell et al., 2024), an ice shelf buildup-collapse mechanism (Hulbe, 1997; Hulbe et al., 2004), and an

underwater melt-driven grounding line retreat modulated by glacial isostatic adjustment (GIA, Bassis et al., 2017). However, an extensive study simultaneously investigating the relative role of each proposed HE hypothesis is still missing. Furthermore, previous model-based tests of HE-related Hudson Strait surge cycling have largely ignored uncertainties in key potentially relevant processes and inputs. These include the deep geothermal heat flux under Hudson Strait, glacio-isostatic adjustment, and the form of the basal drag law employed.

Here we run sensitivity experiments with the 3D thermo-mechanically coupled glacial systems model (GSM, Tarasov et al., 2024) and a high-variance (with respect to ensemble parameters and ice sheet configuration) ensemble-based approach to determine the role that relevant physical system processes play in HEs. A challenge in this context is the robust modeling of ice stream cycling associated with 2 of the 3 hypotheses discussed herein. This challenge is largely due to the abrupt changes with ice stream activation/de-activation, resulting in potentially high sensitivity to the implementation and discretization of

the relevant dynamical equations (Hank et al., 2023). As a step towards addressing this, we use **M**inimum **N**umerical **E**rror **E**stimates (MNEEs) as a threshold for the numerical significance of model results.

  In particular, we address the following research questions.

$Q_1$ *What are the characteristics of Hudson Strait ice stream surges and how do they depend on the geothermal heat flux and basal drag law?*

Given the critical role of basal temperature and basal drag on ice stream cycling, we examine the to-date mostly unassessed sensitivity of Hudson Strait ice stream surge characteristics to deep GHF input uncertainty and different basal sliding laws.

$Q_2$ *Can the sudden reduction of the buttressing effect of ice shelves trigger Hudson Strait ice stream surging?*

A 2°C increase in the sub-surface ocean temperature has been shown to cause a 6 fold increase in the ice shelf basal melt rates in front of Hudson Strait ($\sim 6\,\mathrm{m\,yr^{-1}}$ to 35-40 $\mathrm{m\,yr^{-1}}$) in simulations with an ocean/ice-shelf model (Marcott et al., 2011). Such an increase can significantly degrade the buttressing effect of a confined ice shelf and, thereby, potentially trigger ice stream activation or surging (e.g., Álvarez-Solas et al., 2011). To test this scenario, forced ocean warming experiments are carried out with forcing timing set to that of the HE record (timing based on average of Table 6.3 in Bradley (2014), "HE shelf forcing").

$Q_3$ *Can a sudden breakup of fringing ice shelves along the Canadian coast explain the IRD records (without the need for surges)?*

HEs occurred during extremely cold climates. Hulbe et al. (2004) propose that these cold conditions led to the formation of fringing ice shelves along the Canadian coast. A sudden disintegration of the ice shelves would provide a source for Canadian-source icebergs (Hulbe, 1997) and could potentially explain the IRD records found in ocean sediment cores from the North Atlantic.

$Q_4$ *How does sub-surface ocean warming affect HEs?*

In the HE mechanism proposed by Bassis et al. (2017), a glacial isostatic uplifted bed topography at the Hudson Strait mouth protects the retreated ice sheet front from sub-surface ocean warmings (SSOWs) attributed to Dansgaard–Oeschger events (DO events). The ice sheet grows, eventually depressing the bed topography. Once the bed topography is depressed below the upper limit of the SSOW, the ice sheet front is vulnerable to ocean forcing. Due to a retrograde sloping bed, the ice sheet rapidly retreats during the next SSOW, allowing the bed topography to rise and isolate the ice sheet front from ocean forcing.

Due to the numerous differences in the model setup (e.g., model domain considered, grid discretization near the grounding line, GIA model, calving and sub-shelf melt implementations, and the lack of ice thermodynamics in Bassis et al. (2017)), we do not aim to directly replicate the experiments in Bassis et al. (2017). Instead, we examine the role of SSOW in a HE context by applying a sub-surface ocean temperature increase for every DO event (similar to the approach used by, e.g., Alvarez-Solas et al. (2010, 2013)).

$Q_5$ *What is the role of GIA in a HE context?*

Due to its effect on, e.g., relative sea level changes and ice sheet mass balance, GIA has long been known to have a significant impact on ice sheet evolution (e.g., Tarasov and Peltier, 1997). It also plays a central role in the HE hypothesis of Bassis et al. (2017). We assess the impact of GIA representation, associated earth model uncertainties, and GIA absence on Hudson Strait ice stream surge cycling.

Starting with a brief description of the GSM, the most relevant model features in a HE context are outlined in Sec. 2.1. A complete list of model features is described in Tarasov et al. (2024). The applied geothermal heat flux and ocean temperature

forcing are detailed in Sec. 2.2 and 2.3, respectively. Sec. 2.1 to 2.3 also outline how the model setup for the corresponding sensitivity experiments differ from the GSM reference setup. The experimental design (summary table of all sensitivity experiments, derivation of reference parameter ensemble, MNEEs, and run comparison analysis) is summarized in Sec. 2.4. Following the structure of the research questions ($Q_1$ to $Q_5$), the results of the sensitivity experiments are presented in Sec. 3 and discussed in Sec. 4. The supplementary material referenced throughout this text (indicated by a capital S) provides additional information to support the corresponding claim but is not essential to the understanding of this study.

## 2 Methods

### 2.1 Glacial Systems Model

All experiments within this study are conducted with the glacial systems model (GSM, Tarasov and Peltier, 1999, 2007; Pollard and DeConto, 2012; Tarasov et al., 2024). The GSM is a 3D thermo-mechanically coupled glaciological model designed for large ensemble modeling in glacial cycle modeling contexts. Ice thermodynamics and basal melt for grounded ice are computed via an energy-conserving finite-volume solver. While the model domain in this study only covers North America and a stub North-West Greenland (topography and sediment cover of the entire model domain are shown in Fig. S1), the model configuration is informed by modeling of all last glacial cycle major ice sheets as well as major ice caps (or minor ice sheets) such as Icelandic and Patagonian.

**Model initialization**

The GSM is initialized from ice-free conditions at 122 kyr BP and is run to present day with a maximum ice dynamics time step size of 1 yr. The ice dynamics time step size is automatically decreased to meet CFL (Courant–Friedrichs–Lewy) constraint (Courant et al., 1928) or when the matrix solver fails to converge. Model runs automatically terminate once the time step size is reduced below the set minimum of 0.015625 yr. Such ("crashed") runs are not considered for any analysis conducted within this study. The horizontal grid resolution is $\Delta_{\mathrm{lon}} = 0.5°$, $\Delta_{\mathrm{lat}} = 0.25°$ (finer horizontal grid resolutions are currently computationally unfeasible in the context of this study as the GSM in its current incarnation is not parallelized).

**Ice dynamics**

The GSM uses hybrid shallow shelf/ice physics based on the ice dynamical core originally developed by Pollard and DeConto (2012). The hybrid ice dynamics are activated once the shallow-ice approximation basal velocity within a grid cell exceeds $30 \mathrm{~m~yr}^{-1}$ over soft sediments and $200 \mathrm{~m~yr}^{-1}$ over hard bedrock. The default basal drag law for our experiments is a Weertman-type power law,

$$\boldsymbol{u}_{\mathbf{b}} = C_{\mathrm{b}} |\boldsymbol{\tau}_{\mathbf{b},\mathbf{W}}|^{n_{\mathrm{b}}-1} \boldsymbol{\tau}_{\mathbf{b},\mathbf{W}}, \tag{1}$$

where $\boldsymbol{u}_{\mathbf{b}}$ is the basal sliding velocity (imposed upper limit of $40 \mathrm{~km~yr}^{-1}$), $\boldsymbol{\tau}_{\mathbf{b},\mathbf{W}}$ the basal stress, and $n_{\mathrm{b}}$ the bed power strength. In contrast to the version used in Hank et al. (2023), $n_{\mathrm{b}}$ has separate hard and soft bed values ($n_{\mathrm{b,hard}} = 4$ (Maier

et al., 2021) and ensemble parameter $n_{b,soft}$ in Table S1). To indirectly account for sub-temperate sliding, the basal sliding coefficient $C_b$ is calculated according to

$$C'_b = \max\left[F_{warm} C_{warm}; C_{froz}\right], \tag{2}$$

where $C'_b$ is the basal sliding coefficient in an intermediate calculation step and $F_{warm}$ the estimated warm-based fraction of a grid cell (Hank et al., 2023). $C_{froz} = 2 \cdot 10^{-4}$ m yr$^{-1}$ $\left(5 \cdot 10^{-6}$ Pa$^{-1}\right)^4$ is the fully cold-based sliding coefficient (needed for numerical regularization). $C_{warm}$ is the fully warm-based sliding coefficient with dependence on bed properties:

$$C_{warm,soft} = C_{rmu} \cdot \min\left[1; \max\left[0.2; \frac{F_{sub,till}}{0.01\,\sigma_{hb}}\right]\right] \tag{3a}$$

$$C_{warm,hard} = C_{slid} \cdot \min\left[1; \max\left[0.1; \frac{F_{sub,slid}}{0.01\,\sigma_{hb}}\right]\right] \cdot (1 + 20\,F_{sed}). \tag{3b}$$

Here $C_{rmu}$ and $C_{slid}$ are the ensemble parameters for the Weertman soft- and hard-bed sliding coefficients, respectively (Table S1). The ensemble parameters $F_{sub,till}$ and $F_{sub,slid}$ (Table S1) impose the Weertman basal drag dependencies on the subgrid standard deviation of the bed elevation $\sigma_{hb}$ and $F_{sed}$ is the subgrid fraction of soft bed cover. $C_b$ is also dependent on the effective pressure $N_{eff}$

$$C_b = C'_b \cdot \min\left[10; \max\left[0.5; \frac{N_{eff,Fact}}{N_{eff} + N_{eff,min}}\right]\right], \tag{4}$$

where $N_{eff,Fact}$ is the effective-pressure factor (ensemble parameter, Table S1). $N_{eff,min} = 10$ kPa regularizes the dependence of basal drag on effective pressure.

The effective pressure itself is given by an empirically-derived dependence (Flowers, 2000) on basal water thickness $h_{wb}$:

$$N_{eff} = g\rho_{ice}H \cdot \left(1 - \min\left[\frac{h_{wb}}{h_{wb,Crit}}; 1.0\right]^{3.5}\right), \tag{5}$$

where $g = 9.81$ m s$^{-2}$ is the acceleration due to gravity, $\rho_{ice} = 910$ kg m$^{-3}$ the ice density, $H$ the ice thickness, and $h_{wb,Crit}$ an estimated effective bed roughness scale (ensemble parameter in Table S1, see also Flowers and Clarke, 2002; Drew and Tarasov, 2023). The local basal hydrology model nominally sets the time derivative of $h_{wb}$ to the difference between the basal melt rate $M_b$ (from conservation of energy at the ice sheet base) and a constant bed drainage rate $R_{b,drain}$ (ensemble parameter, Table S1). The basal water thickness is limited to $h_{wb,max} = 10$ m and is set to $h_{wb} = 0$ m where the ice thickness is less than $h_{hyd,lim} = 10$ m and where the temperature with respect to the pressure melting point ($T_{bp}$) is below $T_{bp,lim} = -0.1°$C. Previous experiments showed changes in $h_{wb,max}$, $h_{hyd,lim}$, and $T_{bp,lim}$ do not significantly affect surge characteristics (Hank et al., 2023). Additional experiments using $h_{hyd,lim} = 100$ m conducted within this study support this finding (Fig. S2 and S3).

To examine the effects of the choice of basal drag law on surge characteristics, we also run experiments with a Coulomb sliding law

$$\boldsymbol{\tau}_{\mathbf{b,C}} = \max\left[1\text{ kPa}; N_{eff}C_c \tan(\Theta_t)\right], \tag{6a}$$

or in a much more numerically stable regularized form:

$$\boldsymbol{\tau}_{\mathbf{b,C}} = \max\left[1 \text{ kPa}; N_{\text{eff}}C_{\text{c}}\tan\left(\Theta_{\text{t}}\right)\left(\frac{\boldsymbol{u_{\mathbf{b}}}^2}{\boldsymbol{u_{\mathbf{b}}}^2 + UV_{\text{C,reg}}}\right)^{1/6}\right], \tag{6b}$$

where $C_{\text{c}}$ is a variable drag coefficient (scalar ensemble parameter, Table S1) and $\Theta_{\text{t}}$ the elevation-dependent till friction angle (Tarasov et al., 2024). According to CFL constraints, we expect even the regularized Coulomb law (Eq. 6b) to be more unstable than the Weertman law, because it negligibly increases basal drag for basal velocities beyond the order of the regularization threshold $UV_{\text{C,reg}} = 20 \text{ m yr}^{-1}$. This is compounded by the grounding-line flux iteration in the SSA solution.

When Coulomb drag is high, Weertman-type enhanced deformation around controlling obstacles can still occur (especially given the physical separation of the Coulomb plastic deformation process within the till layer) and dominate the basal sliding. Therefore, the basal shear stress used in the GSM when the Coulomb drag option is activated follows Tsai et al. (2015):

$$\boldsymbol{\tau_{\mathbf{b}}} = \min\left[\boldsymbol{\tau_{\mathbf{b,W}}}; \boldsymbol{\tau_{\mathbf{b,C}}}\right]. \tag{7}$$

The calculation of the ice flux across the grounding line in the GSM is independent of the choice of sliding law for the rest of the ice sheet. In the reference setup (Weertman-type power law only, Eq. 1), the basal shear stress at the grounding line $\boldsymbol{\tau_{\mathbf{b,GL}}}$ follows Eq. 7. To determine the implications of this grounding line ice flux implementation on surge characteristics, we also run experiments with $\boldsymbol{\tau_{\mathbf{b,GL}}} = \boldsymbol{\tau_{\mathbf{b,W}}}$. Furthermore, the GSM reference setup uses the Schoof (2007) grounding line flux condition as implemented in Pollard and DeConto (2012). When using this approach, issues can arise for complex 2D geometries (likely of most consequence for Antarctica; Reese et al., 2018). Therefore, we examine the effect of using a revised validated treatment (Pollard and Deconto, 2020). The surge characteristics are not significantly affected by the revised grounding line treatment (GLT, Fig. S2 and S3).

The GSM includes the dynamics of Marine Ice Cliff Instability (MICI, DeConto and Pollard, 2016), but only for ice-covered grid cells adjacent to neighbouring open ocean.

**Heinrich Event-relevant model features**

Modeling sensitivities and numerical requirements for a model configuration that had reduced grid resolution dependence in an ice stream surge cycling context were determined based on an slightly earlier version of the GSM (Hank et al., 2023). The inclusion of basal sliding at sub-freezing temperatures (Hank et al., 2023), a basal hydrology representation, and the dampening effect of a bed thermal model on basal-temperature changes significantly affect the surge characteristics. To account for these modeling aspects, the GSM version used within this study is run with a resolution-dependent basal temperature ramp, a local basal hydrology model (Drew and Tarasov, 2023), and a $4 \text{ km}$ deep bed thermal model.

Additionally, the GSM configuration within this study incorporates an asynchronously coupled global visco-elastic GIA solver (Tarasov and Peltier, 1997) and an asynchronously coupled geographically-resolved energy balance climate model (EBM; Deblonde et al., 1992). The time step size for the GIA solver and EBM is $100 \text{ yr}$ (compared to a maximum ice dynamics time step size of $1 \text{ yr}$). The global GIA solver uses ice sheet chronologies from recently completed history matching

for all ice sheets not within the model domain. To determine the effect of the global GIA solver, we run sensitivity experiments
without GIA and with a simple local relaxation time-based GIA model (relaxation time constant $\tau = [3, 4, 5]$ kyr). Instead of a
constant climate forcing (e.g., Roberts et al., 2016) or only varying the ocean forcing (e.g., Alvarez-Solas et al., 2013; Bassis
et al., 2017), we apply a transient last glacial cycle climate forcing.

## 2.2 Geothermal heat flux

The deep geothermal heat flux (GHF) input used in the GSM is constant in time but varies spatially (default GHF shown in
Fig. 1a). It provides the lower boundary flux condition for a 4 km deep bed thermodynamic model fully embedded in the
ice thermodynamic solver. The default GHF input field is from Davies (2013) and represents an upper bound of the literature
estimates. However, GHF data in Hudson Bay and Hudson Strait are sparse, and the estimated values differ significantly
(Fig. S4a,b and, e.g., Jessop and Judge, 1971; Pollack et al., 1993; Shapiro and Ritzwoller, 2004; Blackwell and Richards,
2004; Levy et al., 2010; Goutorbe et al., 2011; Jaupart et al., 2014; Lucazeau, 2019; Cuesta-Valero et al., 2021). To determine
the sensitivity of model results to the GHF, we weigh the default GHF field $\mathrm{GHF_{def}}$ against a modified input $\mathrm{GHF_{mod}}$ (Fig. 1b
given the lower regional values in, e.g., Blackwell and Richards, 2004))

$$\mathrm{GHF} = w_{\mathrm{GHF}} \cdot \mathrm{GHF_{def}} + (1 - w_{\mathrm{GHF}}) \cdot \mathrm{GHF_{mod}}, \tag{8}$$

where $w_{\mathrm{GHF}}$ is the weight ranging from 0 to 1. The reference GSM setup uses $w_{\mathrm{GHF}} = 1$. We run sensitivity experiments
for $w_{\mathrm{GHF}} = [0, 0.2, 0.4, 0.6, 0.8]$ applied to both Hudson Bay and Hudson Strait as well as $w_{\mathrm{GHF}} = 0$ applied separately to
Hudson Strait (Fig. S4c) and Hudson Bay (Fig. S4d). When only considering grid cells affected by the GHF modifications
(Fig. 1b), the $w_{\mathrm{GHF}}$ values for the combined Hudson Bay and Hudson Strait experiments correspond to a mean GHF of
$\mathrm{GHF_{ave}} \approx [15, 26, 37, 48, 59]\,\mathrm{mW\,m^{-2}}$ ($\mathrm{GHF_{ave}} \approx 70\,\mathrm{mW\,m^{-2}}$ for reference setup). Additionally, we determine the effects of
a smaller GHF when applied to a larger area (Fig. S4e,f).

## 2.3 Ocean temperature forcing and marine mass loss processes

This section describes both the default and additional ocean temperature forcings employed herein. The additional forcings
are only used in specific experiments to determine the effects of ice shelves (HE shelf forcing $T_{\mathrm{HE}}$, Sec. 2.3.1) and sub-
surface ocean warmings attributed to DO events ($T_{\mathrm{DO}}$, Sec. 2.3.2) in a HE context. To increase confidence in the results of
these experiments and bound the effects of ocean forcings in a HE context, we run additional experiments with more extreme
scenarios (end-member scenarios in Sec. 2.3.3).
The default ocean temperature $T_{\mathrm{ocean}}$ in the GSM is based on the summer weighted ocean temperature field
($0.5 \cdot (T_{\mathrm{ave,Jul:Oct}} + T_{\mathrm{ave,Jan:Dec}})$) derived from the TraCE deglacial simulation run with the Community Climate System
Model Version 3 (CCSM3, Liu et al., 2009). Using a glacial index approach, the ocean temperature chronology is interpolated
between full glacial (last glacial maximum) and present day conditions for all other time slices (Tarasov et al., 2024). Within a
limited *ocean forcing area* (Fig. 2) and below the upper depth limit of $d_{\mathrm{OF}}$ (default value is 250 m), the 2 types of additional

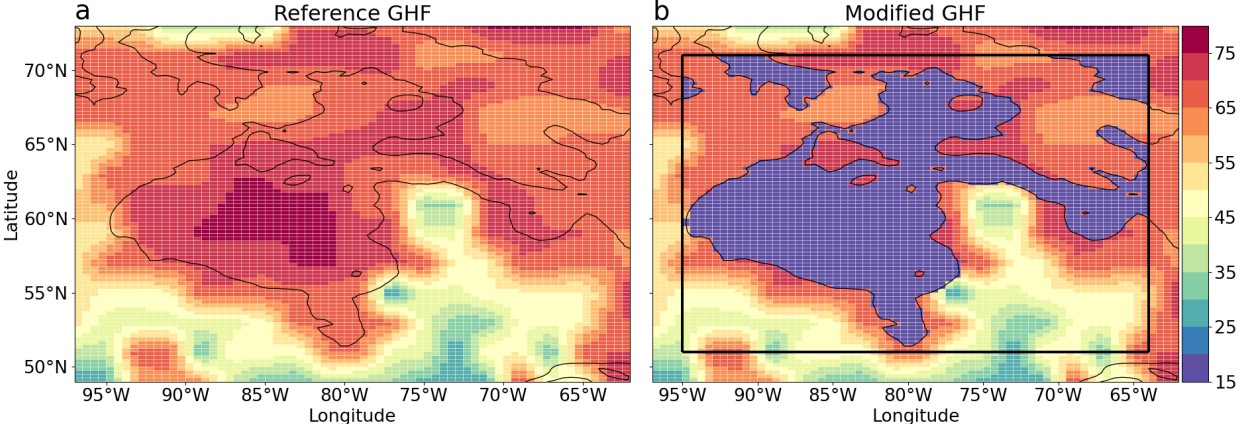

**Figure 1.** GSM input geothermal heat flux (GHF) applied at $4\,\mathrm{km}$ depth in $\mathrm{mW\,m^{-2}}$. Panel a shows the reference input field (Davies, 2013), whereas panel b shows the modified map used for the weighting of the GHF sensitivity experiments (Sec. 2.2). In the modified field, we set GHF$= 15\,\mathrm{mW\,m^{-2}}$ for all grid cells below present-day sea level and within the black square. The black contour line shows the present-day sea level (coastline) used in the GSM.

ocean temperature forcing described below ($T_{\mathrm{HE}}$ and $T_{\mathrm{DO}}$) are added to the background ocean temperature for grid cells with floating ice.

The ocean temperature at the relevant depth is then used to calculate the sub-shelf melt $M_{\mathrm{SSM}}$ and terminus face melt $M_{\mathrm{face}}$. The GSM determines $M_{\mathrm{SSM}}$ based on a parametrization of buoyant meltwater plumes (Lazeroms et al., 2018) and parameter ranges set so that computed melt brackets present-day observations for major Antarctic ice shelves (e.g., Depoorter et al., 2013;

Enderlin and Howat, 2013; Alley et al., 2015). For a floating ice grid cell, the sub-shelf melt is given by:

$$M_{\mathrm{SSM}} = (C_{\mathrm{SSM}} + d \cdot A_{\mathrm{WF}} \cdot (f_{\mathrm{SGWF}})^{0.39}) \cdot T_{\mathrm{ocean,SSM}} \cdot F_{\mathrm{slope}}(\mathrm{SL}) \cdot f_{\mathrm{SSM,slope}} \qquad (9)$$

and $0\,\mathrm{m\,yr^{-1}}$ otherwise. $C_{\mathrm{SSM}}$ is the scaled ensemble parameter for the sub-shelf melt (Table S1), $f_{\mathrm{SGWF}}$ the basal meltwater flux into the sub-shelf melt convection cell, $A_{\mathrm{WF}}$ is a scaling coefficient for the meltwater dependence, and $d$ is the depth. $T_{\mathrm{ocean,SSM}}$ is the ocean temperature at the depth of the ice shelf base and $f_{\mathrm{SSM,slope}}$ is an overall scaling coefficient (Tarasov

et al., 2024). $F_{\mathrm{slope}}$ is a nonlinear function of the maximum lower ice surface slope SL (across the grid cell interfaces with adjacent ice).

Face melt is determined only at the marine ice margin. It is calculated according to

$$M_{\mathrm{face}} = \left(C_{\mathrm{face}} \cdot H_{\mathrm{eff}} + d \cdot A_{\mathrm{WF}} \cdot (f_{\mathrm{SGWF}})^{0.39}\right) \cdot T_{\mathrm{ocean,face}}, \qquad (10)$$

where $C_{\mathrm{face}}$ is the scaled ensemble parameter for the face melt (Table S1). $T_{\mathrm{ocean,face}}$ is the ocean temperature adjacent to the

ice face. This is set with the assumption that face melt water consists of equal parts of local water and shelf water. $H_{\mathrm{eff}}$ is the effective terminal ice thickness defined as $H_{\mathrm{eff}} = \max\left[0.95 \cdot H; 200\,\mathrm{m}\right]$.

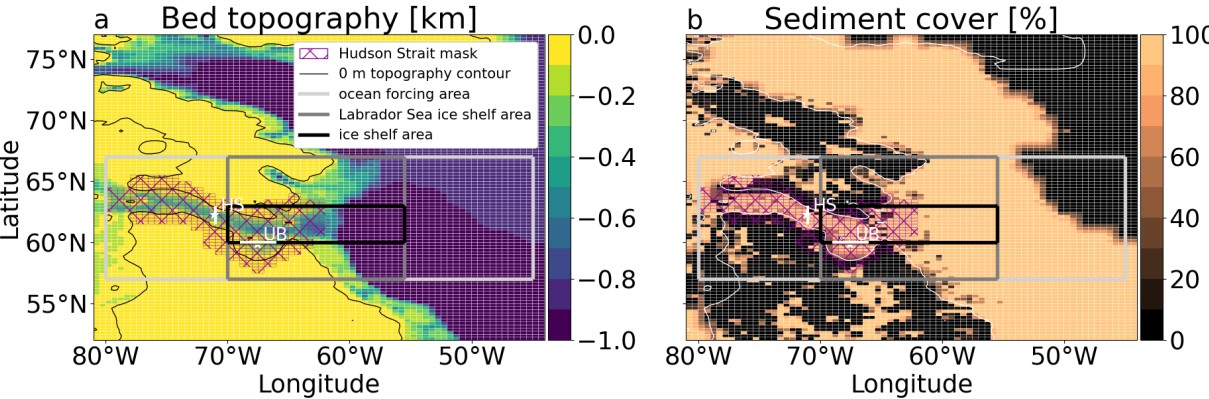

**Figure 2.** GSM input present-day bed topography and sediment cover. The white asterisks and lines indicate the location of Hudson Strait (*HS*) and Ungava Bay (*UB*) ice thickness calculation and flux gate, respectively. The *Hudson Strait mask* is used to determine Hudson Strait ice volume, mean and max basal ice velocity, and warm-based area. The black (left) and white (right) thin contour lines show the present-day sea level (coastline) used in the GSM. The *ocean forcing area* outlines the region affected by the ocean temperature increases discussed in Sec. 2.3. The *Labrador Sea ice shelf area* outlines the area within which the Labrador Sea ice shelf area and volume are calculated. Finally, the *Hudson Strait* area is used to calculate Hudson Strait ice shelf area, ice shelf volume, and the backstress exerted by the floating ice on the Hudson Strait ice stream. To prevent ice sheet growth beyond a stub North-West Greenland, the surface elevation in the corresponding area has been set to well below sea level.

Calving in the GSM is based on a crevasse propagation parameterization (Pollard et al., 2015). To account for the impact of land-fast perennial sea ice, calving is inhibited when the mean $2\,\mathrm{m}$ summer (JJA) temperature falls below $-2.0°$C (Tarasov et al., 2024). However, calving has no temperature control once the ice shelf extends beyond the continental shelf break
(present-day depth $> 860\,\mathrm{m}$).

### 2.3.1 Ice shelf removal

For experiments testing the impact of floating ice removal near Hudson Strait, an increase in the ocean temperature is imposed during HEs ($T_{\mathrm{HE}}$). The ocean forcing is applied to either the entire water column ($d_{\mathrm{OF}} = 0\,\mathrm{m}$) or to an upper ocean forcing limit of $d_{\mathrm{OF}} = 250\,\mathrm{m}$. Sensitivity experiments are conducted for maximum ocean temperature increases of $T_{\mathrm{max,HE}} = [1, 2, 3]°$C
($T_{\mathrm{max,HE}} = 2°$C shown in Fig. 3, amplitudes based on Gibb et al., 2014). To test for a potential warm bias of the GSM ocean temperature forcing, we also run a reverse experiment with a maximum ocean temperature decrease $T_{\mathrm{max,HE}} = -2°$C during HEs. Further details on the calculation of the ice shelf removal ocean forcing are provided in Sec. S4.1.

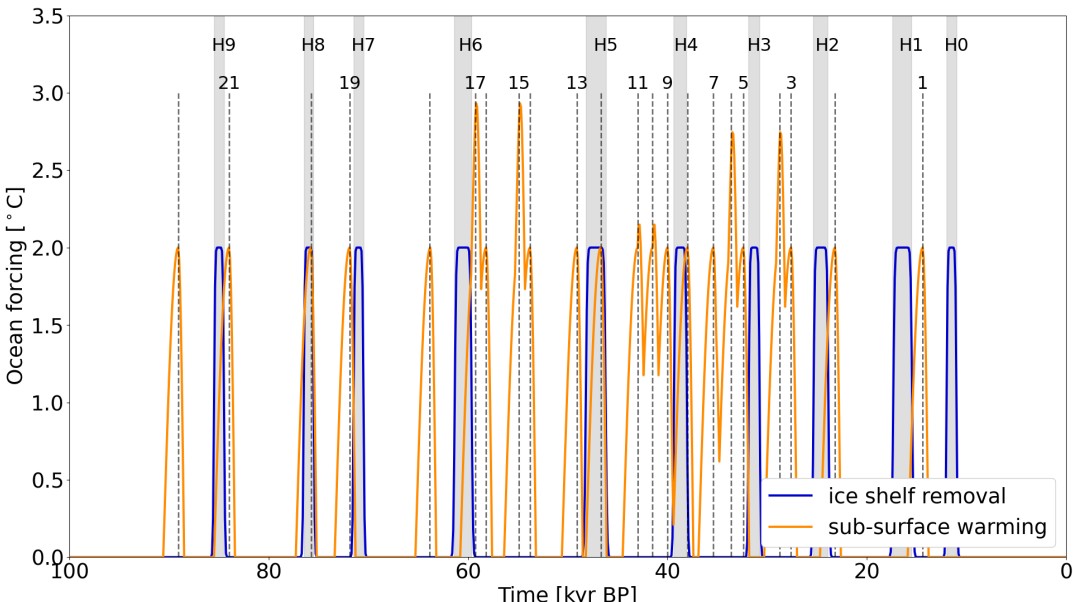

**Figure 3.** $2°C$ ocean temperature forcing used to remove the ice shelves ($T_{HE}$, entire water column) and for the sub-surface ocean warming ($T_{DO}$, below depth $d_{OF} = 250$ m in reference setup). The shaded grey areas represent the timing of HEs based on the average of Table 6.3 in Bradley (2014). The vertical dashed lines indicate the timing of DO events based on peaks in the NGRIP $\delta^{18}$O time series (Bazin et al., 2013; Veres et al., 2013).

### 2.3.2 Sub-surface ocean warming

Similar to the preceding ice shelf removal experiments, the sub-surface warming preceding DO events ($T_{DO}$) is implemented
as ocean temperature anomalies (OTAs). The specification of the OTAs are based on proxies from marine cores in the North
Atlantic and the Nordic Seas at mid depths (Rasmussen and Thomsen, 2004; Marcott et al., 2011). In the initial OTA sensi-
tivity experiment, the maximum increase of sub-surface (below depth $d_{OF} = 250$ m) temperatures of a single OTA is set to
$T_{max,DO} = 2°C$ (Fig. 3), but experiments are conducted for different sub-surface warming depths ($d_{OF} = [100, 500]$ m) and
maximum temperature increases ($T_{max,DO} = [1, 3]°C$). Due to overlaps of individual OTAs, the final temperature increase can
exceed the chosen $T_{max,DO}$ (Fig. 3). Further details on the calculation of the sub-surface warming are provided in Sec. S4.2.

### 2.3.3 End-member scenarios

For all experiments within this section, the additional ocean forcings ($T_{add}$, $T_{DO}$, and $T_{HE}$) are applied to the whole water
column ($d_{OF} = 0$ m) and for all grid cells within the *ocean forcing area* (Fig. 2, not only the grid cells with floating ice). The
experiments are:

EMS$_1$: DO event ocean forcing ($T_{DO}$, $T_{max,DO} = 2°C$)

EMS$_2$:  HE shelf forcing ($T_{\mathrm{HE}}$, $T_{\mathrm{max,HE}} = 2°$C)

EMS$_3$:  additional ocean forcing $T_{\mathrm{add}} = -2°$C applied after 100 kyr BP

EMS$_4$:  no calving after 100 kyr BP

EMS$_5$:  HE shelf forcing ($T_{\mathrm{HE}}$, $T_{\mathrm{max,HE}} = 2°$C) with $T_{\mathrm{add}} = -2°$C ocean forcing applied outside of HEs and after 100 kyr BP

EMS$_6$:  HE shelf forcing ($T_{\mathrm{HE}}$, $T_{\mathrm{max,HE}} = 2°$C) with no calving outside of HEs and after 100 kyr BP

Experiments EMS$_1$ and EMS$_2$ examine the effect of the ocean forcing restriction to grid cells with floating ice. Experiment EMS$_3$ aims to decrease sub-shelf melt, face melt, and calving to enable the growth of larger ice shelves. In the more extreme experiment EMS$_4$, the growth of larger ice shelves is enabled by inhibiting calving entirely. Experiments EMS$_5$ and EMS$_6$ target the collapse of larger ice shelves, with experiment EMS$_6$ being the more extreme scenario.

## 2.4  Experimental design

The GSM configuration in this study uses 52 model input parameters (ensemble parameters). The ensemble parameter space covered within this study is summarized in detail in Tab. S1. A parameter vector holds one value for each of these ensemble parameters. As such, each parameter vector fully specifies how the configuration of the GSM varies between ensemble members. To partly address potential non-linear dependencies of model results on the ensemble parameters, we use a 20 member (20 parameter vectors) high-variance reference ensemble (with respect to ensemble parameters and ice sheet configuration) instead of just a single reference run (1 parameter vector). The creation of the reference ensemble is described in detail in Sec. 2.4.1. The ice volume time series and the ice sheet surface elevation over the entire model domain at 60 and 24 kyr BP are shown for all 20 parameter vectors in Fig. S5 to S7.

For every sensitivity experiment (e.g., different basal sliding law), we then re-run the GSM for all 20 parameter vectors. A list of all sensitivity experiments conducted within this study is shown in Tab. 1. The analysis of each sensitivity experiment in comparison to the reference ensemble is described in detail in Sec. 2.4.2.

### 2.4.1  Reference ensemble

The source ensemble is from a North American history-matching for 52 GSM parameters (Tarasov et al., 2024; Tarasov and Goldstein, 2021). For the experiments herein, $\sim 15000$ coarse resolution ($\Delta_{\mathrm{lon}} = 1°$, $\Delta_{\mathrm{lat}} = 0.5°$) GSM runs from the history matching are initially sieved for total North American ice volume ($V_{\mathrm{tot}}$) and area ($A_{\mathrm{tot}}$) at specific time steps, the last time the center of Hudson Bay (lat $= 61.25°$N, lon $= 84.5°$W) became free of grounded ice ($t_{\mathrm{ice,HB}}$), the southernmost ice extent between $94.7°$W and $80.3°$W (lat$_{\mathrm{ice}}$), and the southernmost ice extent at $69.7°$W (lat$_{\mathrm{ice,Q}}$). The exact sieves are $V_{\mathrm{tot,110}} > 0.29 \cdot 10^7$ km$^3$, $V_{\mathrm{tot,100}} < (V_{\mathrm{tot,110}} - 0.15 \cdot 10^7$ km$^3)$, $V_{\mathrm{tot,60}} > 0.6 \cdot 10^7$ km$^3$, $V_{\mathrm{tot,30}} < (V_{\mathrm{tot,20}} - 0.1 \cdot 10^7$ km$^3)$, $A_{\mathrm{tot,20}} > 12 \cdot 10^3$ km$^2$, 9.0 kyr BP $> t_{\mathrm{ice,HB}} > 7.8$ kyr BP, lat$_{\mathrm{ice}} < 39°$N, and lat$_{\mathrm{ice,Q}} < 45.6°$N. The numbers in the subscripts indicate the time step in kyr BP.

| Sensitivity experiment | # ($\#_{\text{lowGHF}}$) | Difference to reference model setup |
|---|---|---|
| horizontal grid resolution | 1 | $\Delta_{\text{lon}} = 1.0°$, $\Delta_{\text{lat}} = 0.5°$ |
| stricter numerical convergence (Sec. 2.4.2) | 2 (2) | **MNEE$_\mathbf{C}$ and MNEE$_\mathbf{C+I}$** |
| geothermal heat flux (Sec. 2.2) | 2 | $\text{GHF}_{\text{ave}} = [15, 25]\,\text{mW}\,\text{m}^{-2}$ (set values, e.g., Fig. 1b for $15\,\text{mW}\,\text{m}^{-2}$) |
|  | 4 | $\text{GHF}_{\text{ave}} \approx [26, 37, 48, 59]\,\text{mW}\,\text{m}^{-2}$ (weighting between Fig. 1a and b) |
|  | 1 | $\text{GHF}_{\text{ave,HSonly}} = 15\,\text{mW}\,\text{m}^{-2}$ (Fig. S4c) |
|  | 1 | $\text{GHF}_{\text{ave,HBonly}} = 15\,\text{mW}\,\text{m}^{-2}$ (Fig. S4d) |
|  | 1 | revised GHF (Fig. S4e) |
|  | 1 | lower revised GHF (Fig. S4f) |
| basal sliding law (Sec. 2.1) | 2 | (regularized) Coulomb sliding law (Eq. 6) |
|  | 1 | $\boldsymbol{\tau}_{\mathbf{b,GL}} = \boldsymbol{\tau}_{\mathbf{b,W}}$ instead of Eq. 7 |
|  | 1 | $h_{\text{hyd,lim}} = 100\,\text{m}$ |
|  | 1 | Pollard and Deconto (2020) GLT |
| ice shelf removal (HE shelf forcing, Sec. 2.3.1) | 1 | $d_{\text{OF}} = 250\,\text{m}$ and $T_{\text{max,HE}} = 2°\text{C}$ (Eq. S1) |
|  | 4 (1) | $d_{\text{OF}} = 0\,\text{m}$ and $T_{\text{max,HE}} = [-1, 1, \mathbf{2}, 3]°\text{C}$ (Eq. S1) |
|  | 1 | $d_{\text{OF}} = 0\,\text{m}$, $T_{\text{max,HE}} = 3°\text{C}$, and no imposed calving restriction (Sec. 2.3) |
| sub-surface ocean warming (DO event ocean forcing, Sec. 2.3.2) | 3 (1) | $d_{\text{OF}} = 250\,\text{m}$ and $T_{\text{max,DO}} = [1, \mathbf{2}, 3]°\text{C}$ (Eq. S5) |
|  | 2 | $d_{\text{OF}} = [100, 500]\,\text{m}$ and $T_{\text{max,DO}} = 2°\text{C}$ (Eq. S5) |
| end-member scenarios (Sec. 2.3.3) | 6 (6) | **EMS$_1$ to EMS$_6$** |
| glacial isostatic adjustment (Sec. 2.1) | 1 (1) | **no GIA** |
|  | 9 | global visco-elastic GIA with different earth rheologies (Fig. S37) |
|  | 3 (1) | local GIA with $\tau = [3, \mathbf{4}, 5]\,\text{kyr}$ |

**Table 1.** Sensitivity experiments conducted within this study. Each sensitivity experiment differs from the reference model setup only by the changes highlighted here. The reference model setup uses $\Delta_{\text{lon}} = 0.5°$, $\Delta_{\text{lat}} = 0.25°$, $\text{GHF}_{\text{ave}} = 70\,\text{mW}\,\text{m}^{-2}$, the Weertman-type power law (Eq. 1), Eq. 7 at the grounding line (GL), $h_{\text{hyd,lim}} = 10\,\text{m}$, Schoof (2007) grounding line treatment (GLT), no additional ocean forcing, and global visco-elastic glacial isostatic adjustment (GIA). Sensitivity experiments highlighted in bold (last column) were repeated with $\text{GHF}_{\text{ave}} = 25\,\text{mW}\,\text{m}^{-2}$. # and $\#_{\text{lowGHF}}$ represent the number of sensitivity experiments and number of sensitivity experiments repeated with $\text{GHF}_{\text{ave}} = 25\,\text{mW}\,\text{m}^{-2}$ corresponding to each row, respectively. In total, this study encompasses 60 sensitivity experiments run with the same 20 parameter vectors (Sec. 2.4.1, total of 1200 model runs).

For the remaining $\sim 10000$ runs, we use a peak prominence algorithm to determine surges (*SciPy* version 1.6.3, Virtanen et al., 2020). This is similar to the approach used in (Hank et al., 2023). However, due to the more realistic model domain used herein, not every increase in mid-Hudson Strait ice flux (determined at the Hudson Strait flux gate shown in Fig. 2) corresponds to a decrease in Hudson Strait ice thickness and vice versa. Since we are most interested in ice flux changes, the algorithm is applied directly to mid-Hudson Strait ice flux (minimum threshold of $0.0035\,\text{Sv}$, across black vertical line (*HS*) in Fig. 2) instead of the mean Hudson Strait ice thickness time series (as in Hank et al., 2023). We use mid-Hudson Strait ice flux instead of, e.g., the ice flux across the Hudson Strait grounding line, as it is less noisy and a more reliable indicator of large-scale

surges. 2 different sieves are then applied to the surge characteristics. The runs pass if at least one of the following conditions is met:

1. $3 \leq$ #surges $\leq 10$, $0.2\,\mathrm{kyr} \leq$ mean surge duration, $5.0\,\mathrm{kyr} \leq$ mean periodicity $\leq 11.0\,\mathrm{kyr}$ (for surges between $100\,\mathrm{kyr}$ BP and $15\,\mathrm{kyr}$ BP), and/or

   2. $3 \leq$ #surges $\leq 10$ (for surges between 65 kyr BP and 10 kyr BP),

where #surges represents the number of surges in a run. The second sieve is added to increase the number of runs that show surges within the period set by HE proxy records (e.g., Hemming, 2004). The bounds for the mean duration and period are

based on the literature HE estimates in Table 2.

The $\sim 200$ not ruled out yet (NROY) runs are re-submitted at a finer horizontal grid resolution ($\Delta_{\mathrm{lon}} = 0.5°$ and $\Delta_{\mathrm{lat}} = 0.25°$, corresponding to $\sim 25\mathrm{x}25$ km in Hudson Strait). As for the coarse resolution runs, we apply the peak prominence algorithm to mid-Hudson Strait ice flux to determine surges. Since we are now most interested in surges that could explain the IRD layers observed in the North Atlantic (rather than simply sieving for runs that show surges), we reject surges for which

any of the following conditions are fulfilled:

   1. change in mid-Hudson Strait ice flux $< 0.0025$ Sv ($\sim 25\,\%$ of average marine isotope stage 3 (MIS3, $\sim 57$ to $27\,\mathrm{kyr}$ BP, Lisiecki and Raymo, 2005) mid-Hudson Strait ice flux in the GSM)

   2. surge duration $> 3$ kyr (maximum estimated duration of 2.3 kyr in Hemming, 2004)

   3. change in Hudson Strait ice volume $> -5 \cdot 10^3$ km$^3$ (only within area outlined by the *Hudson Strait mask* in Fig. 2,
$\sim 0.8\,\%$ of average MIS3 Hudson Strait ice volume in the GSM)

As there is no proxy record for directly inferring the change in mid-Hudson Strait ice flux or Hudson Strait ice volume during HEs, the above thresholds were chosen after testing various values. They led to the most consistent (with respect to the surge mode across all runs) detection of surges while ensuring that mid-Hudson Strait ice flux increases and Hudson Strait ice volume decreases.

Based on the remaining surges in the fine resolution runs, we hand-pick a high variance (with respect to the surge characteristics) sub-ensemble of 10 runs with less than 3 surges and 10 runs with more than 2 surges between 100 kyr BP and 10 kyr BP ($\sim 50\,\%$ of all fine resolution runs). The ensemble members that show $\leq 2$ surges at the fine resolution (but $> 2$ surges at the coarse resolution) are considered as *likely to surge* and are included to test if, e.g., the repetitive removal of the buttressing effect of an ice shelf can cause ice stream surge cycling in Hudson Strait.

Crashed runs are not considered in any of the above steps. Therefore, the reference ensemble only contains parameter vectors with a successful run completion at the fine horizontal grid resolution.

### 2.4.2   Run comparison

Reference ensemble members with $\leq 2$ and $> 2$ surges are generally analyzed separately. For runs with $> 2$ surges, percentage differences in surge characteristics (number of surges, mean surge duration, mean period between surges, mean increase in

Hudson Strait ice flux, and mean Hudson Strait ice volume change during a surge) for every sensitivity/reference run pair are computed individually and then averaged over all runs (further details in Sec. S5 in Hank et al., 2023). Since a sensitivity experiment/reference run pair uses the same parameter vector, the differences in model results are caused by the sensitivity experiment in question and not, e.g., different climate forcings. Averaging over all sensitivity experiment/reference run pairs (albeit partly) accounts for uncertainties in ensemble parameters. This increases confidence in the identified significant physical

processes in a HE context and reduces the possibility that the modeling response is only due to, e.g., the chosen climate forcing.

To determine if a change in model setup (sensitivity experiment) leads to numerically significant differences in the surge characteristics, we use the **M**inimum **N**umerical **E**rror **E**stimates (MNEEs) threshold initially developed in Hank et al. (2023). The MNEEs are essentially the model response to a sensitivity experiment with imposed stricter (than default) numerical convergence criteria in the ice dynamics solver. More specifically, the final MNEEs for the surge characteristics (SC) in question

(shaded grey regions in Fig. 6) are

$$\mathrm{MNEE(SC)} = \max\left[\mathrm{MNEE_C(SC)}; \mathrm{MNEE_{C+I}(SC)}\right], \tag{11}$$

where $\mathrm{MNEEs_C}$ are the percentage differences of a sensitivity experiment with stricter numerical convergence and $\mathrm{MNEEs_{C+I}}$ the percentage differences of a sensitivity experiment with stricter numerical convergence with increased maximum iterations for the outer Picard loop (from 2 to 3, solving for the ice thickness) and the non-linear elliptic SSA (Shallow-Shelf Approxima-

tion) equation (from 2 to 4, solving for horizontal ice velocities). Differences smaller than the MNEEs should be interpreted as model response not resolvable given the numerical sensitivities. To further increase confidence in model numerics, we extracted yearly ice stream velocities (default time series output is $100\,\mathrm{yr}$) for a few diagnostic grid cells (including Hudson Strait and Hudson Bay) from the reference ensemble and found no significant high-frequency instability ("noisy" ice streams).

For runs with $\leq 2$ surges, the percentage differences in surge characteristics are large due to the small number of surges

or impossible to determine for reference runs with #surges$= 0$. Therefore, we use alternative approaches (e.g., kernel density and time series plots) to determine the effects of physical system processes for the $\leq 2$ #surges sub-ensemble. For the kernel density plots, we use the Python module *seaborn* (version 0.12.2), which uses a Gaussian kernel and automatically sets the bandwidth according to the *SciPy* (version 1.6.3) function *gaussian_ kde(bw_ method='scott')* (yielded the overall best data representation of all bandwidths tested). The density in each panel of a kernel density plot is scaled so that the sum of all

individual integrals within this panel is 1.

## 3 Results

### 3.1 Hudson Strait ice stream surges

The Hudson Strait ice stream is active for most of the last glacial cycle when using the default GHF map (e.g., Fig. 4a). This is contrary to the pattern of long quiescent intervals interspersed by short-lived surges proposed by, e.g., MacAyeal (1993);

Payne (1995); Calov et al. (2002). Instead, the surges are preceded by a reduction (e.g., surges P0, P7 to P9 for parameter vector 1 shown in Fig. 4a) or complete de-activation (or shutdown; P1 to P6) of the Hudson Strait ice stream. Depending on

the background ice flux before the surge, the following ice flux increase occurs either rapidly (P0 to P7) or more gradually (P8 and P9). As per our criteria (Sec. 2.4.1), all identified surges have a decrease in Hudson Strait ice volume. The change is especially pronounced for surges preceded by a complete de-activation of the Hudson Strait ice stream. Times of complete Hudson Strait ice stream de-activation generally coincide with increased buttressing (shown as fraction of the grounding line longitudinal stress, Fig. 4f). The decreased ice shelf volume indicates that the increase in buttressing is due to a decrease in the longitudinal stress rather than an increase in the back stress. Therefore, the changes in buttressing are a consequence of the small Hudson Strait ice flux and not the cause. Instead, the de-activation of the Hudson Strait ice stream is caused by a decrease in the warm-based area (Fig. 4c).

The more gradual increases in Hudson Strait ice flux (P8 and P9) are linked to surges in Ungava Bay (Fig. 4e and S8). Due to the rapid increase in ice flux in Ungava Bay, these surges are still in line with the concept of rapid surge onset, although the resultant increase in Hudson Strait ice flux is more gradual.

Some of the detected surges show an increase in Hudson Strait ice flux without a significant change in the effective pressure or the warm-based area in Hudson Strait (e.g., P1 to P3 in Fig. S9). During these surges, ice transport from Hudson Bay and Foxe Basin through Hudson Strait (and other outlets) towards the ice sheet margin increases (e.g., Fig. S10). The increased downstream ice transport and consequential increase in driving stress eventually increase the ice flux in Hudson Strait itself. Therefore, these surges originate further upstream and are initiated by ice and climate dynamics outside of Hudson Strait.

Surges, particularly those preceded by a complete de-activation of the Hudson Strait ice stream, tend to occur before $65 \, \mathrm{kyr \, BP}$ (e.g., Fig. 4). The absence of complete cessation of ice streaming during MIS3 when using the default deep GHF map is due to increased basal temperatures. This is a result of the growing ice volume and, thereby, thicker ice and larger supply of warm-based upstream ice. The increased basal temperatures also lead to longer surges with reduced ice volume change (Fig. S11).

To get a better understanding of the physical mechanism behind Hudson Strait surges, we analyze surge P1 of parameter vector 1 in greater detail (Fig. 5 and video 01 of Hank, 2023). At $87.5 \, \mathrm{kyr \, BP}$, the ice sheet in Hudson Strait is warm-based, enabling basal melting. Basal water thicknesses reach up to $10 \, \mathrm{m}$ (upper limit of local sub-glacial hydrology model), leading to small effective pressures and, consequently, high basal ice velocities. Due to a combination of cold ice advection from further upstream and/or from the ice stream margins, thinning of the ice stream, and flattening of the ice sheet surface (hence reduced gravitational driving), the basal temperature eventually falls below the pressure melting point. The basal water refreezes and the effective pressure increases by an order of magnitude, increasing the basal drag ($87.0 \, \mathrm{kyr \, BP}$). The ice streaming ceases, allowing the ice thickness to increase.

Due to increased insulation from the cold surface temperatures (thicker ice sheet) and heat contribution from the deformation work, the upstream basal temperature increases (video 02 of Hank, 2023). Once the basal temperature is close to the pressure melting point, basal sliding contributes further heat and the warm-based area starts to extend downstream (video 02 of Hank, 2023). Basal melting then leads to the build-up of a sub-glacial water layer, decreasing the effective pressure and enabling high basal ice velocities. Due to the built-up ice, Hudson Strait ice flux initially increases beyond the pre-ice stream de-activation values before gradually returning to similar ice fluxes, and completing the cycle ($86.1 \, \mathrm{kyr \, BP}$).

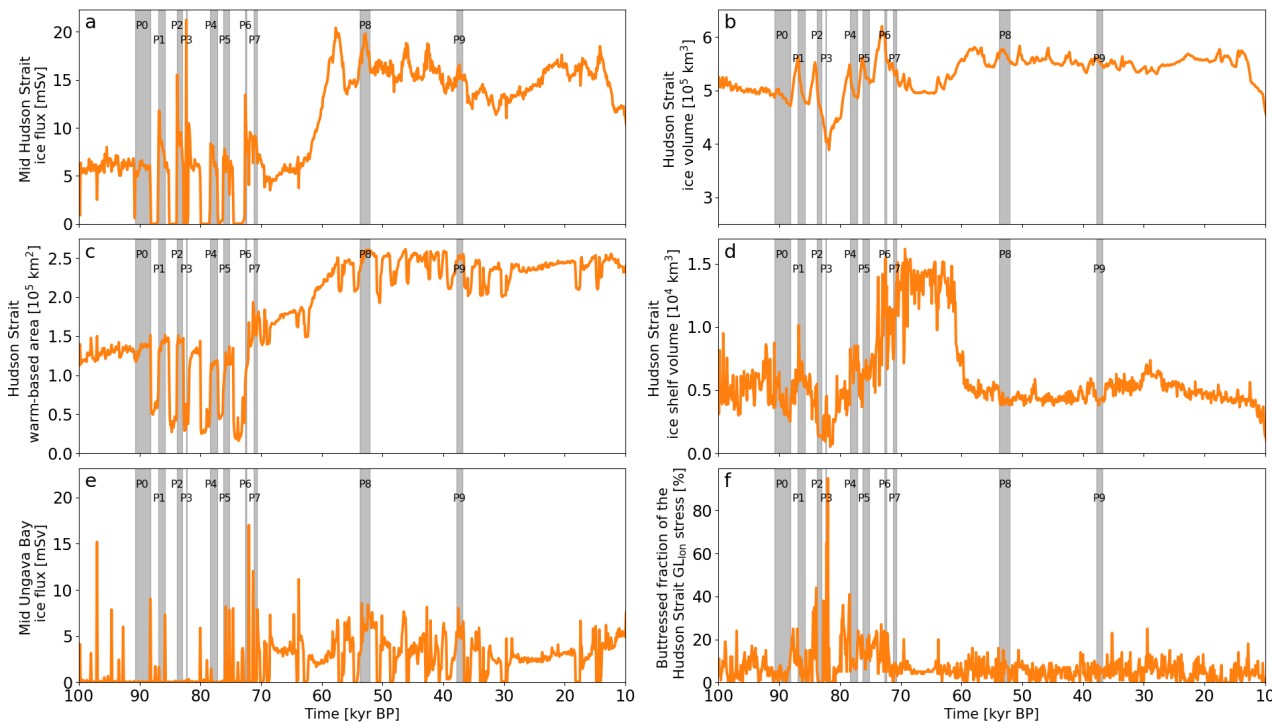

**Figure 4.** Time series of parameter vector 1. The shaded grey areas and black numbers mark Hudson Strait ice stream surges as determined by the automated detection algorithm (Sec. 2.4.1). Hudson Strait ice flux ($1 \text{ mSv} = 10^{-3} \text{ Sv} = 10^3 \text{ m}^3 \text{ s}^{-1}$) is determined at the flux gate marked with *HS* in Fig. 2. The Hudson Strait ice volume and warm-based area are calculated within the *Hudson Strait mask* (Fig. 2). The Hudson Strait ice shelf volume is determined within the *Hudson Strait* area (Fig. 2). The mean buttressing along the Hudson Strait grounding line (only within *Hudson Strait* area in Fig. 2) is given as fraction of the grounding line longitudinal (GL$_{\text{lon}}$) stress ($\left( \frac{\tau_{\text{xx}}}{\tau_{\text{f}}} \right)^{\frac{n}{m_{\text{s}}+1}}$ in Pollard and DeConto (2012)). The last panel shows the northward Ungava Bay ice flux determined at the flux gate marked with *UB* in Fig. 2. Note that the automated detection algorithm does not identify all smaller increases in mid-Hudson Strait ice flux (e.g., between 50 and 40 kyr BP) due to the additional requirement of a Hudson Strait ice volume decrease of at least $5 \cdot 10^3 \text{ km}^3$ (Sec. 2.4.1).

Overall, the mean surge characteristics of the reference setup for the sub-ensemble with more than 2 surges are in agreement with HE estimates in the literature (Table 2). However, the predominance of complete de-activation surges (generally the surges with the largest ice volume change) prior to 65 kyr BP is a major discrepancy (Fig. S11). One inferential caveat is that proxy records can provide only indirect bounds on Hudson Strait ice flux and ice volume change. A comparison to other ice sheet modeling studies (e.g., Schannwell et al., 2023) is not straightforward, as the considered metrics vary significantly (e.g., Hudson Strait area). Using an intermediate mid-Hudson Strait ice flux during a surge (0.01 Sv, e.g., Fig. 4a) and the mean surge duration (1.4 kyr, Table. 2) yields an ice volume discharge of $\sim 44 \cdot 10^4 \text{ km}^3$. This value is well within the range set by modeling and proxy estimates in the literature ($3 \cdot 10^4$ to $946 \cdot 10^4 \text{ km}^3$, Roberts et al., 2014, and references therein).

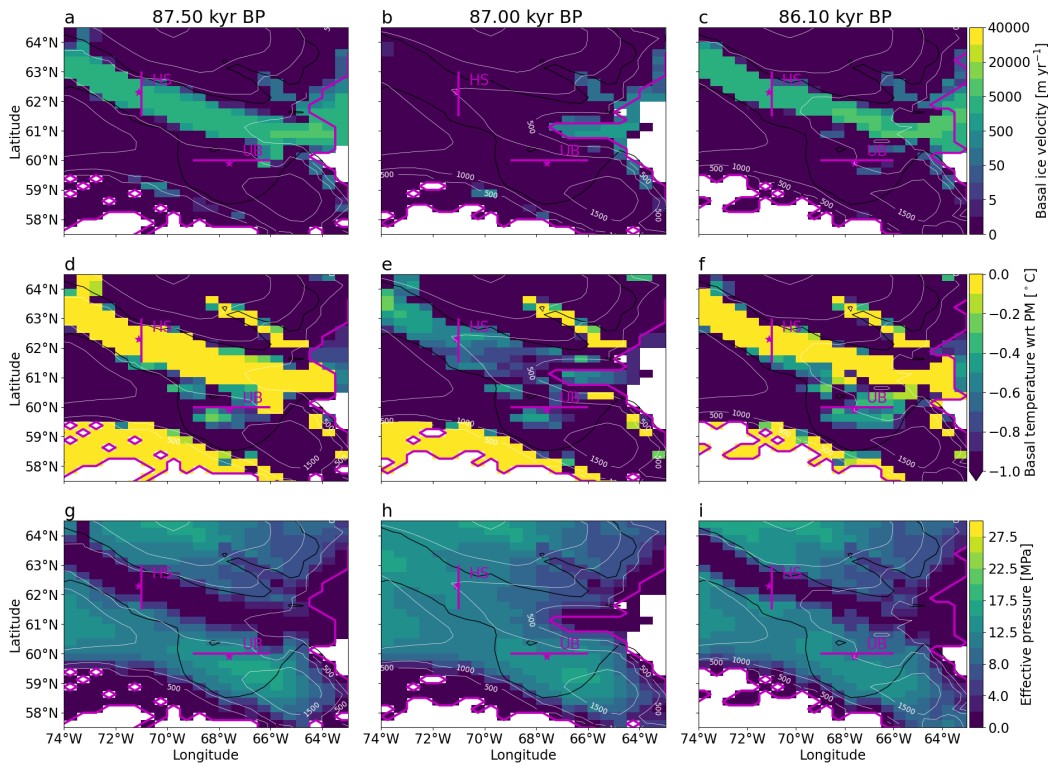

**Figure 5.** Basal ice velocity (first row), basal temperature with respect to the pressure melting point (second row), and effective pressure (third row) for surge P1 of parameter vector 1 (Fig. 4). The 3 time slices (columns) show the active ice stream before the surge (87.5 kyr BP), the quiescent period (87.0 kyr BP), and the surge at its maximum ice flux (86.1 kyr BP). The magenta asterisks and lines indicate the location of Hudson Strait (*HS*) and Ungava Bay (*UB*) ice thickness calculation and flux gate, respectively. The magenta and white contour lines represent the grounding line and ice sheet surface elevation in meters, respectively. The black contour is the present-day sea level (coastline) used in the GSM.

Due to the inclusion of a resolution-dependent basal temperature ramp (Hank et al., 2023), the differences in surge characteristics between the coarse resolution runs (horizontal grid resolution of $\Delta_{\mathrm{lon}} = 1.0°$, $\Delta_{\mathrm{lat}} = 0.5°$) and the reference runs ($\Delta_{\mathrm{lon}} = 0.5°$, $\Delta_{\mathrm{lat}} = 0.25°$) are generally within the MNEEs (Fig. 6). While finer (than the reference setup) horizontal grid resolutions are currently unfeasible in the context of this study, given the results of resolution response testing of surge cycling down to $3.125$ km horizontal grid resolution in Hank et al. (2023), the differences in surge characteristics for finer resolutions are also expected to be within the MNEEs.

## 3.2    Geothermal heat flux

Since the Hudson Strait ice stream is active for most of the last glacial cycle and the GHF is poorly constrained in this area, a lower GHF could increase the time of ice stream inactivation. Therefore, we examine the effect of smaller aver-

| Metric | reference setup (mean $\pm$ SD) | literature HE estimate |
|---|---|---|
| number of surges | $5.4 \pm 2.5$ | 6 to 10 (Table 6.3 in Bradley, 2014) |
| period | $13.6 \pm 8.8$ kyr | 4 to 15 kyr, mean$= 8.0 \pm 2.7$ kyr (Table 6.3 in Bradley, 2014) |
| duration | $1.4 \pm 0.4$ kyr | 0.2 to 2.3 kyr (Hemming, 2004) |
| Hudson Strait ice flux increase | $4.9 \pm 2.0 \cdot 10^{-3}$ Sv | - |
| Hudson Strait ice volume change | $-3.6 \pm 2.3 \cdot 10^{4}$ km$^3$ | - |

**Table 2.** Surge characteristics of the reference setup compared to literature estimates. Only runs with #surges $> 2$ (between 100 kyr BP and 10 kyr BP) are considered. The literature HE estimates are also based on the time between 100 kyr BP and 10 kyr BP. As the automated detection algorithm determines the increase in mid-Hudson Strait ice flux (not the total flux) and the change in Hudson Strait ice volume (not the total discharge), we refrain from providing literature estimates for these two metrics here. Refer to the Sec. 3.1 for a comparison of the ice volume discharge.

age Hudson Bay/Hudson Strait GHFs (GHF$_{\text{ave}}$) on the surge characteristics. Sensitivity experiments are run for GHF$_{\text{ave}} \approx$
$[15, 26, 37, 48, 59]$ mW m$^{-2}$ (Sec. 2.2).

In general, decreasing the GHF in Hudson Strait and Hudson Bay leads to a decrease in the basal temperature, a smaller warm-based area, and, consequently, a decrease in mid-Hudson Strait ice flux. For small enough GHFs (15 mW m$^{-2}$), 9 parameter vectors have fully transitioned from an almost continuously active Hudson Strait ice stream to the classic binge-purge surge mode (e.g., Fig. 7a). The exact transition point depends on the parameter vector in question but generally requires a Hud-
son Strait/Hudson Bay GHF$_{\text{ave}} \leq 37$ mW m$^{-2}$ ($w_{\text{GHF}} = 0.4$). The #surges $> 2$ and #surges $\leq 2$ sub-ensembles both show significant increases in the number of surges and mean Hudson Strait ice volume change for decreasing GHFs (Fig. S13 and S14, Table 3). On the other hand, the mean surge duration significantly decreases for smaller GHFs. Therefore, the colder basal temperatures and increased Hudson Strait ice volume lead to stronger and more rapid surges (e.g., Fig. 7a,b,c).

A key feature of the binge-purge mode is the increase of strong Hudson Strait surges during MIS3 compared to the number
of surges for the near continuous ice streaming mode (e.g., Fig. 7a, S11, and S12). This is in closer correspondence with the actual IRD record.

Separately modifying the GHF in Hudson Strait and Hudson Bay indicates that the GHF modification in Hudson Strait generally has a larger impact than modifications in Hudson Bay (Fig. S15 and S16). However, GHF modifications in both regions are required to obtain continuous binge-purge surge cycling. GHF modifications applied to a larger region (Fig. S4e,f,
based on Blackwell and Richards (2004)) lead to similar conclusions (Fig. S17). Therefore both types of Hudson Strait ice stream surge cycling are consistent within available GHF constraints.

Our analyses below are based on the GSM default GHF map. However, similar conclusions are obtained when applying GHF$_{\text{ave}} = 25$ mW m$^{-2}$ to Hudson Bay and Hudson Strait (Fig. S18, S19, S33, and S34). Any differences in model response are outlined in the specific sections.

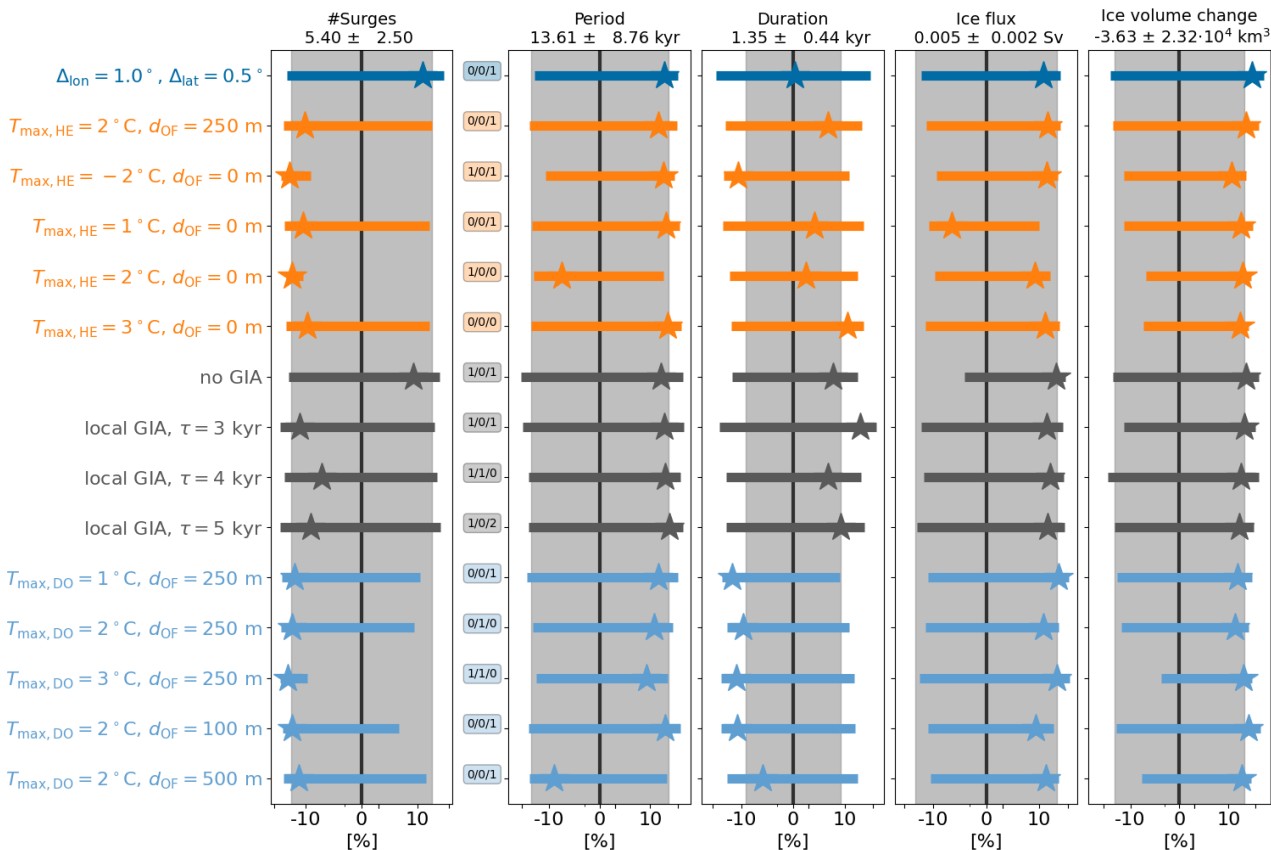

**Figure 6.** Percentage differences in surge characteristics compared to the reference setup for the $> 2$ #surges sub-ensemble. The surge characteristics of the reference setup are shown in the title of each column. The shaded regions represent the MNEEs. The different colours provide visual alignment of the individual model setups. The stars and horizontal bars are the $> 2$ #surges sub-ensemble mean percentage differences and standard deviations, respectively. The three numbers between the first and second column show the number of crashed runs, the number of runs without a surge, and the number of runs with only one surge in the comparison setup. The x-axes are logarithmic. The model setups, from top to bottom, are: $\Delta_{\mathrm{lon}} = 1°$, $\Delta_{\mathrm{lat}} = 0.5°$ horizontal grid resolution, Heinrich Event shelf forcing ($T_{\mathrm{max,HE}} = 2°\mathrm{C}$, $d_{\mathrm{OF}} = 250\,\mathrm{m}$), whole water column ($d_{\mathrm{OF}} = 0\,\mathrm{m}$) Heinrich Event shelf forcing with $T_{\mathrm{max,HE}} = [-2, 1, 2, 3]°\mathrm{C}$ (Sec. 2.3.1), no GIA model, local GIA model that is simply a damped-return to isostatic equilibrium with relaxation time constant $\tau = [3, 4, 5]$ kyr, DO event sub-surface ocean forcing with $T_{\mathrm{max,DO}} = [1, 2, 3]°\mathrm{C}$ and $d_{\mathrm{OF}} = [100, 250, 500]\,\mathrm{m}$ (Sec. 2.3.2).

**3.3 Basal sliding law**

As it is unclear which sliding law should be used in a surge cycling context, we examine the impact of using a (regularized) Coulomb sliding law (Eq. 6) in combination with the default Weertman-type power law (Eq. 1 and 7). Generally, the Coulomb sliding law experiments are numerically more unstable, leading to only 6 successfully completed runs for the Coulomb (Eq. 6a, 14 runs crashed) and 7 for the regularized Coulomb sliding law (Eq. 6b, 13 runs crashed). Additionally, the remaining runs

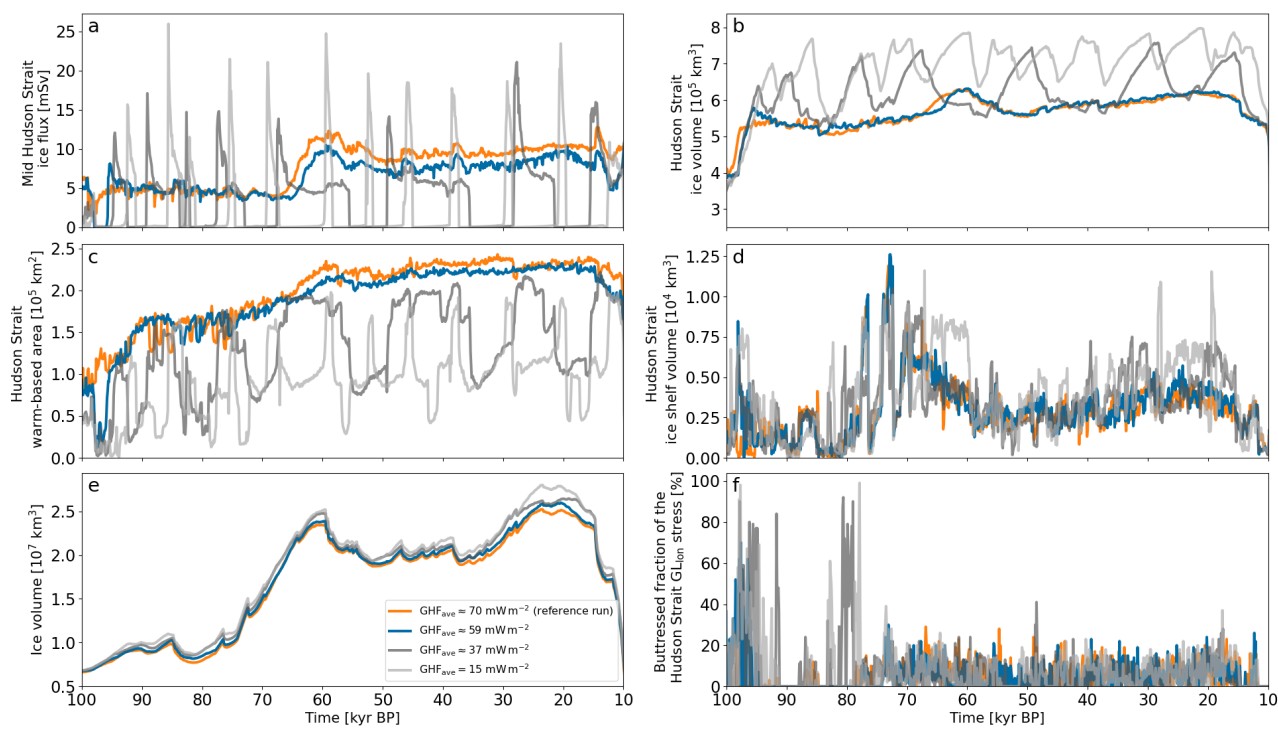

**Figure 7.** Time series of parameter vector 18 for different GHFs. The Hudson Strait ice stream surges are not highlighted for clarity. Panel e shows the overall North American ice volume. Otherwise as Fig. 4.

| Metric | reference setup (mean $\pm$ SD) | $\mathrm{GHF_{ave}} = 15\,\mathrm{mW\,m^{-2}}$ (mean $\pm$ SD) | regularized Coulomb sliding law (mean $\pm$ SD) |
|---|---|---|---|
| number of surges | $5.4 \pm 2.5$ | $10.6 \pm 5.3$ | $52.4 \pm 18.8$ |
| period | $13.6 \pm 8.8$ kyr | $10.7 \pm 5.5$ kyr | $1.6 \pm 0.5$ kyr |
| duration | $1.4 \pm 0.4$ kyr | $0.7 \pm 0.3$ kyr | $0.3 \pm 0.2$ kyr |
| Hudson Strait ice flux increase | $4.9 \pm 2.0 \cdot 10^{-3}$ Sv | $23.0 \pm 13.0 \cdot 10^{-3}$ Sv | $27.8 \pm 8.8 \cdot 10^{-3}$ Sv |
| Hudson Strait ice volume change | $-3.6 \pm 2.3 \cdot 10^{4}$ km$^3$ | $-6.6 \pm 1.1 \cdot 10^{4}$ km$^3$ | $-4.1 \pm 1.3 \cdot 10^{4}$ km$^3$ |

**Table 3.** Surge characteristics for 3 different setups: the reference setup, a setup with the GHF in Hudson Bay and Hudson Strait set to $15\,\mathrm{mW\,m^{-2}}$ (Sec. 2.2), and a setup using a combination of the default Weertman-type power law (Eq. 1) and regularized Coulomb sliding law (Eq. 6b and 7). Only runs with #surges $> 2$ (between 100 kyr BP and 10 kyr BP) are considered (10 for the reference setup, 13 for $\mathrm{GHF_{ave}} = 15\,\mathrm{mW\,m^{-2}}$, 7 for the regularized Coulomb sliding law).

show an average increase in the model run time of $\sim 447.5\,\%$ and $\sim 32.5\,\%$ for the Coulomb and regularized Coulomb sliding law, respectively.

The limited number of successful runs and increase in model run time hinder determining the effects of a Coulomb sliding law, especially in the case without regularization (Fig. S20 and S21). Therefore, the analysis below only examines the regularized Coulomb sliding law. However, both forms of Coulomb basal drag law lead to a noisy (large number of peaks in short succession) mid-Hudson Strait ice flux time series (e.g., Fig. 8a), posing a challenge for the automated peak prominence detection algorithm (with potential overestimation of the number of surges and overlap of surges). While smoothing the mid-Hudson Strait ice flux time series and/or adding an additional requirement (e.g., minimum time difference between detected peaks) would likely minimize these issues, it would also lead to a loss of information and obscure the comparison to the other setups. Therefore, we continue the analysis without these changes.

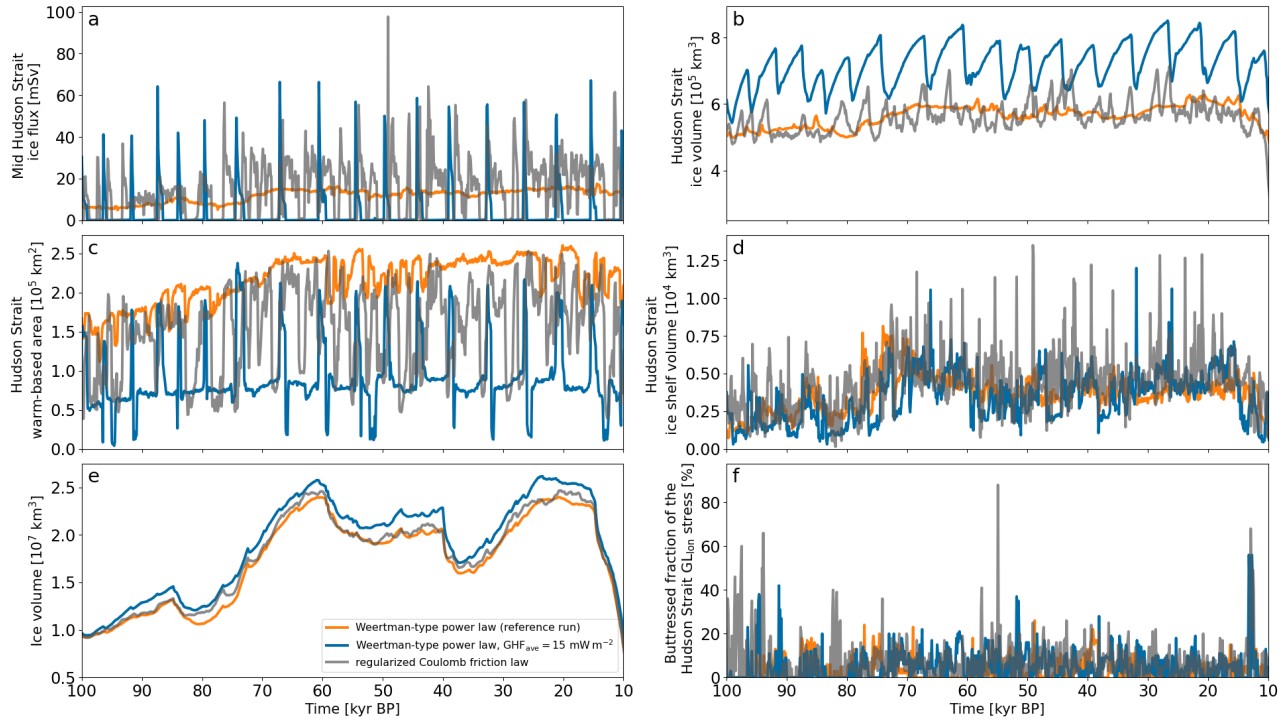

**Figure 8.** Time series of parameter vector 16 for different sliding laws (Weertman-type power law exponent is 2) and geothermal heat fluxes. Panel e shows the overall North American ice volume. Otherwise as Fig. 4.

The most significant impact of using the regularized Coulomb sliding law is a shift to the classic binge-purge surge cycling (e.g., Fig. 8a). While a change in surge mode also occurs for a lower GHF, using the regularized Coulomb sliding law leads to more frequent binge-purge surges (Table 3, Fig. 8a and 9). This conclusion holds even when considering an overestimation of the detected surges due to the noisy mid-Hudson Strait ice flux time series when using the regularized Coulomb sliding law. All successful regularized Coulomb sliding law runs show surges, even those with parameter vectors that have a continuous ice stream in the $GHF_{ave} = 15\,mW\,m^{-2}$ experiment (e.g., Fig. S22).

Due to the large increase in the number of surges, the mean period for the regularized Coulomb sliding law is well below the literature estimates (Table 2 and 3). Given the differences in the basal sliding law, an assessment of whether the two different basal sliding laws (Weertman and Coulomb) are implemented with scale-matched parameter values would be ambiguous. However, the regularized Coulomb (with default GHF) and $\mathrm{GHF_{ave}} = 15\,\mathrm{mW\,m^{-2}}$ (with Weertman-type power law) ensembles have comparable ranges for Hudson Strait ice flux increases, while the latter has larger mean Hudson Strait ice volume changes during surges (primarily due to the longer surge duration, Table 3). Furthermore, our chosen basal drag coefficient ensemble parameters cover typical ranges (Table S1).

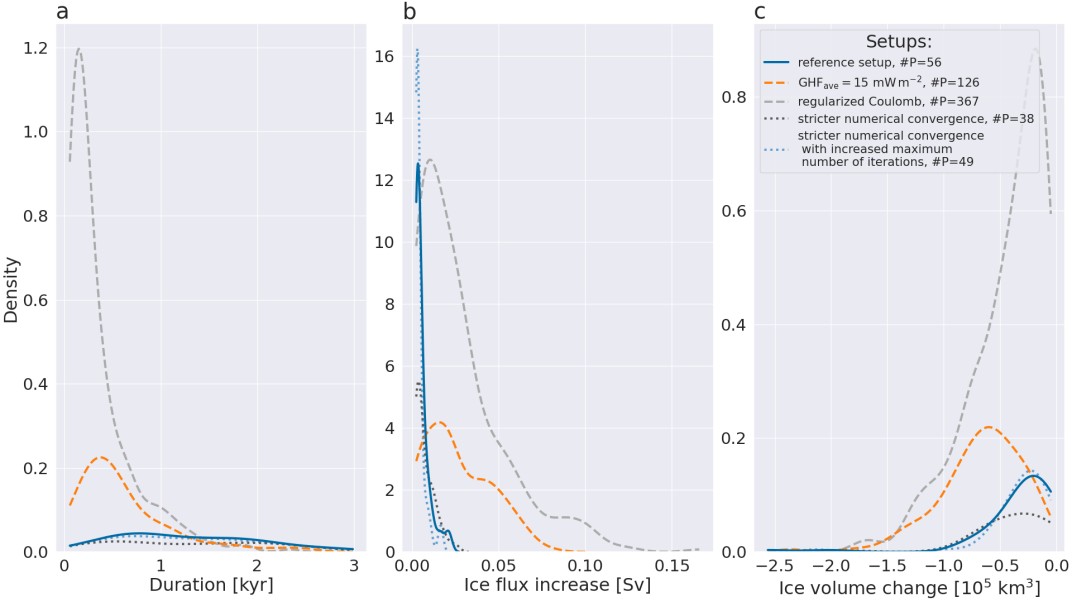

**Figure 9.** Kernel density plot for the whole ensemble (20 parameter vectors). The $\mathrm{GHF_{ave}} = 15\,\mathrm{mW\,m^{-2}}$ and MNEEs setups use the Weertman-type power law. The regularized Coulomb sliding law and MNEEs setups use the default $\mathrm{GHF_{ave}} \approx 70\,\mathrm{mW\,m^{-2}}$. Note that 2 runs crashed for the $\mathrm{GHF_{ave}} = 15\,\mathrm{mW\,m^{-2}}$ setup and 13 for the regularized Coulomb sliding law. Crashed runs are not considered here. #P indicates the total number of surges across all successful runs of the corresponding ensemble.

Only using the Weertman-type power law to calculate the ice flux across the grounding line ($\boldsymbol{\tau_{b,GL}} = \boldsymbol{\tau_{b,W}}$ instead of Eq. 7) and otherwise the reference setup (Eq. 1) has no significant effect on the surge characteristics (Fig. S20 and S21). Therefore, uncertainties in the appropriate grounding line ice flux treatment are not critical.

Due to the numerically more unstable runs, the large increase in model run time, and the surge detection issues, we refrain from repeating all sensitivity experiments within this study with a (regularized) Coulomb sliding law.

### 3.4 Ice shelf removal

The hypothesized ice shelf collapse trigger for surge initiation is contingent on a large enough ice shelf to provide adequate buttressing of the Hudson Strait ice stream. However, the maximum area covered by an ice shelf across all reference runs is

$\sim 0.5 \cdot 10^5$ and $\sim 1.0 \cdot 10^5$ km$^2$ (Fig. 10) of *Hudson Strait* (total area of $\sim 2.6 \cdot 10^5$ km$^2$, Fig. S23) and the *Labrador Sea ice shelf area* (total area of $\sim 8.7 \cdot 10^5$ km$^2$, Fig. 11), respectively (both areas outlined in Fig. 2). The mean ice shelf cover never exceeds $0.4 \cdot 10^5$ km$^2$ in the *Labrador Sea ice shelf area*. These rather small ice shelves have only limited potential to buttress the Hudson Strait ice stream.

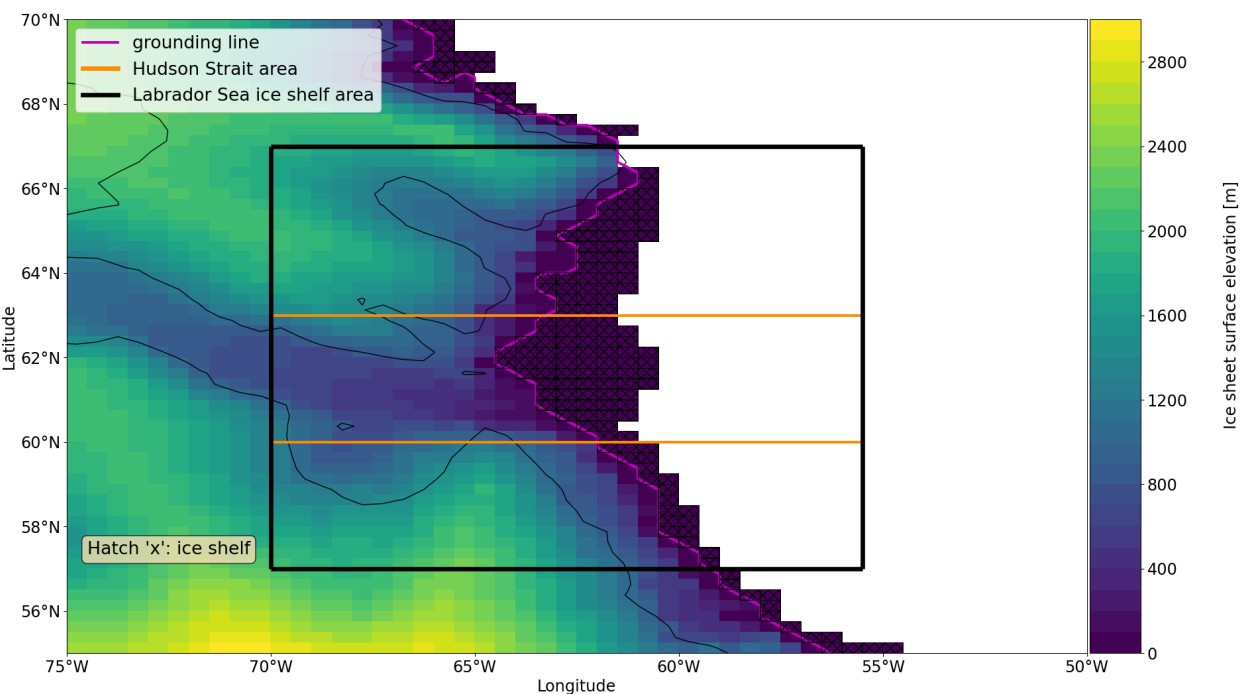

**Figure 10.** Ice sheet surface elevation at 70 kyr BP (time step closest to the maximum Hudson Strait ice shelf area; reference run in Fig. S24) for parameter vector 1. The black contour is the present-day sea level (coastline) used in the GSM.

Given system uncertainties, we examine the change in ice shelf area when decreasing the ocean temperature by $2°$C (entire water column, EMS$_3$ in Sec. 2.3.3) and inhibiting calving in the *ocean forcing area* (EMS$_4$ in Sec. 2.3.3, area outlined in Fig. 2). Decreasing the ocean temperature leads only to a minor increase in Labrador Sea ice shelf cover (Fig. 12). When completely inhibiting calving, the Labrador Sea ice shelf cover increases by at least a factor of 4, at times covering all grid cells in the *Labrador Sea ice shelf area* not covered by grounded ice (maximum of $\sim 4.0 \cdot 10^5$ km$^2$, Fig. 12). However, even

with the extreme calving restriction, some parameter vectors show only minor Labrador Sea ice shelves (e.g., Fig. S30). This demonstrates the breadth of sub-shelf melt across our ensemble (maximum melt rates in EMS$_4$ runs vary between $\sim 22\,\mathrm{m\,yr}^{-1}$ ($1.6°$C ocean temperature) and $\sim 165\,\mathrm{m\,yr}^{-1}$ ($4.7°$C ocean temperature)) and the resultant bound of process uncertainties. Nonetheless, as the maximum ocean temperature in the *Labrador Sea ice shelf area* when applying $T_\mathrm{add} = -2°$C (EMS$_3$) is within $1°$C of the freezing point for 10 (out of 20) runs, calving is the main restricting factor for the growth of large ice shelves.

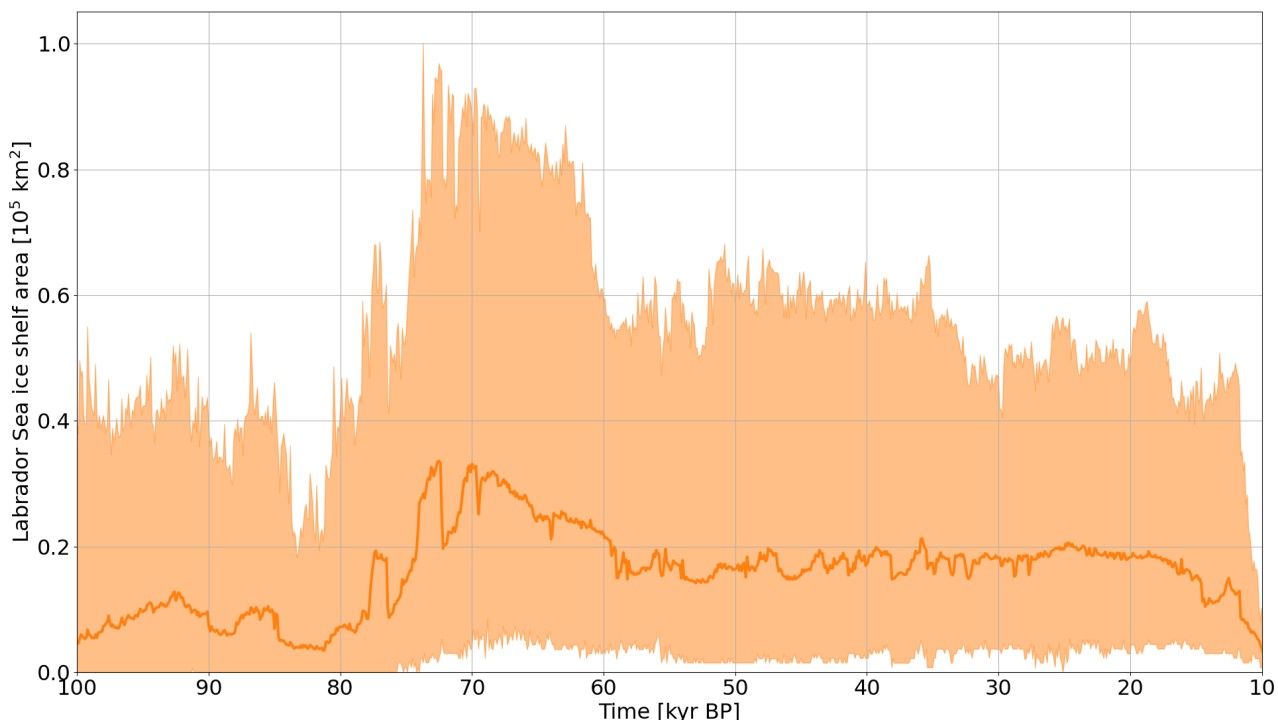

**Figure 11.** Labrador Sea ice shelf cover in the *Labrador Sea ice shelf area* outlined in Fig. 2 (total area of $\sim 8.7 \cdot 10^5$ km$^2$). The thick line represents the mean of the 20 run ensemble. The shaded area marks the minimum and maximum of the ensemble. The maximum ice shelf area at 70 kyr BP is shown in Fig. 10.

To test the role of ice shelf buttressing (along with potential grounding line destabilization from submarine melt) in a Heinrich Event context, we apply different ocean temperature forcings (with default calving). In particular, we adjust the ocean temperature during HEs by a maximum temperature change $T_{\mathrm{max,HE}} = 2$°C below 250 m of the sea level, and $T_{\mathrm{max,HE}} = [-2, 1, 2, 3]$°C for the entire water column (Sec. 2.3.1 and 2.3.3). Except for the surge duration, none of these "HE shelf forcing" experiments significantly affects the surge characteristics of the $> 2$ #surges sub-ensemble (Fig. 6 as well as EMS$_2$ and EMS$_5$ in S31). Even at $T_{\mathrm{max,HE}} = 3$°C, there are only minor differences between the reference and HE shelf forcing runs (e.g., Fig. S24).

The $\leq 2$ #surges sub-ensemble generally shows only small changes in the number of surges for the "HE shelf forcing" experiments (Fig. S25 as well as EMS$_2$ and EMS$_5$ in Fig. S32). The maximum difference (increase in the total number of surges in the sub-ensemble from 2 to 7) occurs for $T_{\mathrm{max,HE}} = 1$°C and below depth $d_{\mathrm{OF}} = 0$ m. In comparison, the minimum increase in the total number of surges over the $\leq 2$ #surges sub-ensemble (10 runs) for the GIA experiments is 10, and the MNEE experiments show a maximum difference of 2 (Fig. 16). Furthermore, the onset of the additional surges does not necessarily align with the ocean forcing (e.g., Fig. S26). The increase in surge number is a result of slightly different ice configurations rather than a direct response to the removal of the ice shelves and their potential buttressing.

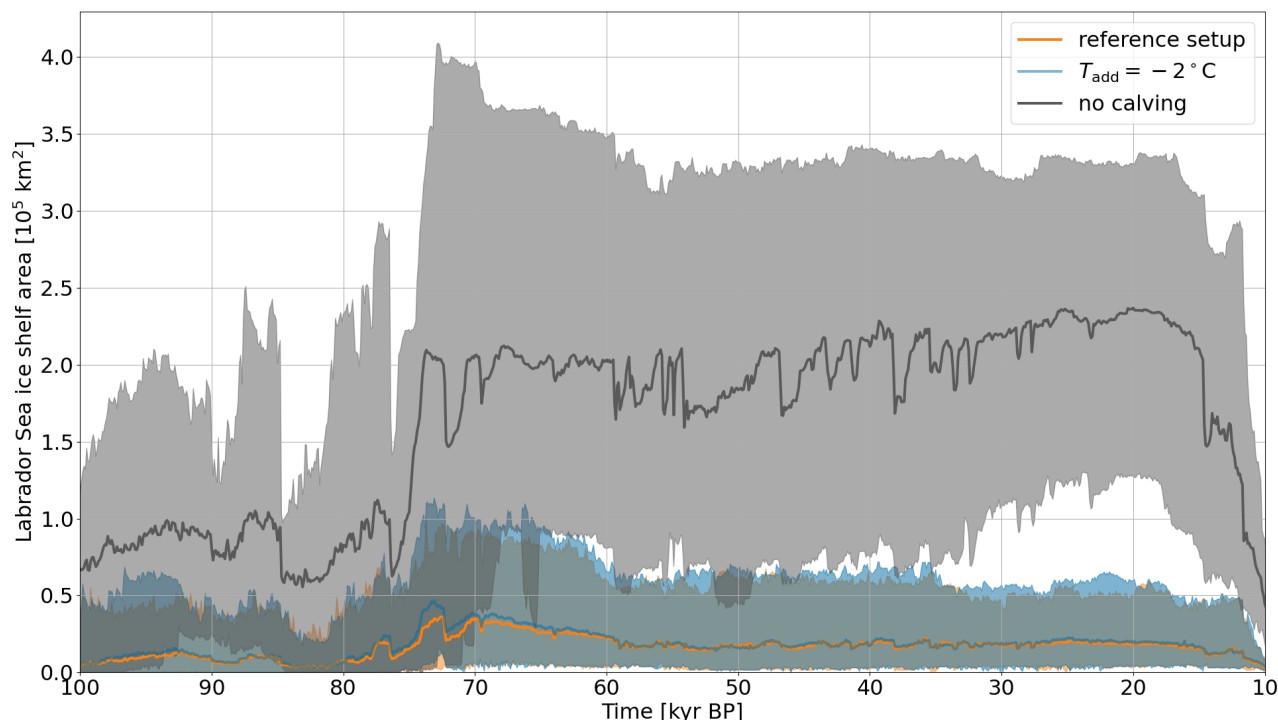

**Figure 12.** Labrador Sea ice shelf cover in the *Labrador Sea ice shelf area* for the reference setup compared to setups with $2°$C colder ocean temperatures and without calving in the *ocean forcing area* (EMS$_3$ and EMS$_4$; both areas outlined in Fig. 2). The thick line represents the mean of $18$ runs (the runs for parameter vectors 8 and 15 crashed in both comparison setups and were not included for all setups shown). The shaded area marks the minimum and maximum of the remaining runs.

During cold climatic conditions, the $2$ m summer surface temperatures near the ice shelves stay below $-2.0°$C, allowing calving only where the ice shelf extends beyond the continental shelf break (e.g., Fig. S27, see Sec. 2.3 for details). To ensure the minor ice shelf sensitivity to ocean temperature changes during HEs is not solely a consequence of this calving restriction, we also run an experiment with a maximum ocean temperature forcing during HEs $T_{\mathrm{max,HE}} = 3°$C applied to the whole water column ($d_{\mathrm{OF}} = 0$ m) and without any imposed restriction on calving within the *ocean forcing area* (Fig. 2). This has no significant effect (Fig. S28).

Therefore, the small effect of the ocean temperature forcing on surges of the Hudson Strait ice stream is a consequence of the relatively small ice shelf in front of Hudson Strait, providing insignificant buttressing. While the experiments with colder ocean temperatures ($-2°$C only during HEs and $-2°$C after $100$ kyr BP) slightly increase the ice shelf cover, a complete calving shutdown is required to build up large ice shelves (e.g., Fig. 12). The larger ice shelves provide increased buttressing, leading to more gradual changes in mid-Hudson Strait ice flux, a more stable Hudson Strait ice volume, and consequentially fewer surges (Fig. S31 and S35).

Using the HE ocean forcing with a maximum temperature increase of $T_{\mathrm{max,HE}} = 2°C$ and no calving outside of HE forcing and after 100 kyr BP ($\mathrm{EMS_6}$) leads to a rapid collapse of the large ice shelves during HE forcing intervals (due to calving and warmer ocean temperatures, Fig. 13d). The ice shelf disintegration increases Hudson Strait ice flux, decreasing Hudson Strait ice volume. Even in this extreme scenario, mid-Hudson Strait ice flux and Hudson Strait ice volume changes are relatively small (e.g., Fig. 13a,b). However, the timing of at least 1 mid-Hudson Strait surge is directly affected by the ocean forcing and consequential reduction in buttressing for 10 (out of 20) parameter vectors.

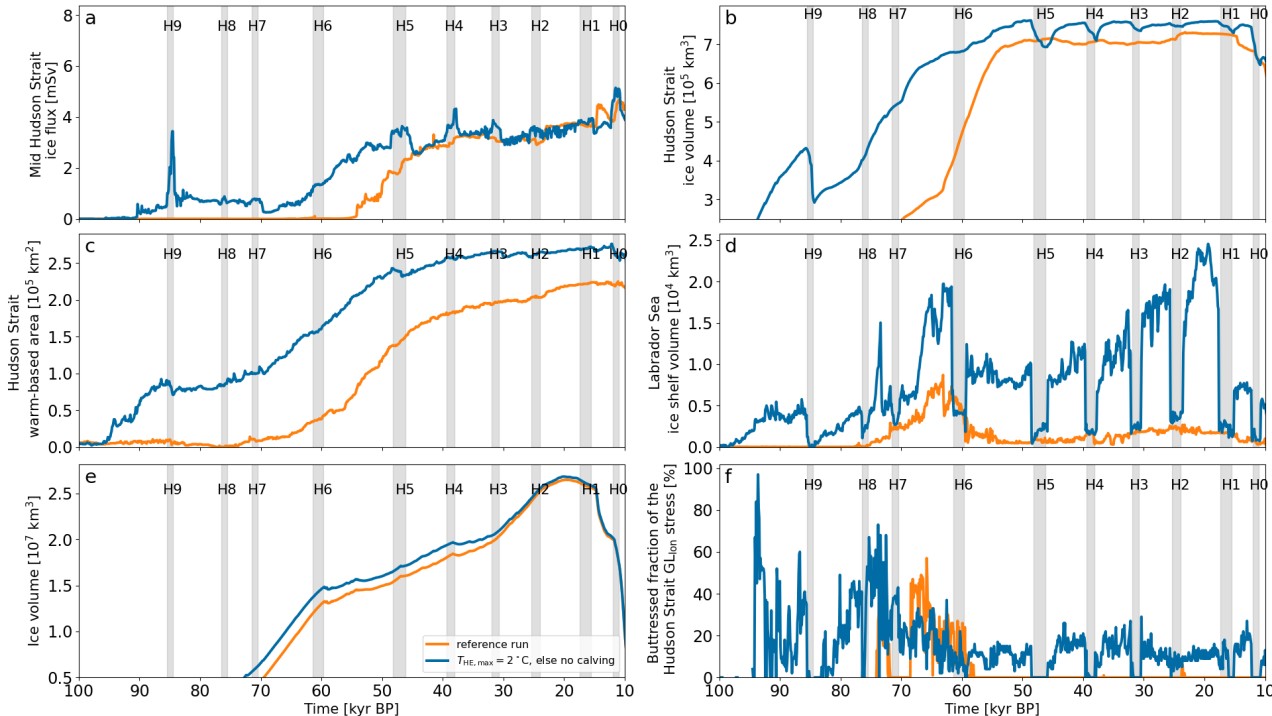

**Figure 13.** Time series of parameter vector 14 for the reference setup compared to the HE ocean forcing with a maximum temperature increase of $T_{\mathrm{max,HE}} = 2°C$ (shaded grey areas) and no calving outside of HEs and after 100 kyr BP ($\mathrm{EMS_6}$ in Sec. 2.3.3). The Labrador Sea instead of Hudson Strait ice shelf volume is shown (panel d). Panel e shows the overall North American ice volume. Otherwise as Fig. 4.

## 3.5 Adding underwater warming pulses

As for the ocean forcing associated with HEs, the more frequent DO event sub-surface ocean warming has no significant effect on the surge characteristics, except for the surge duration (Fig. 6 and $\mathrm{EMS_1}$ in Fig. S31). Similarly, the $\leq 2$ #surges sub-ensemble shows minor changes in the number of surges when applying the DO event ocean forcing (Fig. 14 and $\mathrm{EMS_1}$ Fig. S32). The only exception is $T_{\mathrm{max,DO}} = 2°C$, $d_{\mathrm{OF}} = 250$ m, which increases the total number of surges across the sub-ensemble from 2 to 11 (spread across 3 runs, maximum increase of 4 surges per run). Considering that there are 22 ocean temperature increases per run, this is still a rather small increase. Furthermore, the additional events do not necessarily align

with increased ocean temperatures. As for the ice shelf removal experiments, the increase in surges is a consequence of small changes in the overall ice configuration, particularly in Hudson Strait, rather than a direct response to the ocean forcing itself (e.g., Fig. S29).

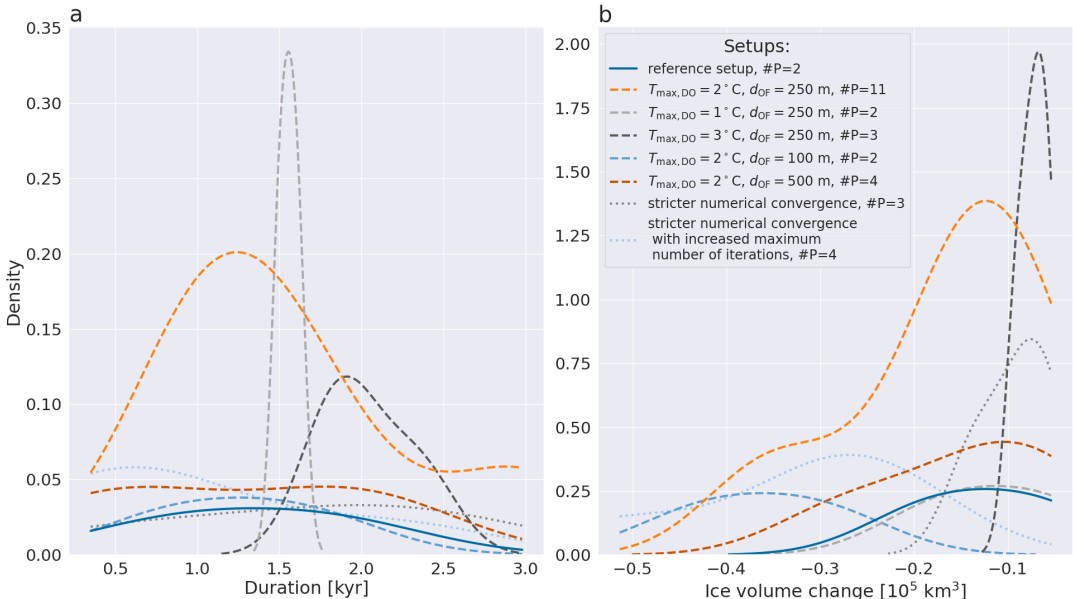

**Figure 14.** Kernel density plot for the $\leq 2$ #surges sub-ensemble (10 parameter vectors). The reference and MNEEs setups do not use an additional ocean forcing. #P indicates the total number of surges across all runs of the sub-ensemble.

Therefore, the ocean forcing and the consequential increase in sub-shelf melt and face melt are insufficient to significantly affect the surge characteristics (number of surges, mean surge duration, mean period between surges, mean increase in Hudson Strait ice flux, and mean Hudson Strait ice volume change during a surge) or trigger new surges. However, sub-surface ocean warming can indirectly affect the overall timing of surges through changes in ice sheet evolution.

## 3.6 Glacial isostatic adjustment

At first, we determine the effects of GIA on the overall ice sheet by comparing the reference setup to runs without GIA. On average, GIA leads to thicker ice sheets (Fig. 15c). The bed depression caused by the weight of the ice sheet lowers the ice sheet surface elevation. Due to the atmospheric lapse rate and the Clausius–Clapeyron formula for saturation vapour pressure, precipitation (and therefore accumulation over most of the ice sheet) generally increases for a decreasing surface elevation (except where orographic forcing is strong). This leads to an overall thicker ice sheet than without GIA (Fig. 15b,c).

Close to the ice sheet margin, a reduced ice sheet surface elevation (due to GIA) has the potential to increase the ablation zone. However, the reduced ice sheet surface elevation could also lower the driving stress near the margin and decrease the ice flux to the ablation zone. While the bed topography under the ice sheet, including the marginal areas, is generally depressed

(Fig. S36, leading to a lower ice sheet surface elevation for the same ice thickness), the total melt tends to be slightly smaller with GIA (Fig. 15d). It is difficult to disentangle the exact underlying cause as various processes affect surface melt. As this is not our primary focus, we defer exploring the nuanced effects of GIA on surface melt to future studies.

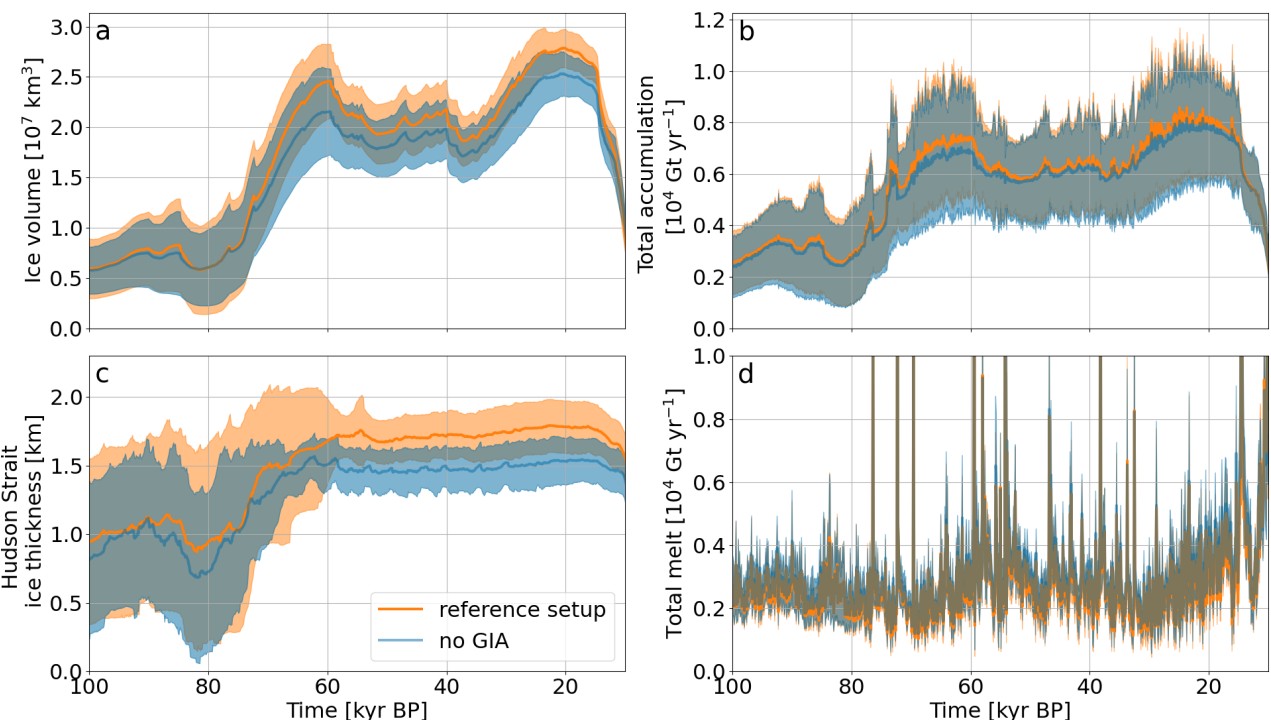

**Figure 15.** Ice sheet volume, total ice sheet accumulation, Hudson Strait ice thickness (Hudson Strait mask outlined in Fig. 2), and total ice sheet melt for the reference setup (global GIA model) and runs without GIA. The thick lines and shaded areas represent the mean and mean±standard deviation of 18 runs, respectively (the runs for parameter vectors 8 and 15 crashed in the comparison setup and were not included).

Before analyzing the effects of GIA on the surge characteristics, we determine the ice sheet's sensitivity to different earth rheology models. All 20 parameter vectors (whole ensemble) were run with 9 different earth rheologies. The rheologies differ
in the thickness of the Lithosphere ($d_L$), and the viscosity of the upper and lower mantle ($\eta_{um}$ and $\eta_{lm}$, respectively). The sensitivity of the mean (across all 20 parameter vectors) North American ice volume to the earth rheology is generally small. The largest differences occur between 60 and 40 kyr BP, with a maximum difference of $0.3 \cdot 10^7$ km$^3$ at $\sim 50$ kyr BP (Fig. S37). Similarly, the surge characteristics show minor sensitivities to a change in the earth rheology. However, a thinner Lithosphere and a smaller upper mantle viscosity tend to favour shorter surges (Fig. S38).
The number of surges in the $\leq 2$ #surges sub-ensemble increases significantly for all experiments with local GIA (a damped-return to isostatic equilibrium with relaxation time constant $\tau = [3, 4, 5]$ kyr) and without GIA (Fig. 16). This is likely due to a smaller North American and Hudson Strait ice volume and the resulting change in basal temperature (e.g., Fig. S39). The larger

ice sheet in the reference setup (global GIA) leads to more stable basal temperatures and a continuously active Hudson Strait ice stream. On the other hand, the smaller ice sheet in runs with local GIA and without GIA leads to colder basal temperatures, reduced Hudson Strait warm-based area (Fig. S40), and at times complete de-activation of the Hudson Strait ice stream. As described in Sec. 3.1, this de-activation eventually leads to a surge.

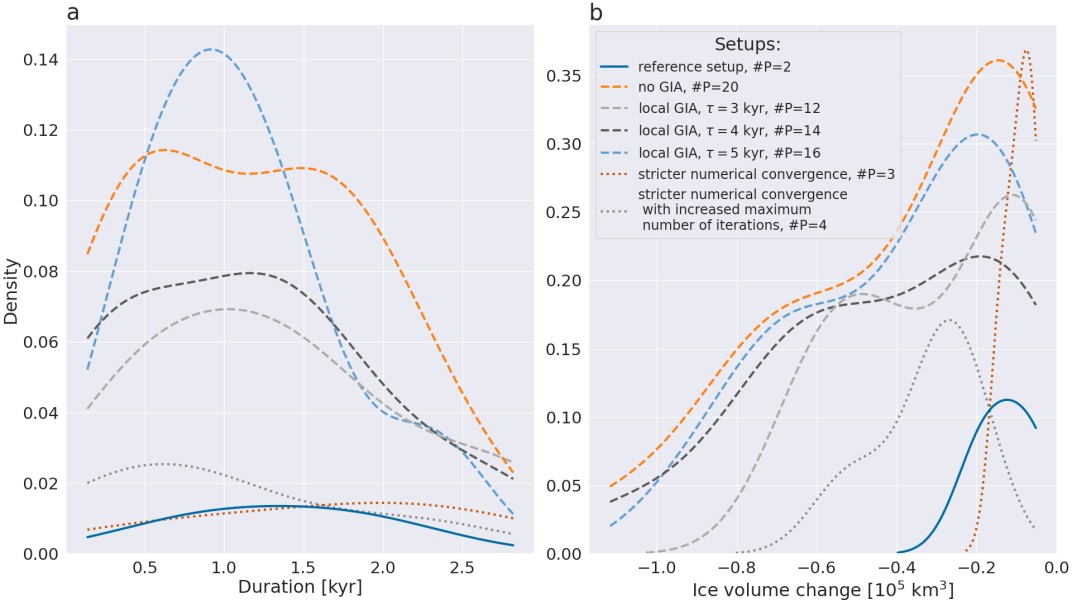

**Figure 16.** Kernel density plot for the $\leq 2$ #surges sub-ensemble. The reference and MNEEs setups use a global GIA model. #P indicates the total number of surges across all runs of the sub-ensemble.

For the $> 2$ #surges sub-ensemble, using local GIA leads to a similar response as completely deactivating GIA, except for the number of surges. The number of surges increases without GIA and decreases for all local GIA models, but the differences are smaller than the MNEEs (Fig. 6). The different response is caused by a change in ice configuration. For example, local GIA ($\tau = 4$ kyr) significantly reduces Hudson Strait ice volume before $80$ kyr BP, leading to a smaller but more stable warm-based area (e.g., no rapid changes due to advection of cold ice) and consequentially a continuously active ice stream instead of activation/de-activation cycles (Fig. S41). Therefore, a change in ice stream behaviour can be caused by differences in Hudson Strait basal temperature due to glacial isostatically driven ice volume changes (e.g., decreased insulation due to a thinner ice sheet). The mean period, duration, increase in Hudson Strait ice flux, and mean ice volume change show an increase for experiments with local and without GIA, but most differences are smaller than the MNEEs.

In summary, the GIA experiments with the default GHF indicate that the occurrence of Hudson Strait ice stream surges is sensitive to the overall North American ice configuration. The surge characteristics, however, show only minor changes. In contrast, the experiments with local GIA and without GIA lead to more significant changes for the classic binge-purge surge mode (Fig. S18 and S19). The surge duration decreases, while the number of surges, increase in mid-Hudson Strait ice flux,

and Hudson Strait ice volume change increase. The different model response for high and low GHFs is due to the differences in thermal conditions at the ice sheet base (e.g., Fig. S15).

## 4 Discussion

$Q_1$ *What are the characteristics of Hudson Strait ice stream surges and how do they depend on the geothermal heat flux and basal drag law?*

The surge characteristics are sensitive to the applied GHF and sliding law. Use of either a regularized Coulomb sliding law (Eq. 6) in combination with the default Weertman-type power law (Eq. 1 and 7) or a lower GHF each lead to a similar shift in surge mode (from nearly continuous Hudson Strait ice streaming to long quiescent periods interspersed with short surges) and surge characteristics (more surges, shorter duration, larger increase in mid-Hudson Strait ice flux, Table 3).

For the default GHF input field and Weertman-type power law, the Hudson Strait ice stream is almost continuously active throughout the last glacial cycle. Surges occur after a short quiescent period or as an increase above the background ice flux. Furthermore, the strongest surges (preceded by a complete de-activation of the Hudson Strait ice stream) occur before MIS3. While there is evidence for pre-MIS3 HEs (e.g., Table 6.3 in Bradley, 2014), HEs are usually associated with MIS3. The limited number of strong surges within MIS3 is a consequence of increased basal temperatures (increased insulation due to a thicker ice sheet), leading to a continuously active ice stream (no ice build-up before surge).

### Geothermal heat flux

Decreasing the GHF in Hudson Bay/Hudson Strait below $\sim 37\,\mathrm{mW\,m^{-2}}$ leads to a surge mode consistent with the originally proposed binge-purge mechanism (long quiescent periods interspersed with short surges, MacAyeal, 1993), increasing the Hudson Strait ice flux and ice volume discharge. This shift in surge mode is in agreement with the results 575 of a previous study examining the effect of GHF on HEs (Schannwell et al., 2023). Depending on the parameter vector, the surges can now occur continuously between $\sim 110$ and $\sim 10\,\mathrm{kyr}$ BP (surges before $100\,\mathrm{kyr}$ BP are not considered in the surge characteristics). Critically, the binge-purge mode increases the number of strong surges during MIS3 (compared to the near continuous ice streaming mode), which is more in accord with the timing of Heinrich Layers in sediment cores (e.g., Hemming, 2004).

During a surge, both low and high GHF scenarios lead to a Hudson Strait ice flux and ice volume discharge consistent with literature estimates (Roberts et al., 2014, and references therein). As HEs are identified by IRD layers, the applied GHF must enable the entrainment of enough sediment. However, depending on the mechanism considered, sediment can be entrained for a cold-based (e.g., Meyer et al., 2019) as well as warm-based Hudson Bay/Hudson Strait (Drew, 2023). What ultimately controls the sediment discharge during a surge is the sediment availability. Furthermore, depending on 585 the study and method used, GHF estimates in Hudson Bay and Hudson Strait vary between 20 and $80\,\mathrm{mW\,m^{-2}}$ (e.g.,

Jessop and Judge, 1971; Pollack et al., 1993; Shapiro and Ritzwoller, 2004; Blackwell and Richards, 2004; Levy et al., 2010; Goutorbe et al., 2011; Jaupart et al., 2014; Davies, 2013; Lucazeau, 2019; Cuesta-Valero et al., 2021).

Therefore, both surge modes occur within GHF constraints and are consistent with proposed sediment entrainment mechanisms.

When using the default GHF, the GSM temperatures in the bed are colder or similar to borehole measurements south of Hudson Bay (Fig. S42 and S43). While these boreholes indicate a negative GHF gradient toward the North and older bedrock material tends to be colder (less radioactive decay), no deep boreholes are available in Hudson Bay and Hudson Strait. Therefore, better constraints on the GHF in Hudson Bay and Hudson Strait as well as inclusion of a fully coupled sediment model are required to better determine the most likely surge mode.

**Basal sliding law**

Similar to the effects of a lower GHF in Hudson Strait and Hudson Bay, using a regularized Coulomb sliding law leads to a shift to the binge-purge mode and a large increase in surges, including the number of strong surges during MIS3. While the increase in mid-Hudson Strait ice flux and Hudson Strait ice volume change during a surge are within literature estimates (Roberts et al., 2014, and references therein), the large increase in the number of surges and, therefore, the shorter periods are not (Table 2 and 3). This is in contrast to the lower GHF ensembles, rendering the regularized Coulomb sliding law experiments a less likely representation of the binge-purge mode.

$Q_2$ *Can the sudden reduction of the buttressing effect of ice shelves trigger Hudson Strait ice stream surging?*

In line with both proxy data indicating ice shelf free conditions during most of the last glacial cycle (Hillaire-Marcel et al., 1994; Hesse et al., 1999; De Vernal et al., 2000; Gibb et al., 2014) and other modeling studies (Schannwell et al., 2023), there is no significant Labrador Sea ice shelf in the GSM runs. This conclusion still holds even when running the GSM with a full North American and Greenland ice sheet configuration (Fig. S44, instead of just the stub North-West Greenland shown in Fig. S1).

The relatively small ice shelves in front of the Hudson Strait terminus provide only minor buttressing and are barely affected by the applied ocean temperature forcing. Reducing the ocean temperature by $-2°C$ leads to minor ice shelf growth, as calving remains a restricting factor. Even when completely inhibiting calving in front of Hudson Strait (but using the default ocean forcing), not all parameter vectors yield ice shelves large enough to be buttressed against Greenland.

Ice shelf collapse in runs that have large ice shelves does lead to minor increases in mid-Hudson Strait ice flux. Our overall results indicate that collapse of buttressing ice shelves are likely not the main trigger of Hudson Strait ice stream surging but, depending on the parameter vector, can affect the timing of surges. While changes in whole ice sheet evolution caused by the applied ocean forcing contribute to the differences in surge timing, there is no clear causal relationship between the timing of ocean forcing and the surge cycling response.

This is in contrast to the findings of Alvarez-Solas et al. (2013, Hudson Strait surge induced by collapse of buttressing ice shelf), which use fixed glacial climatic boundary conditions except for changes in the sub-surface ocean temperature. However, the ice shelf in their simulation covers the entire area between Hudson Strait and Greenland, significantly increasing the buttressing and enabling a larger Hudson Strait ice volume (larger surges when removing the buttressing). As discussed above, this scenario is inconsistent with available marine records.

$Q_3$ *Can a sudden breakup of fringing ice shelves along the Canadian coast explain the IRD records (without the need for surges)?*

Based on assumptions about the terrigenous material transported by floating ice during HEs ($100\ \mathrm{km}^3$, Alley and MacAyeal, 1994), the debris concentration in basal glacier ice (5 to 35 %, Lawson et al., 1998), and the accreted ice thickness (1 % of total ice thickness), Hulbe et al. (2004) estimate that a minimum ice shelf volume of $2.8 \cdot 10^4$ to $20 \cdot 10^4\ \mathrm{km}^3$ is required to explain the IRD records by disintegration of fringing ice shelves. The maximum Labrador sea ice shelf volume across all reference runs is $2.5 \cdot 10^4\ \mathrm{km}^3$ (Fig. S44; $12.6 \cdot 10^4\ \mathrm{km}^3$ when calving is completely inhibited in the *ocean forcing area* outlined in Fig. 2). Even when considering fringing ice shelves along the Canadian coast (52.5 to 75.0°N), the maximum ice shelf volume between 100 and 10 kyr BP across all reference runs is only $6.2 \cdot 10^4\ \mathrm{km}^3$ and is below the minimum estimate of Hulbe et al. (2004) for 12 (out of 20) runs. Therefore, it is unlikely that the IRD layers found in the North Atlantic are solely a consequence of ice shelf disintegration. However, this conclusion neglects potential contributions of other source regions with similar geological material, such as the Boothia ice stream (Sanford and Grant, 1998; Hulbe et al., 2004; Naafs et al., 2013). Future work, e.g. ice sheet modeling with a fully coupled sediment model, is required to determine the contributions of individual source regions.

$Q_4$ *How does sub-surface ocean warming affect HEs?*

In the idealized setup of Bassis et al. (2017), grounding line retreat driven by underwater melt and, in turn, modulated by GIA leads to Hudson Strait surges. Although not exactly replicating the experiments of Bassis et al. (2017), we examine the effect of sub-surface ocean warming in a HE context by applying sub-surface ocean forcings of different magnitudes at varying depths. In general, applying a similar sub-surface ocean forcing in the GSM does not significantly affect the surge characteristics and does not trigger new Hudson Strait surges. As for the collapse of buttressing ice shelves, sub-surface ocean forcing can significantly affect the timing of surge cycling (e.g., Fig. S29), but there is no clear relationship between the timing of sub-surface ocean forcing and the surge cycling response. Furthermore, sub-surface ocean forcing also affects whole ice sheet evolution, further complicating the interpretation of the causal relationship.

Depending on the melt coefficients and the ocean temperature (Sec. 2.3), sub-marine melt can reach up to $400\,\mathrm{m\,yr^{-1}}$ in our end-member scenario simulations (specifically designed to increase the melt rate; melt rates in the reference setup are generally below $100\,\mathrm{m\,yr^{-1}}$), indicating that the minor model response concerning surges is not an issue of insufficient sub-shelf melt. The different model response is likely a consequence of the less idealized model setup and the large variety of physical system processes affecting the surges in the GSM that are not present in the modeling of Bassis et al.

(2017, such as their lack of ice thermodynamics). Different implementations of GIA and calving along with different grid resolutions further contribute to the different results.

$Q_5$ *What is the role of GIA in a HE context?*

GIA leads to lower ice sheet surface elevations, increased accumulation, and consequently, larger ice sheets (Fig. 15). The reduction in overall North American and Hudson Strait ice volume when using a local or no GIA model leads to, on average, lower basal temperatures (Fig. S40) and fosters surges in runs that otherwise show a continuous Hudson Strait ice stream.

Analyzing the surge characteristics with the default GHF shows a tendency towards longer and stronger surges for simulations with local or no GIA. The differences are caused by the change in ice configuration but are generally on the same order of magnitude as the MNEEs. Changes in the earth rheology used by the global GIA model have minor impact.

The classic binge-purge surge mechanism in the low GHF experiments is more sensitive to GIA. Due to the different thermal conditions at the ice sheet base, the experiments with local or without GIA now lead to shorter but stronger surges.

Due to its effect on the overall North American ice volume, the consequential change in basal temperatures in Hudson Strait, and the limited range of ice sheet configurations for which Hudson Strait ice stream surges occur, global GIA plays a critical role in modeling ice stream activation/de-activation cycles. These results are especially relevant for interpreting HE modeling experiments that do not use a physically-based GIA scheme.

## 5 Conclusions

Within this study, we investigate Hudson Strait ice stream surges and determine the role of geothermal heat flux (GHF), basal sliding laws, ice shelves, sub-surface ocean temperature forcings, and glacial isostatic adjustment (GIA) in a HE context. The model results are based on the first HE simulations with transient last glacial cycle climate forcing, global visco-elastic glacial isostatic adjustment model, sub-glacial hydrology model, and high-variance sub-ensemble retrieved from North American history matching for the last glacial cycle.

Consistent with proxy records, no large ice shelves develop in the Labrador Sea (unless extreme calving restrictions are applied), leading to minor buttressing effects. Even when completely inhibiting calving in the Labrador Sea, the collapse of large ice shelves leads to only a minor increase in mid-Hudson Strait ice flux. Except for the exact timing of surges, sub-surface ocean warming does not significantly affect the surge characteristics. The occurrence of Hudson Strait surges is sensitive to GIA due to its significant effect on the overall ice sheet configuration and resulting change in basal temperature. Since the ice shelf volume along the Canadian coast is generally smaller than the estimated minimum required for HEs, and since the fringing ice shelves in front of Hudson Strait provide only minor buttressing, our results indicate that even when considering a

combined mechanism of sub-surface ocean warming, breakup of fringing ice shelves, and consequential sudden reduction of the buttressing effect, HEs can not be explained without Hudson Strait ice stream surge cycling.

However, the surge pattern is highly sensitive to the basal sliding law and the GHF in Hudson Bay and Hudson Strait. The GHF estimates for this region vary by a factor of $4$. While better constraints on the GHF are essential to determine the likelihood of the classic binge-purge mechanism compared to a near continuous ice stream with pre-Heinrich Event shutdowns, the increased number of strong surges during MIS3 in the binge-purge mode suggests that the deep GHF for Hudson Bay and Hudson strait is under $35\,\mathrm{mW\,m^{-2}}$. Using a regularized Coulomb sliding law also leads to a shift to the binge-purge mode and increases the number of strong surges during MIS3. However, the overall large increase in surges leads to periods well below those inferred for HEs. In contrast, the period and duration of the reference and low GHF setup are in agreement with literature estimates.

Overall, our experiments indicate that Hudson Strait ice stream surge cycling is the most likely primary Heinrich Event mechanism, but ocean forcings can indirectly affect the timing of surges through a change in ice sheet evolution. The key HE characteristic our experiments have not resolved is HE synchronization with the coldest phases of the Bond cycles (though not always the case, e.g., HE1). While ice shelf collapse and ocean forcing are insufficient to synchronize Hudson Strait ice stream surge cycling in the experiments presented here, a synchronization mechanism without the need for a trigger event during the stadial (Schannwell et al., 2024) could potentially provide the missing link and should be explored in future studies. A further caveat is the usual assumption that IRD flux is approximately proportional to ice flux (be it from Hudson Strait ice stream surge cycling or collapsing ice shelves). A follow-up submission with a fully coupled sediment model will evaluate the correlation between ice and sediment discharge and better link the glacial processes examined here to the IRD layers.

*Code availability.* TEXT

*Data availability.* TEXT

*Code and data availability.* Data is available upon request from the corresponding author. An archive of a representative sample of simulation output (reference setup) is available online at https://doi.org/10.5281/zenodo.13840063 (Hank, 2024). A GSM description has been submitted to GMD and is already available as a preprint on Lev Tarasov's web page (Tarasov et al., 2024).

*Sample availability.* TEXT

*Video supplement.* Supplementary videos are available online at: https://doi.org/10.5281/zenodo.10214928 (Hank, 2023).

*Author contributions.* KH prepared the experimental design, ran the model, and analyzed the results. Both authors contributed to the interpretation of the results and writing of the paper. LT provided the source parameter vector ensemble.

*Competing interests.* The authors have no competing interests.

*Disclaimer.* TEXT

*Acknowledgements.* The authors thank Anne de Vernal and Claude Hillaire Marcel for their guidance on the interpretation of marine proxy records in a Heinrich Event context. We also thank Clemens Schannwell, Jorge Alvarez-Solas, an anonymous reviewer, and the handling topic editor Pepijn Bakker for their constructive comments.

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
