# Peer review of "The comparative role of physical system processes in Hudson Strait ice stream cycling: a comprehensive model-based test of Heinrich event hypotheses"

_EGUsphere, 2024_

## Referee Comment (RC1)

Review of
*"The comparative role of physical system processes in Hudson Strait ice stream cycling: a comprehensive model-based test of Heinrich event hypotheses"*
by Kevin Hank and Lev Tarasov

Kevin Hank and Lev Tarasov present a comprehensive study on the potential causes of Heinrich events. Their aim is to investigate several hypotheses which have been proposed in the literature, mainly: the effect of geothermal heat flux and basal drag, buttressing effect, subsurface melt and GIA. For this, they use the glacial system model (GSM) and for a set of experiments (20) they conduct a test of sensitivity experiments.

While I believe the manuscript holds significance and aligns with the focus of Climate of the Past, I suggest some restructuring prior to publication. While the introduction effectively outlines the study's goals, the subsequent sections lack cohesion, making it challenging for readers to follow. There are too many figures which are referenced several times without following a chronological order necessarily which makes it easy to lose the guiding thread. Additionally, the interpretation of the experiments may be difficult to follow.

My biggest concern is related to the investigated ensemble parameters. I think the choice of parameter space feels a bit arbitrary and unnecessarily extensive for the paper's aims. I want to emphasize that I recognize the value of this manuscript but stress the importance of enhancing its readability for a broader audience. Below you can find my main concerns.

**General comments:**

**Main manuscript:**

**Parameter choice:**
I do not understand how you choose your 20 parameter vectors. First you run ~15000 simulations, from which you consider ~200 simulations as realistic based on your sieves. Then you redo these 200 simulations at higher spatial resolution and apply new sieves based on IRD layers. From those you hand pick 20 simulations. Have I understood it correctly?

I am a bit skeptical about some of your parameters. A lot of your parameters are related to climatologies (for example: global LGM temperature scale factor, desert-elevation exponent, temporal Empirical Orthogonal Function weight 1), but you are not assessing the role of different climatologies in your study. Since this is not the focus of your manuscript I think you need to apply the same climatologies to all of your experiments. If not, the occurrence of surges in your experiment could be caused by different climatologies rather than ice

dynamics or other forcings. This would be also very beneficial for the readers since you introduce a lot of parameters (52 in Table 1) which you do not explain and are very technical.

I get the feeling that many of your selected parameters are the same as those of Tarasov et al. in preparation describing the model GSM. Though investigating such a large parameter space makes sense for a description paper, I do not think it is intended for this manuscript.

**Reference state**
This point goes a bit in line with my previous comment. Many of your plots only show one vector parameter but it's not always the same vector parameter. To me this feels confusing, I would prefer to have a reference vector over which you change conditions rather than different states. If you are trying to assess the effect of different boundary conditions, friction laws or oceanic forcings it would be more useful to have one reference state over which you change conditions.

**Hydrology model:**
Based on Figure 4 and Figure 7 it seems that basal melt below grounded ice plays a major role in your surges. You need to explain how you compute basal melt for grounded ice points and your hydrology model. Since it is a local hydrology model and you state that you saturate your water thickness at 10 meters i assume that you do not conserve mass, right? How does your water thickness affect your surges? I guess it will play a major role in your surges and I think it is necessary to investigate that parameter.

**Additional**
What does your LGM ice sheet look like? How does it compare to other studies? I am also missing your forcing index. You could add it in your time series plots.

**Supplementary Material:**

41 figures in the Supplementary Material is way too much and makes it difficult to follow the paper. These figures are referenced too many times and cuts the flow for the reader. Please consider reducing the amount of figures with those which are strictly necessary for your article. For example, you could merge figures S2-S6.

Table S1 is complicated to understand for the reader. First, I think you investigate too many parameters, but if the authors decide to follow this approach, then I would suggest splitting it in different sections, such as ice dynamics, climatologies, GIA, GHF (you could add a horizontal line). In addition, you define parameter names which you do not use in your manuscript, I do not think this is necessary.

**Technical comments:**

**Section 2.3**

Please describe the equation of your Weertman-type power law. You state that your parameter C_warm depends on other parameters such as C_rmu and C_fslid but you do not give further detail. Either you remove that sentence or you explain how C_warm depends on those parameters.

Do you apply any basal-stress scaling at the grounding line? If you are using a coarse resolution, scaling basal stress at the grounding line has shown to help to simulate grounding line migration in agreement with high resolution. Actually, this could help to simulate more surges potentially. Do you apply any melt at the grounding line?

**Section 2.6**

I do not understand your MNEE method and this sentence is confusing to me, please rephrase to make it clearer.

*"As in Hank et al. (2023), the final MNEEs (shaded grey regions in Fig. 6) are the maximum percentage difference for the metric in question of a setup with stricter numerical convergence and a setup with stricter numerical convergence with increased maximum iterations for the outer Picard loop (from 2 to 3, solving for the ice thickness) and the non-linear elliptic SSA (Shallow-Shelf Approximation) equation (from 2 to 4, solving for horizontal ice velocities). These MNEEs are then used as a threshold to determine if a change in model configuration leads to numerically significant differences in the surge characteristics. Differences smaller than the MNEEs should be interpreted as model response not resolvable given the numerical sensitivities."*

**Section 3.3**

I am surprised that using a (regularized) Coulomb law leads to model crashes and a model run time much slower than a Weertman law. Do you have an explanation for this?

**Section 3.6**

I would try to avoid using acronyms as section titles.

**Figures**

As a general comment for your figures I would suggest indexing. Figure 4 for instance has 6 subplots but when you refer to it I do not know which panel I should be looking at.

**Figure 4**

- Please use another naming for your "surges" since notation S9 can be confused between you Figure S9 or your surge S9.
- How is it possible that you obtain the highest buttressing value when your ice shelf is small?

**Figure 7, 8, 13, 15**
- Put the legend in one plot, you do not need to put it in every plot.

**Figure 9**
- You do not reference Figure 9 in your manuscript. Do you need that figure?

**Figure 10**
- You do not need Figure 10 since you are already showing your reference state in Figure 12.

---

## Referee Comment (RC2)

**Review of Hank & Tarasov "The comparative role of physical system processes in Hudson Strait ice stream cycling: a comprehensive model-based test of Heinrich event hypotheses"**

May 22, 2024

**General comments:**

The manuscript by Hank & Tarasov presents a suite of simulations that aims to investigate the sensitivity of the Glacial Systems Model (GSM) to processes that have previously been shown to have an influence on Heinrich Event characteristics in other models. The novelty in comparison to previous studies is that the authors use transient forcing and present an ensemble that covers a much larger parameter space. Based on this, they confirm earlier findings from e.g. Mann et al. [2021], Schannwell et al. [2023] that GSM also exhibits two different states - a streaming and cycling surging state. Moreover, they show that in their model, geothermal heat is a major controlling factor while ice shelves and their buttressing matter very little.

Overall, the topic of the paper is interesting and I believe that the science behind it is sound. However, in its present form the reader is drowned in the sheer amount of simulations that are presented in a rather unstructured fashion. The lack of structure in the paper and particularly in the methods sections leaves the reader confused as to what exactly the authors did and makes it challenging to find a common thread through the manuscript as well as evaluate the scientific results. Therefore, I do not think the paper is ready for publication in its current form and a substantial restructuring and part-rewrite is required. I recommend the authors take into account my comments listed below. I hope the authors find my comments helpful.

**Specific comments:**

1. The entire methods sections felt really incoherent almost as if sections were being added as new ideas for sensitivity simulations were designed. By the time I got to the results,

there were so many simulations and sub-ensembles that I was utterly confused what was done in the end and what the different simulations were actually referring to. The absence of an experiment table or a flowchart of how the different sub-ensembles were generated just confounded my confusion. My suggestion is to remove section 2.1. "Modelling approach" and really start with the model description (your current section 2.3). In there, I would like to see what type of model GSM is! I believe it is an ice-sheet model but this is never really spelled out. This section should also incorporate all of the coupling that is included e.g. subglacial hydrology etc. as well as timesteps, resolution and the like. Then, I would follow that by all the boundary conditions e.g. your current sections 2.4 to 2.5.2. Then move on to the section that you call "Ensemble parameter vector" which I suggest to rename to "Creation of sub-ensembles" as I believe it makes it more accessible. In this subsection, you absolutely need a table with the different sub-ensembles as well as a flowchart illustrating how each of these sub-ensembles was generated, e.g Baseline ensemble $->$ sieve $1->$ sieve 2 etc. Also, I find the section title "Bounding experiments" somewhat confusing. Do you mean endmember scenarios? If so, why not name it like that?

2. In your results section you often speak about specific parameter vectors, for example in the caption of Figure 4 you mention "parameter vector 1". It is unclear to me to what sub-ensemble this refers to as parameter vector 1 is never really defined in the manuscript. So, I strongly recommend to define these parameter vectors somewhere in the methods section (e.g. in a table), so that the reader can go back and check what this refers to.

3. This issue was already raised by the other reviewer, but I believe that the effect of the transient climate forcing certainly warrants a discussion. First of all, it is never mentioned what climate forcing is used to drive the model. This goes back to my previous point that I am not sure whether GSM is an ice-sheet model or a coupled climate-ice sheet model. In any case, it remains unclear to me whether the differences you report here are due to your parameter changes or due to the changing transient forcing and it seems quite likely that there is at least some signal from the transient climate forcing in your results. This ought to be acknowledged and discussed.

4. On a related note, is it correct that your reference setup (e.g. Fig 7) does not show any Heinrich events, but simply a steady ice stream? I believe it is never mentioned, but your algorithm does not detect any surges in that time series right? But then it is unclear to me how you calculate the period and number of surges presented in Table 1.

5. You introduce the term "Minimum Numerical Error Estimate (MNEE)" which at least to me is a new concept. I think it should be clearly stated whether this is a common concept from the field of numerical analysis or something that you have introduced

(I see that you used a the same approach in your previous paper). If you introduced it, it would be helpful to briefly(!!) motivate why this is a useful quantity. Because from what you mention in between lines 274 – 279, this seems a rather arbitrary choice (increasing your Picard iteration by one). I am also confused that your only parameter to measure MNEE is the "number of iterations". At the very least, I was expecting a combination of "number of iterations" and "residuals". The reason for this is that I expect the residual to be higher after 4 iterations if you are in a rapidly changing state than after 2 iterations in a relatively stable state. In my view, the way you have defined it is inconsistent because it does not tell anything about whether your solver has actually converged or not. For example, you can increase your number of iterations to 50, but that does not say anything about the quality of your solution. Also since GSM is run in serial, why not compare it to the solution from a direct solver like UMFPACK?

6. At the end of the introduction you list five research questions that your paper aims to answer. This is admittedly a bit of a subjective matter, but my impression was that these questions are very generic and could be summarised with something like are our Hudson surges sensitive to perturbations in geothermal heatflux, GIA, ocean melting, etc. While certainly worth exploring, to me, this type of question layout is more suited to a thesis format but not necessarily for a paper. I also could not help, but get the impression that you do a bit of everything which results in a lack of focus what you are really trying do address here. For example, I do not think that your paper actually addresses Q3 as you solely focus on the Hudson ice stream. As an add-on, I am also in favour of a short paper roadmap at the end of the introduction especially when it is as complicated and long as this paper.

**Technical corrections:**

Abstract:
L2: I think instead of using the term Glacial Systems Model, it would be better to refer to the type of model like coupled ice sheet-subglacial hydrology model. Of course, if you'd like to keep GSM in there, you could combine these two.

L20: Add a reference to the original Heinrich paper?

L35: You could add for completion our atmospherically driven mechanism recently published in Schannwell et al. [2024].

L38: Not to be too picky, but I would argue that we did test the sensitivity of mPISM to the geothermal heatflux in our Schannwell et al. [2023] paper

L85-86: All models are parallelisable, but GSM in its current incarnation is not parallelised.

L89: What is the physical mechanism for the choice to allow sliding at sub-freezing temperatures?

L93: Be precise here! What components are included in the current setup? Also asynchronous coupling mean different things to different communities. Does that mean you run your GIA model accelerated? Moreover, for your GIA model what kind of ice thickness distribution do you prescribe in the southern hemisphere or the other northern hemispheric ice sheets? This confusion originates from the fact that you never specify what your modelling domain is. In addition, it would be helpful to provide more detail about the GIA model, because out of the blue in the results section, you start talking about local and non-local GIA.

L110: It took me quite a while to realise that what you refer to as mid-Hudson Strait ice flux is a flux gate that is some distance upstream from the grounding line. I think it would be much clearer if you spell this out explicitly somewhere.

L116: Maybe worth defining that # stands for number of surges

L121: What is that resolution in kms for the Hudson ice stream. How does this coarse resolution affect your ability to model surging. We saw quite dramatic changes when we increased resolution from 50 km down to 20 km some years ago.

L133: Delete second "with"

L167: From your description it is unclear whether or not you do "Schoofing", meaning whether you prescribe the Schoof flux as an internal boundary condition at the grounding line. Please clarify.

L164 and L205: Here and throughout, I am a supporter of self-contained papers and hence not a big fan of excessive referring to papers that are in preparation and may or may not get published in the future. But I leave this to the editor to decide what CPs policy concerning this is.

L194: $d_{OF}$ has not been defined yet.

L197: What do you do if you do not have any ocean points under your floating ice shelf? Do such situations arise?

L199: Do you mean you tune your parameters for present-day ice sheets, because there is no Laurentide today.

L223: This makes it sound as if GSM has an ocean component coupled to it, where in fact you are using ocean temps. from a GCM simulations, right? If so, please rephrase.

L230: Even for me as an ice-sheet modeller, this is getting quite hard to follow here. I am not sure, but is your ice shelf removal simply a very high basal melt rate that melts your ice shelf away? If so, what is the time it takes for the ice shelf to disappear and how might this potentially rather gradual removal affect the response in comparison to a sudden removal of the ice shelf?

L310: As mentioned in my main points above, I simply do not know what parameter vector 1 refers to? Is this a single simulation, a composite, or a sub-ensemble?

L344: Again an explanation, definition what your refer to as parameter vector would be highly appreciated.

Table 1: Again it is unclear to me what the reference setup really is?

L349: Again, this is pretty much what we showed in Schannwell et al. [2023].

L365: What does crashing mean here? Solvers did not converge anymore?

L366: I think, this certainly needs some discussion why you think the switch from Weertman to Coulomb makes such a big difference in the run time.

L392: In the interest of shortening the paper, consider removing everything regarding the regularised Coulomb and simply state that run times were too long to make these runs feasible.

L457: Are you sure your mean "increase" here? It is possible, but I would argue an elevation decrease is more often than not associated with an accumulation decrease.

L569: What are pseudo-Hudson Strait surges? Bassis et al. use an idealised setup that is based on the geometry of the Hudson Strait. Does GSM have the marine-ice-cliff instability mechanism implemented? This is an integral part of the Bassis et al. mechanism and renders the comparison pretty far-fetched if it doesn't.

L574: I find these melt-rates unreasonably high. I mean maximum present-day melt rates are around 100 m/yr and you have four times the melt during the glacial? That seems very hard to believe.

L606: I am pretty sure that you did not show synchronisation of your HEs with the Dansgaard-Oeschger cycles.

L614–616: Are you sure you want to end with something like this?

**Figures:**

The Figures are overall of good quality. What I am missing is a Figure of the modelling domain. If it is a global setup, this is not needed, but then this needs to be stated clearly in the text.

Fig. 2: The contours are very hard to see in the left but primarily in the right panel.

Fig. 4: and throughout. I am not a big fan of pythons default option to have the scientific notation on top of each subplot in pretty small font. For a second I thought that your flux was as high as 2 Sv. You could try using mSv instead or work that exponent into your axis titles.

Fig. 4 caption: Again, it is unclear what parameter vector 1 is. I think one way of helping the reader other than a clear definition is a more intuitive name such as GIA sub-ensemble.

Fig. 5: Consider removing repetitive colourbars and axis labels for the benefit of larger panels.

Fig. 7: Judging from this, your reference simulations has no Heinrich events? But why is there an ice volume increase at the same time as the ice flux increases? What is the origin of this?

Fig. 9: Somewhere you should mention what type of kernel you use and what your bandwidth is and how/why you chose it.

Supplementary Figures: I can also only reiterate what the other reviewer mentioned, 40 Figures in the supplement is a lot and you should really make sure that each of these Figures serves a purpose. For example, what does the Figure S8 add? In the text it is referenced to show a more gradual increase in ice flux, but I have no idea how a snapshot of the velocity field could convey such a message.

Sincerely, Clemens Schannwell

**References**

L. E. Mann, A. A. Robel, and C. R. Meyer. Synchronization of heinrich and dansgaard-oeschger events through ice-ocean interactions. *Paleoceanography and Paleoclimatology*, 36(11), Oct. 2021. ISSN 2572-4525. doi: 10.1029/2021pa004334. URL http://dx.doi.org/10.1029/2021PA004334.

C. Schannwell, U. Mikolajewicz, F. Ziemen, and M.-L. Kapsch. Sensitivity of heinrich-type ice-sheet surge characteristics to boundary forcing perturbations. *Climate of the Past*, 19(1):179–198, Jan. 2023. ISSN 1814-9332. doi: 10.5194/cp-19-179-2023. URL http://dx.doi.org/10.5194/cp-19-179-2023.

C. Schannwell, U. Mikolajewicz, M.-L. Kapsch, and F. Ziemen. A mechanism for reconciling the synchronisation of heinrich events and dansgaard-oeschger cycles. *Nature Communications*, 15(1), Apr. 2024. ISSN 2041-1723. doi: 10.1038/s41467-024-47141-7. URL http://dx.doi.org/10.1038/s41467-024-47141-7.

---

## Referee Comment (RC3)

In this article, Hank and Tarasov, perform several simulations of the Laurentide ice sheet of the last glacial period in order to elucidate the most likely candidates to explain the origin of Heinrich events. The subject and the way it is addressed in the introduction and discussion are of very high relevance. The paper is potentially very well suited for publication in Climate of the Past. I see, however, two major deficiencies of the current manuscript: A questionable and irreproducible experimental setup and the lack of adequate conclusions. I expand these aspects in the following.

**General comments regarding the experimental setup and the exploration of the parameter space:**

Your experimental setup is firstly very hard to understand and secondly not reproducible at all. It is hardly understandable because your entire exploration of the parameter space relies on perturbing the values of the parameters of your table S1. However the majority of these parameters are not explained at all in the manuscript.

The very pertinent questions you are addressing in this article are conveniently described at the end of the Introduction paragraph. They summarize the different mechanisms triggering Hudson Strait ice surges in the context of HEs. However, it is difficult to understand how exploring, for example, the values of the "northwestern desert-elevation cutoff" is relevant in this context. The equations on which these not-defined parameters apply are not present in the manuscript, so the reader can not even guess what influence they have on the scientific questions you are addressing. This is the case because the exploration of these parameters only helps to see whether a given realization of the model has fulfilled or not your sieves, but does not give any mechanistic picture of their influence on ice stream behavior. Nor these experiments can be reproduced with other models of similar physics. An article should be as shelf-contained as possible, so if you analyze an ensemble of simulations that have fulfilled a given sieve, the reader can understand why that is the case and the implications for your conclusions without having to open the experimental setup of another manuscript or guess that it is going to be more deeply explained in another article in preparation.

In my opinion, both these two problems would be solved by the following strategy: You could fix several of the parameters that have already allowed you to fulfill the sieves and that are mechanistically irrelevant for the ice stream behavior to a single documented value. The equations on which these parameters appear could be summarized in an appendix. Then, you could systematically explore the parameter space of the processes that are suspected to be very influential on the ice-stream dynamics (e.g. sliding laws, effective pressure, dependence on bed type and base thermal state…; see below). And finally you could describe how these different values affect the nature of the simulated Hudson Strait ice stream variability.

**Specific comments regarding the experimental setup and the exploration of the parameter space:**

What are the values of C_warm and C_froz? (absent in table S1)

What is the value of n_b for hard bedrock? (I could guess it is a 3 from a dimensional analysis based on the units of table S1, but it is not present in that table)

Have you explored different values of n_b, hard? I assume not (from what is shown in table S1). If not, why not simply use the same value as for n_b, soft? That will reduce one degree of freedom.

According to equations (2) and (3) your sliding coefficient, C_b is a function of the bedrock type (soft sediments vs hard), the thermal character of the base and the effective pressure. Why do you need additionally to change the exponent of the sliding law over soft and hard bedrocks?

Please justify the 3.5 exponent in equation (4)

I deduce from table S1 that your maximum allowed value for h_wb,Crit is 1.0. Why limit it to that relatively low value? What happens if you significantly increase it?

Additionally, what does the model do if h_wb reaches h_wb,Crit?
Neff could not be reduced further (it would be purely 0!! From equation (4)), but what happens with the excess of heat? Can you create additional basal water? If not, could you diagnose the "free" values of h_wb?
How sensitive are the surging cycles to this basal water limit?

You introduce a Coulomb dragging law in equations (5a) and (5b).
How is C_c variable? In space or in time? According to what? (not deducible from the manuscript or table S1)

The necessity of equation (6) is justified in the following manner: " To account for possible Weertman-type sliding when Coulomb drag is high …"
A high Coulomb drag will potentially stabilize the flow and prevent the appearance of a limit cycle. Why convoluting a Weertman-type sliding law and a Coulomb one?
What is the effect of using the "pure" Coulomb law (without limiting tau_b ad hoc) on the surges?

GHF reconstructions:

Blackwell and Richards, 2004 is just a map without any reference to a published peer-reviewed work. Shapiro and Ritzwoller, 2004 show values around 55 mW/m² with a standard deviation around 20 mW/m².  The minimum value for the whole region in Pollack 1993 is higher than 45 mW/m² and the Hudson Bay and Strait regions would show a mean value closer to 60 mW/m². Goutorbe et al., 2011 showed a Hudson Bay around 40-45 mW/m² and higher values in Hudson Strait for their first method, and a regional mean around 50 mW/m² for the same region with some hot spots of more than 70 mW/m². Lucazeau, 2019 reconstructed values in Hudson Bay

are around 50 mW/m² in their first method and around 40 mW/m² in the other two methods, while they show significantly higher values (around 60 mW/m²) for the three methods in the Hudson Strait Area. Finally, as far as I could see, Cuesta-Valero et al., 2021 do not show any reconstruction of geothermal heat flow nor seems the intention of their paper.

Line 345 reads: "The exact transition point [to the binge-purge mode] depends on the parameter vector in question but generally requires a Hudson Strait/Hudson Bay GHF ave ≤ 37 mW/m². And line 357 states: "Therefore both types of Hudson Strait ice stream surge cycling are consistent within available GHF constraints."

In light of the reconstructed values described in your referenced studies, this last sentence seems highly inaccurate or simply wrong. It should rather say something like: "The binge-purge mode is, under our experimental setup, only accessible if GHF<37 mW/m² which represents the lower bound of available constraints".

**Conclusions of the manuscript in light of the experimental setup**

In the following I will enumerate a number of questions and concerns that, in my opinion, make some of the current conclusions highly questionable.

What is the influence of h_wb,Crit on the transition to binge-purge?
Would this transition be possible if basal water is conserved?

Your basal friction laws are designed in a way that effective pressure can be reduced several orders of magnitude (even to purely 0!) when there's enough basal water.
Also, Cb will accordingly reach extremely high values (equation (3)) when effective pressure drops. Do you think this is glaciologically realistic (as opposed to Leguy's parameterisation, for example, or even the PISM treatment?

The regularized Coulomb law is thought to perform well, even capturing the heterogeneous character of the bed without needing to guess different friction coefficients in soft and hard beds (Joughin et al., 2019).  In section 3.3 it is stated that the majority of the Coulomb runs crashed. And that it significantly increased the needed computational time for those who survived.
Why is this? Could it be because of equation (6)? Or because of the additional complexity of the parameterisations over the friction coefficient and effective pressure introduced in equations (2), (3) and 4? Again, what would be the effect of letting the Coulomb law do its job, without limiting tau_b?

Runs that are subjected to severe numeric problems can often illustrate that the physical problem is not well-posed. Can you discard this is what is happening here?
How many of your reference ensemble runs crashed?

Looking at figure 5, the reader might notice that the upper limit of the basal velocities bar is 40,000 m/yr. It is not known whether the simulated velocities ever approach that value, because the manuscript does not contain the time series of velocities over a whole ice stream cycle, but I imagine that they are not far from the upper limit of the depicted scale. From figure 5 it is clear that they reach up to 20,000 m/yr for several hundred of kilometers upstream of the grounding line. Current observed surface velocities in Antarctica do not go beyond ~1,500 m/yr downstream of the grounding line. Do you think your simulated velocities are realistic?

Illustrating the dependence of your results to different GHF values seems completely adequate. However, as described above, spontaneous cycling ice stream behavior seems to appear only (given aside other questionable choices of your basal sliding laws) for the lower limit of available constraints.

All in all, unless the authors show otherwise, their experimental setup is constructed in a way that the relevant-for-the-problem parameter space (and thus the phase space of the associated physical problem) is very narrow and situated in a very specific region.

In the absence of a rigorous illustration of the influence of the particular assumptions of the experimental setup (basal sliding laws, basal water, effective pressure, dependence of the friction coefficient on the nature and temperature of the bed, numerical issues…) on the mechanisms favoring spontaneous ice stream cycling, the conclusions reached here are based on the analysis of a very particular experimental setup. (Which in my opinion is, by construction, prompt to oscillate in a questionable physical manner).

And this leads me to my next point:
What happens if you run the Antarctic ice sheet under such a configuration?

To answer this question, you would need to make some assumptions concerning the bed type. One reasonable choice would be to assume the presence of soft sediments in every Antarctic marine sector. Do you expect that such a simulation would give reasonable results?

In my modeling experience, this is unlikely.

GRISLI and Yelmo can also show spontaneous ice stream oscillations under extreme conditions (e.g, limiting basal water and having very distinct spatial basal frictions). But when you apply such a physics to Antarctica, it becomes very difficult to approach observed velocity values, and furthermore some ice streams become very noisy and others dramatically oscillate in a manner that so far is not observed in Antarctica.

Therefore, I believe the current experimental setup extremely conditions the current conclusions of the paper.

The introduction section nicely ends with the presentation of very relevant questions regarding the different mechanisms exciting the Hudson Strait ice stream variability during HEs. The

discussion section also nicely re-addresses those questions (see some minor comments below) in the context of the new results shown in the manuscript. The conclusions, however, do not fairly summarize the findings and are biased towards the spontaneous ice-stream cycling mechanism.

After everything that is shown in the paper, and given all the limitations of the experimental setup pointed out in this review, concluding that "Based on our results, Hudson Strait ice stream surge cycling is the most likely Heinrich Event mechanism…" seems unjustified.

Under a questionable and very particular experimental setup, your simulations show that you have to go to the very low bound of Geothermal heat forcing for the ice stream surge cycling to emerge.  But, even assuming that a binge-purge like oscillation is a good candidate to explain HEs, there is a remaining puzzling question that you completely ignored. Why are HEs happening at the middle of the cold NH phases, or stadials?
Some synchronization mechanisms have been explored in the literature to potentially answer this question (.e.g. the work of Calov and Ganopolski and Shanwell et al., 2024) but you do not inform whether these synchronization mechanisms are captured in your simulations. So, under your experimental setup, the question of HEs occurrence during the surface NH cold phases is still of concern for your conclusions (see also Barker et al., 2015; Nature).
Therefore, the authors seem to be evaluating certain hypotheses with a level of criticism and rigor not present in the case of their announced more likely mechanism.

You discard the ice-shelf breakup related hypothesis (Q2) because your simulated ice shelves are not big enough to significantly buttress the Hudson strait ice stream and because the paleoreconstructions of the Labrador Sea conditions during MIS3 do not seem compatible wit the existence of a big ice shelf in the Area.
This is a fair criticism, but I would like to point out here that in GRISLI and Yelmo (both codes are available; the first upon request to the former developer, Catherine Ritz or myself, and the second here: https://github.com/palma-ice/yelmo) the emergence of a big Labrador Sea ice shelf is a pretty natural characteristic provided oceanic temperatures are low enough. Even with a relatively warm ocean, Yelmo simulates the existence of very developed fringing ice shelves around the mouth of Hudson Strait (Moreno_parada et al., 2023; The Cryosphere).

The questions Q2, Q3 and Q4 of your discussion are compatible with each other. When answering Q3 you state that your maximum simulated ice shelf (leaving aside the one with inhibited calving) is close to the minimum required by Hulbe 2004 to explain the IRD signal. So, can you discard that a combination of Q2, Q3 and Q4 produces both the icebergs from breaking the ice shelves and the increase in the flux at the grounding line necessary for explaining HEs?

I believe that referencing our paper (Alvarez-Solas et al., 2013) in lines 95 and 548 in a technical and discussing context respectively, without having it cited in the introduction, when the different hypotheses are explained, is academically incorrect. Note, we also explored the conceptual idea of triggering HEs through changes in the oceanic temperature (Alvarez-Solas et

al., 2010; NatGeo), and the effects of an ice-shelf breakup during H1 (Alvarez-Solas et al, 2011, ClimPast), coetaneous with Marcott et al., 2011

Jorge Álvarez-Solas

---

## Author Comment (AC1)

**Author's response to Anonymous Referee 1 Comment 1**

**May 20, 2024**

[Kevin Hank and Lev Tarasov present a comprehensive study on the potential causes of Heinrich events. Their aim is to investigate several hypotheses which have been proposed in the literature, mainly: the effect of geothermal heat flux and basal drag, buttressing effect, subsurface melt and GIA. For this, they use the glacial system model (GSM) and for a set of experiments (20) they conduct a test of sensitivity experiments]

[While I believe the manuscript holds significance and aligns with the focus of Climate of the Past, I suggest some restructuring prior to publication. While the introduction effectively outlines the study's goals, the subsequent sections lack cohesion, making it challenging for readers to follow. There are too many figures which are referenced several times without following a chronological order necessarily which makes it easy to lose the guiding thread. Additionally, the interpretation of the experiments may be difficult to follow.]

[My biggest concern is related to the investigated ensemble parameters. I think the choice of parameter space feels a bit arbitrary and unnecessarily extensive for the paper's aims. I want to emphasize that I recognize the value of this manuscript but stress the importance of enhancing its readability for a broader audience. Below you can find my main concerns.]

We thank the referee for their constructive comments. A point-by-point reply is reported below, with referee comments in orange and our replies in black. We agree with the specific referee comments not listed here and will revise the manuscript accordingly.

To clarify, we use the GSM with a set of 20 parameter vectors (not experiments) for the base ensemble and conduct an ensemble-based sensitivity analysis in more than 40 experiments. As such, each experiment is carried out with a 20-member ensemble.

As suggested by the referee, we will reduce the amount of figures by merging plots with similar content. Where possible, the figures will be rearranged to follow a chronological reference pattern. However, breaking this pattern will be required in some instances (e.g., when highlighting details of figures in the methods section (e.g., line 273)).

The choice of ensemble parameters and parameter ranges within this study is based on extensive model development and testing [Tarasov and Peltier, 1999, 2007, Pollard and DeConto, 2012, Drew and Tarasov, 2023, Hank et al., 2023, Tarasov et al., 2024, in preparation]. While the parameter space is more extensive than in previous Heinrich Event modeling studies, the authors argue that it is still insufficient to fully capture the uncertainties involved. However, computational expenses restrict our ability to further increase the parameter space.

**1 General Comment**

**1.1 Main manuscript**

[Parameter choice - I do not understand how you choose your 20 parameter vectors. First you run  15000 simulations, from which you consider  200 simulations as realistic based on your sieves. Then you redo these 200 simulations at higher spatial resolution and apply new sieves based on IRD layers. From those you hand pick 20 simulations. Have I understood it correctly?] That is generally correct. However, we refer to the 200 simulations as *Not Ruled Out*

*Yet (NROY)* rather than *realistic.* The latter is imprecise, and the former better communicates the contingent nature of our inferences about past earth system evolution, as well as a reminder that explicit criteria are being used for this designation. Furthermore, as stated in the text, the 20 runs were hand-picked to provide two high-variance (in parameter values and run metrics) subgroups: one subgroup of 10 for which all runs have more than 2 surges (at the higher resolution), and one for which this is not the case for the higher resolution run but is the case for the corresponding lower resolution run (i.e., with the same parameter vector).

[Parameter choice - I am a bit skeptical about some of your parameters. A lot of your parameters are related to climatologies (for example: global LGM temperature scale factor, desert-elevation exponent, temporal Empirical Orthogonal Function weight 1), but you are not assessing the role of different climatologies in your study. Since this is not the focus of your manuscript I think you need to apply the same climatologies to all of your experiments. If not, the occurrence of surges in your experiment could be caused by different climatologies rather than ice dynamics or other forcings. This would be also very beneficial for the readers since you introduce a lot of parameters (52 in Table 1) which you do not explain and are very technical.] Relying on only one climatology neglects the uncertainties associated with the climatic forcings during the last glacial cycle. Otherwise, how does one justify arbitrarily chosen climate forcing given the uncertainty in glacial cycle climate? And how does one know if their results have any validity for a different climate history? To (albeit partly) account for these uncertainties, our analysis is based on ensemble results. The key idea behind this approach is to identify physical processes that show a significant effect across simulations with different ensemble parameters. This both increases confidence in our results and reduces the possibility that an observed modeling response is only due to, e.g., the chosen climate forcing.

[Parameter choice - I get the feeling that many of your selected parameters are the same as those of Tarasov et al. in preparation describing the model GSM. Though investigating such a large parameter space makes sense for a description paper, I do not think it is intended for this manuscript.] The reviewer does not seem to appreciate that the ensemble parameters in the GSM are effectively in all paleo ice sheet models not coupled to an EMIC more advanced than the EBM (energy balance climate model) in the GSM. Most models implicitly set most of the parameters in the model to a fixed value, but that does not make the uncertainty associated with that parameter go away. For those ISMs that are coupled to a moderate to advanced complexity EMIC, there will be plenty of new EMIC parameters to replace the climate-forcing parameters in the GSM.

While investigating the parameter space is not the main aim of this paper, Table S1 intends to show the parameter ranges used within this study.

[Reference state - This point goes a bit in line with my previous comment. Many of your plots only show one vector parameter but it's not always the same vector parameter. To me this feels confusing, I would prefer to have a reference vector over which you change conditions rather than different states. If you are trying to assess the effect of different boundary conditions, friction laws or oceanic forcings it would be more useful to have one reference state over which you change conditions.] While we agree that using the same parameter vector might be less confusing, it is not straightforward to determine one base vector. Different parameter vectors respond differently to changes in model configuration. Generally, we plot the parameter vector that best resembles the ensemble response for a specific change in model configuration. However, for every time series (single vector) plot, we also show an ensemble mean plot. The single vector plots are intended to give a physically self-consistent example (which, e.g., an ensemble mean does not provide). On the other hand, the ensemble means and standard deviations provide more robust statistical results for the ensemble response to the change in model configuration.

[Hydrology model - Based on Figure 4 and Figure 7 it seems that basal melt below grounded ice plays a major role in your surges. You need to explain how you compute basal melt for grounded ice points and your hydrology model. Since it is a local hydrology model and you state that you saturate your water thickness at 10 meters i assume that you do not conserve

mass, right? How does your water thickness affect your surges? I guess it will play a major role in your surges and I think it is necessary to investigate that parameter.] The revised draft will include a brief description of the basal melt rate (which is simply from applying conservation of energy to the ice sheet). However, we are confused by the reviewer's question. The basal meltwater from the ice sheet is fed directly into the basal hydrology model, and we cannot envision what else could be done. Furthermore, the GSM is configured as all ice sheet models not fully coupled to a dynamical ocean model: all meltwater that is not refrozen leaves the ice sheet, and therefore mass is not conserved. The GSM internally does conserve mass: integrated: accumulation − melt + refreezing − calving = change in ice mass.

[Additional - What does your LGM ice sheet look like? How does it compare to other studies? I am also missing your forcing index. You could add it in your time series plots.] The LGM ice sheet configuration differs between parameter vectors. A plot showing the ice volume of all parameter vectors when run in the reference configuration, along with MIS 3 and LGM example timeslice map plots, will be added to the revised draft. However, a detailed comparison of the LGM ice sheet configuration to other studies and the details of the glacial forcing index are not the primary goal of this study and will be addressed in manuscripts currently in preparation.

**1.2   Supplementary Material**

[41 figures in the Supplementary Material is way too much and makes it difficult to follow the paper. These figures are referenced too many times and cuts the flow for the reader. Please consider reducing the amount of figures with those which are strictly necessary for your article. For example, you could merge figures S2-S6.] While we agree that the Supplementary Material is quite long, it provides valuable information for readers interested in specific details of the study. However, reducing its length by merging figures with similar content is possible and will be applied in the revised draft.

[Table S1 is complicated to understand for the reader. First, I think you investigate too many parameters, but if the authors decide to follow this approach, then I would suggest splitting it in different sections, such as ice dynamics, climatologies, GIA, GHF (you could add a horizontal line). In addition, you define parameter names which you do not use in your manuscript, I do not think this is necessary] We will re-order and divide the ensemble parameters according to the suggested process category approach.

**2   Technical comments**

[Section 2.3 - Do you apply any basal-stress scaling at the grounding line? If you are using a coarse resolution, scaling basal stress at the grounding line has shown to help to simulate grounding line migration in agreement with high resolution. Actually, this could help to simulate more surges potentially. Do you apply any melt at the grounding line?] The GSM uses the Schoof [2007] grounding line flux condition as implemented in [Pollard and DeConto, 2012]. The authors only recently became aware of issues around this approach for complex 2D geometries likely of most consequence for Antarctica [Reese et al., 2018], and the revised validated treatment [Pollard and Deconto, 2020] has subsequently been implemented in the GSM. Given the geometry of Hudson Strait, we do not expect this change to have much impact. However, to be safe, we are in the process of testing sensitivies for Hudson Strait and will document this in the revisions.

Submarine basal melt is not applied to the grounding line grid cell. Sub-marine face melt is applied at the grounding line if it is a tidewater outlet with an exposed (i.e., no ice shelf) calving face.

[Section 3.3 - I am surprised that using a (regularized) Coulomb law leads to model crashes and a model run time much slower than a Weertman law. Do you have an explanation for

this?] As the regularized Coulomb law negligibly increases basal drag beyond the order of the regularization threshold ($UV_{\mathrm{C,reg}} = 20$ m/yr), we expect it to be much more unstable than the Weertman law according to CFL constraints. This is compounded by the schoofing grounding-line flux iteration in the SSA solution.

It should also be noted that the GSM SSA solution imposes an upper bound of 40 km/yr on SSA ice velocities for this configuration. We suspect that this is higher than most other models. The imposition of this upper bound is itself another non-linearity in the solution that can contribute to both instability (as adding non-linearities will generally decrease convergence of iterative solutions) and stability (by limiting ice velocities).

[Figure 4 - Please use another naming for your "surges" since notation S9 can be confused between you Figure S9 or your surge S9. How is it possible that you obtain the highest buttressing value when your ice shelf is small?] We thank the referee for highlighting this issue. The naming of the surges will be adjusted in the revised draft.

The plotted buttressing time series is given as a fraction of the grounding line longitudinal stress. Therefore, a small ice shelf can lead to large buttressing when the grounding line longitudinal stress is small. We will adjust the y-label in the plot to make this clearer.

[Figure 7, 8, 13, 15 - Put the legend in one plot, you do not need to put it in every plot.] We find interpretation easier when the legend is in each plot. However, we will defer to the editor or Journal staff for guidance on which option is preferred.

**References**

M. Drew and L. Tarasov. Surging of a hudson strait-scale ice stream: subglacial hydrology matters but the process details mostly do not. *The Cryosphere*, 17(12):5391–5415, 2023. doi: 10.5194/tc-17-5391-2023. URL https://tc.copernicus.org/articles/17/5391/2023/.

K. Hank, L. Tarasov, and E. Mantelli. Modeling sensitivities of thermally and hydraulically driven ice stream surge cycling. *Geoscientific Model Development*, 16(19):5627–5652, 2023. doi: 10.5194/gmd-16-5627-2023. URL https://gmd.copernicus.org/articles/16/5627/2023/.

D. Pollard and R. M. DeConto. Description of a hybrid ice sheet-shelf model, and application to Antarctica. *Geoscientific Model Development*, 5(5):1273–1295, 2012. ISSN 1991959X. doi: 10.5194/gmd-5-1273-2012.

David Pollard and Robert M. Deconto. Improvements in one-dimensional grounding-line parameterizations in an ice-sheet model with lateral variations (PSUICE3D v2.1). *Geoscientific Model Development*, 13(12):6481–6500, 2020. ISSN 19919603. doi: 10.5194/gmd-13-6481-2020.

Ronja Reese, Ricarda Winkelmann, and G. Hilmar Gudmundsson. Grounding-line flux formula applied as a flux condition in numerical simulations fails for buttressed Antarctic ice streams. *Cryosphere*, 12(10):3229–3242, 2018. ISSN 19940424. doi: 10.5194/tc-12-3229-2018.

Christian Schoof. Marine ice-sheet dynamics. Part 1. The case of rapid sliding. *Journal of Fluid Mechanics*, 573:27–55, 2007. ISSN 00221120. doi: 10.1017/S0022112006003570.

L. Tarasov and W. Richard Peltier. Impact of thermomechanical ice sheet coupling on a model of the 100 kyr ice age cycle. *Journal of Geophysical Research Atmospheres*, 104(D8):9517–9545, 1999. ISSN 01480227. doi: 10.1029/1998JD200120.

Lev Tarasov and W. R. Peltier. Coevolution of continental ice cover and permafrost extent over the last glacial-interglacial cycle in North America. *Journal of Geophysical Research: Earth Surface*, 112(2):1–13, 2007. ISSN 21699011. doi: 10.1029/2006JF000661.

Lev Tarasov, Kevin Hank, and Benoit S. Lecavalier. The glacial systems model (GSM). 2024.

---

## Author Comment (AC2)

**Author's response to Clemens Schannwell's Comment 1**

June 28, 2024

**1 General Comment**

[The manuscript by Hank & Tarasov presents a suite of simulations that aims to investigate the sensitivity of the Glacial Systems Model (GSM) to processes that have previously been shown to have an influence on Heinrich Event characteristics in other models. The novelty in comparison to previous studies is that the authors use transient forcing and present an ensemble that covers a much larger parameter space. Based on this, they confirm earlier findings from e.g. Mann et al. [2021], Schannwell et al. [2023] that GSM also exhibits two different states - a streaming and cycling surging state. Moreover, they show that in their model, geothermal heat is a major controlling factor while ice shelves and their buttressing matter very little.]

[Overall, the topic of the paper is interesting and I believe that the science behind it is sound. However, in its present form the reader is drowned in the sheer amount of simulations that are presented in a rather unstructured fashion. The lack of structure in the paper and particularly in the methods sections leaves the reader confused as to what exactly the authors did and makes it challenging to find a common thread through the manuscript as well as evaluate the scientific results. Therefore, I do not think the paper is ready for publication in its current form and a substantial restructuring and part-rewrite is required. I recommend the authors take into account my comments listed below. I hope the authors find my comments helpful.]

We thank the referee Clemens Schannwell for their constructive comments. A point-by-point reply is reported below, with referee comments in orange and our replies in black. We agree with the specific referee comments not listed here and will revise the manuscript accordingly.

We will restructure the manuscript according to the referee's helpful suggestions, including a short roadmap at the end of the introduction and a table summarizing the reference setup and all experiments.

**2 Specific comments**

[The entire methods sections felt really incoherent almost as if sections were being added as new ideas for sensitivity simulations were designed. By the time I got to the results, there were so many simulations and sub-ensembles that I was utterly confused what was done in the end and what the different simulations were actually referring to. The absence of an experiment table or a flowchart of how the different sub-ensembles were generated just confounded my confusion. My suggestion is to remove section 2.1. "Modelling approach" and really start with the model description (your current section 2.3). In there, I would like to see what type of model GSM is! I believe it is an ice-sheet model but this is never really spelled out. This section should also incorporate all of the coupling that is included e.g. subglacial hydrology etc. as well as time-steps, resolution and the like. Then, I would follow that by all the boundary conditions e.g. your current sections 2.4 to 2.5.2. Then move on to the section that you call "Ensemble parameter vector" which I suggest to rename to "Creation of sub-ensembles" as I believe it makes it more accessible. In this subsection, you absolutely need a table with the different sub-ensembles as well as a flowchart illustrating how each of these sub-ensembles was generated, e.g

*Baseline ensemble -> sieve 1 -> sieve 2 etc. Also, I find the section title "Bounding experiments" somewhat confusing. Do you mean end member scenarios? If so, why not name it like that?]* The methods section in the revised draft will be restructured following some of the above suggestions, including a table summarizing all experiments, merging the *Modeling approach* and *Model description* sections, and re-naming of sub-sections. Furthermore, a description of the GSM will be submitted to GMD for open review prior to submission of our revisions.

*[In your results section you often speak about specific parameter vectors, for example in the caption of Figure 4 you mention "parameter vector 1". It is unclear to me to what sub-ensemble this refers to as parameter vector 1 is never really defined in the manuscript. So, I strongly recommend to define these parameter vectors somewhere in the methods section (e.g. in a table), so that the reader can go back and check what this refers to.]* In the current model setup, the GSM has 52 input parameters (Table. S1). A parameter vector holds one value for each of these input parameters. As such, each parameter vector fully specifies how the configuration of the GSM varies between sub-ensemble members. Our reference ensemble contains 20 parameter vectors (20 GSM runs). For our experiments, we vary other aspects of the GSM configuration (e.g., a different geothermal heat flux (GHF) than used in the reference ensemble) but use the same 20 parameter vectors. This allows us to determine the effect of the change in model configuration while (albeit partly) considering associated parametric uncertainties. Fig. 4 in our submission shows the GSM output when using the first parameter vector (first set of 52 input parameters) in our reference ensemble. The revised draft will include a more detailed description of this terminology and design.

*[This issue was already raised by the other reviewer, but I believe that the effect of the transient climate forcing certainly warrants a discussion. First of all, it is never mentioned what climate forcing is used to drive the model. This goes back to my previous point that I am not sure whether GSM is an ice-sheet model or a coupled climate-ice sheet model. In any case, it remains unclear to me whether the differences you report here are due to your parameter changes or due to the changing transient forcing and it seems quite likely that there is at least some signal from the transient climate forcing in your results. This ought to be acknowledged and discussed.]* To answer this comment, we expand on the logic described in the comment directly above. Fig. 7, for example, shows the effect when applying different GHFs for parameter vector 18. Therefore, the input parameters and the applied transient climate forcing are the same for all runs shown in Fig. 7. The differences are, therefore, not caused by a different climate forcing.

The climate forcing, however, varies between different parameter vectors depending on the values of the climate input parameters in Table S1. Or to rephrase, each experiment is carried out with the same set of 20 different transient climate forcings (20 parameter vectors). As indicated in our response to the first referee comment, *The key idea behind this approach is to identify physical processes that show a significant effect across simulations with different ensemble parameters. This both increases confidence in our results and reduces the possibility that an observed modeling response is only due to, e.g., the chosen climate forcing.* The ensemble description will be updated in the revised draft, including a plot showing the ice volume of all parameter vectors when run in the reference configuration, and MIS 3 and LGM example time-slice map plots.

*[On a related note, is it correct that your reference setup (e.g. Fig 7) does not show any Heinrich events, but simply a steady ice stream? I believe it is never mentioned, but your algorithm does not detect any surges in that time series right? But then it is unclear to me how you calculate the period and number of surges presented in Table 1.]* It is correct that the algorithm does not detect any surges (or, more precisely, detects less than 3 surges) when parameter vector 18 is run in the reference configuration. As described in the text and our response to the first referee comment, the reference ensemble (20 runs) can be divided into two high-variance (in parameter values and run metrics) subgroups: one subgroup of 10 for which all runs have more than 2 surges (at the higher resolution), and one for which this is not the

case for the higher resolution run but is the case for the corresponding lower resolution run (i.e., with the same parameter vector). Parameter vector 18 belongs to the second subgroup, which is not included in the results presented in, e.g., Table 1. However, Fig. 7 clearly shows that surges occur when a smaller GHF is applied.

[You introduce the term "Minimum Numerical Error Estimate (MNEE)" which at least to me is a new concept. I think it should be clearly stated whether this is a common concept from the field of numerical analysis or something that you have introduced (I see that you used a the same approach in your previous paper). If you introduced it, it would be helpful to briefly(!!) motivate why this is a useful quantity. Because from what you mention in between lines 274 – 279, this seems a rather arbitrary choice (increasing your Picard iteration by one). I am also confused that your only parameter to measure MNEE is the "number of iterations". At the very least, I was expecting a combination of "number of iterations" and "residuals". The reason for this is that I expect the residual to be higher after 4 iterations if you are in a rapidly changing state than after 2 iterations in a relatively stable state. In my view, the way you have defined it is inconsistent because it does not tell anything about whether your solver has actually converged or not. For example, you can increase your number of iterations to 50, but that does not say anything about the quality of your solution. Also since GSM is run in serial, why not compare it to the solution from a direct solver like UMFPACK?] Yes, the concept of MNEEs was introduced in Hank et al. [2023]. They are specified thresholds for determining whether a difference in a sensitivity experiment is within a minimal estimate for the numerical uncertainty of the ice sheet dynamics solution. Given the non-linearity of surge onset and termination, this plays a potentially important part in our analysis. More specifically, as described in the text, MNEEs for the GSM are the maximum model difference across two different model simulations: 1) imposing stricter (than default) numerical convergence (what the referee refers to as smaller residual) and 2) imposing stricter numerical convergence with increased maximum iterations for the outer Picard loop (from 2 to 3, solving for the ice thickness) and the non-linear elliptic SSA (Shallow-Shelf Approximation) equation (from 2 to 4, solving for horizontal ice velocities). Therefore, the MNEEs are based on simulations with 1) smaller residual and 2) the combination of increased maximum number of iterations and smaller residual, but never just on an increased number of maximum iterations. The description of MNEEs will be updated in the revised manuscript to clarify this.

[At the end of the introduction you list five research questions that your paper aims to answer. This is admittedly a bit of a subjective matter, but my impression was that these questions are very generic and could be summarized with something like are our Hudson surges sensitive to perturbations in geothermal heat flux, GIA, ocean melting, etc. While certainly worth exploring, to me, this type of question layout is more suited to a thesis format but not necessarily for a paper. I also could not help, but get the impression that you do a bit of everything which results in a lack of focus what you are really trying do address here. For example, I do not think that your paper actually addresses Q3 as you solely focus on the Hudson ice stream. As an add-on, I am also in favour of a short paper roadmap at the end of the introduction especially when it is as complicated and long as this paper.] We will add a short roadmap to the end of the introduction.

Yes, what we are addressing is broad, but we do not see an alternative given our stated intention: *However, an extensive study simultaneously investigating the relative role of each proposed HE hypothesis is still missing. Furthermore, previous model-based tests of HE-related Hudson Strait surge cycling have not addressed uncertainties in key potentially relevant processes and inputs. These include the deep geothermal heat flux under Hudson Strait, glacio-isostatic adjustment, and the form of the basal drag law employed.* In short, instead of tidy, idealized experiments addressing single hypotheses and ignoring the role of other possible mechanisms, we aim to address the lack of a comprehensive assessment of the relative role of proposed mechanisms while addressing key relevant uncertainties. Ignoring the latter negates the interpretative value of the former. We believe the stated research questions provide a clear structure for this

assessment, and the question presentation succinctly summarizes the motivation for each question.

Furthermore, we suspect that not every reader will be interested in every detail of the results. Organizing the discussion section according to the research questions outlined in the introduction provides an easy way to get the most relevant information and allows the reader to then jump to individual results for more details.

As the majority of HE IRD in the North Atlantic is attributed to Hudson Strait, examining the ice shelf volume around it allows us to, at least partly, address Q3, especially considering that proxy records indicate an seasonally ice-free Labrador Sea during MIS3 [Hillaire-Marcel et al., 1994, Hesse et al., 1999, De Vernal et al., 2000, Gibb et al., 2014]. Additionally, we will include the Labrador Sea and Baffin Bay ice shelf volume at key time slices in the revised draft.

**3 Technical corrections**

[L2: I think instead of using the term Glacial Systems Model, it would be better to refer to the type of model like coupled ice sheet-subglacial hydrology model. Of course, if you'd like to keep GSM in there, you could combine these two.] Additional details about the exact model type will be added to the revised draft. However, Glacial Systems Model (GSM) is simply the name of our model (e.g., comparable to Parallel Ice Sheet Model (PISM)). Since this name has been used in several publications [e.g., Drew and Tarasov, 2023, Hank et al., 2023], we prefer to keep it.

[L38: Not to be too picky, but I would argue that we did test the sensitivity of mPISM to the geothermal heat flux in our Schannwell et al. [2023] paper] We thank the referee for pointing this out. We will adjust the text accordingly.

[L89: What is the physical mechanism for the choice to allow sliding at sub-freezing temperatures?] Observational and experimental evidence suggests that basal sliding starts below the pressure melting point and ramps up as the pressure melting point is approached [e.g., Barnes et al., 1971, Shreve, 1984, Echelmeyer and Zhongxiang, 1987, Cuffey et al., 1999, McCarthy et al., 2017, Mantelli et al., 2019]. Furthermore, Mantelli et al. [2019] show that an abrupt sliding onset at the transition from a cold-based ice sheet to an ice sheet bed at the pressure-melting point causes refreezing on the warm-based side and, therefore, cannot exist. An additional numerical argument can be made on numerical grounds for coarse horizontal grid resolutions. It is unlikely that an entire grid cell reaches the pressure-melting point within one time step [e.g., 25×25km in 1 year Hank et al., 2023]. All of these aspects are described in detail in Hank et al. [2023]. To clarify this, we will add this reference to the corresponding sentence.

[L93: Be precise here! What components are included in the current setup? Also asynchronous coupling mean different things to different communities. Does that mean you run your GIA model accelerated? Moreover, for your GIA model what kind of ice thickness distribution do you prescribe in the southern hemisphere or the other northern hemispheric ice sheets? This confusion originates from the fact that you never specify what your modelling domain is. In addition, it would be helpful to provide more detail about the GIA model, because out of the blue in the results section, you start talking about local and non-local GIA.] There is no acceleration. Visco-elastic GIA models are generally run on spherical harmonic grids and, therefore, must be global. We use the ice sheet chronologies from recently completed history matching for the non-NA inputs into the GIA calculation. As stated in section 2.1 of our submission: *The topography and sediment cover of the entire model domain are shown in Fig. S1.* As an aside, we have never seen "asynchronous" alone imply acceleration in modeling. Our revised submission will explicitly answer the reviewer's questions.

[L121: What is that resolution in kms for the Hudson ice stream. How does this coarse resolution affect your ability to model surging. We saw quite dramatic changes when we increased resolution from 50 km down to 20 km some years ago.] The grid cell size in Hudson Strait is

roughly 25x25 km. As stated in the text *Due to the inclusion of a resolution-dependent basal temperature ramp [Hank et al., 2023], the differences in surge characteristics between the coarse resolution runs (horizontal grid resolution of $\Delta_{\text{lon}} = 1.0°$, $\Delta_{\text{lat}} = 0.5°$) and the reference runs ($\Delta_{\text{lon}} = 0.5°$, $\Delta_{\text{lat}} = 0.25°$) are generally within the MNEEs (Fig. 6). While finer (than the reference setup) horizontal grid resolutions are currently unfeasible in the context of this study, given the results of resolution response testing of surge cycling down to* 3.125 km *horizontal grid resolution in Hank et al. [2023], the differences in surge characteristics for finer resolutions are also expected to be within the MNEEs.*

[L167: From your description it is unclear whether or not you do "Schoofing", meaning whether you prescribe the Schoof flux as an internal boundary condition at the grounding line. Please clarify.] As stated in our response to the first referee comment: *The GSM uses the Schoof [2007] grounding line flux condition as implemented in [Pollard and DeConto, 2012]. The authors only recently became aware of issues around this approach for complex 2D geometries likely of most consequence for Antarctica [Reese et al., 2018], and the revised validated treatment [Pollard and Deconto, 2020] has subsequently been implemented in the GSM. Given the geometry of Hudson Strait, we do not expect this change to have much impact. However, to be safe, we are in the process of testing sensitivities for Hudson Strait and will document this in the revisions.*

[L197: What do you do if you do not have any ocean points under your floating ice shelf? Do such situations arise?] We are not clear on what context the referee is referring to. However, for each of sub-shelf melt, GIA within the ice sheet grid, and ice dynamics, the GSM handles a changing ocean mask, and sub-marine temperatures are spatially extrapolated as needed for the subshelf melt calculation. This will be detailed in the forthcoming GSM description submission.

[L199: Do you mean you tune your parameters for present-day ice sheets, because there is no Laurentide today.] The text will be corrected to *computed melt brackets present-day observations* for major Antarctic ice shelves (e.g., ...).

[L223: This makes it sound as if GSM has an ocean component coupled to it, where in fact you are using ocean temps. from a GCM simulations, right? If so, please rephrase.] Yes, the ocean temperatures are derived from the TraCE deglacial simulation run with the Community Climate System Model Version 3 [CCSM3, Liu et al., 2009]. We also already state in our submission *Using a glacial index approach, the ocean temperature chronology is interpolated between full glacial (last glacial maximum) and present day conditions for all other time slices.* However, to better avoid the confusion the reviewer is indicating, we will change the phrase to *GSM ocean temperature forcing.*

[L230: Even for me as an ice-sheet modeller, this is getting quite hard to follow here. I am not sure, but is your ice shelf removal simply a very high basal melt rate that melts your ice shelf away? If so, what is the time it takes for the ice shelf to disappear and how might this potentially rather gradual removal affect the response in comparison to a sudden removal of the ice shelf?] The ice shelf removal experiments are simply based on increased ocean temperatures (as shown in Fig. 3). This experimental design is in line with proxy records indicating ocean temperature increases over a 1 to 2 kyr interval prior to HEs [e.g., Marcott et al., 2011]. The mechanism is also explicitly motivated in our initial presentation of $Q_2$: *A $2°C$ increase in the sub-surface ocean temperature has been shown to cause a* 6 *fold increase in the ice shelf basal melt rates in front of Hudson Strait ($\sim 6 \frac{\text{m}}{\text{yr}}$ to 35-40 $\frac{\text{m}}{\text{yr}}$) in simulations with an ocean/ice-shelf model [Marcott et al., 2011].* Since the GSM incorporates the relevant physics for rapid ice shelf disintegration, we consider this approach physically more defensible than artificially removing the entire ice shelf.

As stated in the text *The ocean temperature at the relevant depth is then used to calculate the sub-shelf melt $M_{\text{SSM}}$ and terminus face melt $M_{\text{face}}$.* Since both $M_{\text{SSM}}$ and $M_{\text{face}}$ depend on ensemble parameters and the ocean temperature of the current experiment, the time it takes for the ice shelf to disappear varies between different parameter vectors and experiments. The ice shelf response to a change in ocean temperature and calving varies from no significant change to rapid ($< 100$ yr) ice shelf disintegration (e.g., Fig. 13).

[L349: Again, this is pretty much what we showed in Schannwell et al. [2023].] A brief comparison will be added to the discussion.

[L365: What does crashing mean here? Solvers did not converge anymore?] Yes, the solvers did not converge with the specified minimum model time step of 0.015625 yr. This threshold could be lowered, but given how far this is already below nominal CFL for our grid resolution, such small time-steps raise concerns about non-linear numerical instabilities that will distort our analysis.

[L366: I think, this certainly needs some discussion why you think the switch from Weertman to Coulomb makes such a big difference in the run time.] As mentioned in our response to the first referee comment: *As the regularized Coulomb law negligibly increases basal drag beyond the order of the regularization threshold ($UV_{C,reg} = 20$ m/yr), we expect it to be much more unstable than the Weertman law according to CFL constraints. This is compounded by the schoofing grounding-line flux iteration in the SSA solution.*

*It should also be noted that the GSM SSA solution imposes an upper bound of 40 km/yr on SSA ice velocities for this configuration. We suspect that this is higher than most other models. The imposition of this upper bound is itself another non-linearity in the solution that can contribute to both instability (as adding non-linearities will generally decrease convergence of iterative solutions) and stability (by limiting ice velocities).* A numerical challenge with Coulomb plastic (regularized or not) is that, unlike Weertman, there is a constant basal drag term in the SSA equation that does not automatically/implicitly have the correct sign. We will add a brief description of all this to the revised draft.

[L392: In the interest of shortening the paper, consider removing everything regarding the regularized Coulomb and simply state that run times were too long to make these runs feasible.] We respectfully decline this suggestion. A major motivation of this paper was to comprehensively address key uncertainties not (or at least mostly not) addressed in previous tests of hypotheses for explaining Heinrich events. Uncertainties in basal drag law are one such uncertainty; thus, we consider it important to retain this subsection.

[L457: Are you sure your mean "increase" here? It is possible, but I would argue an elevation decrease is more often than not associated with an accumulation decrease.]

Yes, increase is correct here, but the statement should have been *precipitation (and therefore accumulation over most of the ice sheet) generally increases ....* Assuming everything else is the same, a lower ice sheet surface elevation will have a higher precipitation rate (relation between atmospheric lapse rate and the Clausius–Clapeyron formula for saturation vapour pressure) though this is complicated by slope orographic forcing dependencies of precipitation in the GSM.

[L569: What are pseudo-Hudson Strait surges? Bassis et al. use an idealized setup that is based on the geometry of the Hudson Strait. Does GSM have the marine-ice-cliff instability mechanism implemented? This is an integral part of the Bassis et al. mechanism and renders the comparison pretty far-fetched if it doesn't.] "pseudo"-Hudson Strait surges is an earlier notation and will be removed in the revised manuscript.

Yes, the GSM does have the MICI, but only for iced grid cells adjacent to neighbouring open ocean. This will be made explicit in the revisions. However, as stated in the text: *Due to the numerous differences in the model setup (e.g., model domain considered, grid discretization near the grounding line, GIA model, calving and sub-shelf melt implementations, and the lack of ice thermodynamics in Bassis et al. [2017]), we do not aim to directly replicate the experiments in Bassis et al. [2017]. Instead, we examine the role of SSOW in a HE context by applying a sub-surface ocean temperature increase for every DO event.*

[L574: I find these melt-rates unreasonably high. I mean maximum present-day melt rates are around 100 m/yr and you have four times the melt during the glacial? That seems very hard to believe.] 400 m/yr are the highest melt rates occurring in all of our experiments, including the end-member scenarios. Since some end-member scenarios were specifically designed to increase the melt rate, it is unsurprising that the maximum modeled melt rates exceed the

maximum observed present-day melt rates. The melt rates in the reference setup are generally below 100 m/yr. This will be made explicit in the revisions.

[L606: I am pretty sure that you did not show synchronization of your HEs with the Dansgaard-Oeschger cycles.] We agree that we did not show a synchronization of HEs with the coldest phases of the Bond cycles. However, the timing of surges is affected by the applied additional ocean forcing, indicating the possibility of synchronization. We will clarify this in the revised draft.

**4    Figures**

[The Figures are overall of good quality. What I am missing is a Figure of the modelling domain. If it is a global setup, this is not needed, but then this needs to be stated clearly in the text.] As stated in section 2.1 of our submission: *The topography and sediment cover of the entire model domain are shown in Fig. S1.* Additionally, as indicated above, in our revised version, we will include some whole NAIS (showing whole model grid) example time slices.

[Fig. 4: and throughout. I am not a big fan of pythons default option to have the scientific notation on top of each subplot in pretty small font. For a second I thought that your flux was as high as 2 Sv. You could try using mSv instead or work that exponent into your axis titles.] Agreed, we will change to clearer labelling.

[Fig. 5: Consider removing repetitive colourbars and axis labels for the benefit of larger panels.] Given the number of panels, we find interpretation easier when the colorbars and labels are in each plot frame. However, we will test removing repeated (within row or column) grid axis labels for the revised draft. We will also defer to the editor and Journal staff for further guidance.

[Fig. 7: Judging from this, your reference simulations has no Heinrich events? But why is there an ice volume increase at the same time as the ice flux increases? What is the origin of this?] As already addressed in our response above, parameter vector 18 (base vector shown in Fig. 7) is part of the ensemble with < 3 surges detected. In the Fig. 7 plots, the reference run is shown in orange. The most significant growth in Hudson Strait ice flux occurs between 70 and 60 kyr BP. This is also the time of the most significant whole ice sheet volume growth in the model. The increased ice volume eventually increases ice flux to and, consequently, through Hudson Strait (Fig. 1 in this response). As this is peripheral to the focus of our submission, we see no need to add this brief analysis to it.

[Supplementary Figures: I can also only reiterate what the other reviewer mentioned, 40 Figures in the supplement is a lot and you should really make sure that each of these Figures serves a purpose. For example, what does the Figure S8 add? In the text it is referenced to show a more gradual increase in ice flux, but I have no idea how a snapshot of the velocity field could convey such a message.] As mentioned in our response to the first referee comment, we will reduce the amount of figures by merging plots with similar content. We have verified that every supplemental figure is referenced at least once in the main text. We are aware this is on the long side, but we are generally going by our rule of thumb that if more than one reader is likely to want to see the plot and less than a dozen are likely to, then stick in the supplement. Most readers are free to ignore the supplement and just note that there is graphical documentation of the relevant claim. We will, however, carefully go through all the supplemental figure referencing and reconsider if at least a few can be eliminated.

Fig. S8 is used in the following sentence: *During these surges, ice transport from Hudson Bay and Foxe Basin through Hudson Strait (and other outlets) towards the ice sheet margin increases (e.g., Fig. S8).* The confusion may stem from the abbreviation S8 for surge 8 (appearing in *"gradual increases in Hudson Strait ice flux (S8 and S9)"*), which will be rectified in the revised draft.

[Figure]

Figure 1: The ice sheet surface elevation and basal ice velocity of parameter vector 18 are shown at 70.0 kyr BP and 60 kyr BP. The black contour is the present-day coastline provided by *cartopy* [Met Office, 2010 - 2015].

**References**

P. Barnes, D. Tabor, and J. C. F. Walker. The friction and creep of polycrystalline ice. *Proceedings of the Royal Society of London. Series A, Mathematical and Physical Sciences*, 324(1557):127–155, 1971. ISSN 00804630. URL `http://www.jstor.org/stable/77933`.

Jeremy N Bassis, Sierra V Petersen, and L Mac Cathles. Heinrich events triggered by ocean forcing and modulated by isostatic adjustment. *Nature*, 542(7641):332—334, February 2017. ISSN 0028-0836. doi: 10.1038/nature21069. URL `https://doi.org/10.1038/nature21069`.

K. M. Cuffey, H. Conway, B. Hallet, A. M. Gades, and C. F. Raymond. Interfacial water in polar glaciers and glacier sliding at -17 °C. *Geophysical Research Letters*, 26(6):751–754, 1999. ISSN 00948276. doi: 10.1029/1999GL900096.

Anne De Vernal, Claude Hillaire-Marcel, Jean Louis Turon, and Jens Matthiessen. Reconstruction of sea-surface temperature, salinity, and sea-ice cover in the northern North Atlantic during the last glacial maximum based on dinocyst assemblages. *Canadian Journal of Earth Sciences*, 37(5):725–750, 2000. ISSN 00084077. doi: 10.1139/cjes-37-5-725.

M. Drew and L. Tarasov. Surging of a hudson strait-scale ice stream: subglacial hydrology matters but the process details mostly do not. *The Cryosphere*, 17(12):5391–5415, 2023. doi: 10.5194/tc-17-5391-2023. URL `https://tc.copernicus.org/articles/17/5391/2023/`.

Keith Echelmeyer and Wang Zhongxiang. Direct Observation of Basal Sliding and Deformation of Basal Drift at Sub-Freezing Temperatures. *Journal of Glaciology*, 33(113):83–98, 1987. ISSN 0022-1430. doi: 10.3189/s0022143000005396.

Olivia T. Gibb, Claude Hillaire-Marcel, and Anne de Vernal. Oceanographic regimes in the northwest Labrador Sea since Marine Isotope Stage 3 based on dinocyst and stable isotope proxy records. *Quaternary Science Reviews*, 92:269–279, 2014. ISSN 02773791. doi: 10.1016/j.quascirev.2013.12.010. URL `http://dx.doi.org/10.1016/j.quascirev.2013.12.010`.

K. Hank, L. Tarasov, and E. Mantelli. Modeling sensitivities of thermally and hydraulically driven ice stream surge cycling. *Geoscientific Model Development*, 16(19):5627–5652, 2023. doi: 10.5194/gmd-16-5627-2023. URL `https://gmd.copernicus.org/articles/16/5627/2023/`.

Reinhard Hesse, Ingo Klauck, Saeed Khodabakhsh, and David Piper. Continental slope sedimentation adjacent to an ice margin. III. The upper Labrador Slope. *Marine Geology*, 155(3-4):249–276, 1999. ISSN 00253227. doi: 10.1016/S0025-3227(98)00054-1.

C. Hillaire-Marcel, A. De Vernal, G. Bilodeau, and G. Wu. Isotope stratigraphy, sedimentation rates, deep circulation, and carbonate events in the Labrador Sea during the last ~200 ka. *Canadian Journal of Earth Sciences*, 31(1):63–89, 1994. ISSN 00084077. doi: 10.1139/e94-007.

Z. Liu, B. L. Otto-Bliesner, F. He, E. C. Brady, R. Tomas, P. U. Clark, A. E. Carlson, J. Lynch-Stieglitz, W. Curry, E. Brook, and et al. Transient Simulation of Last Deglaciation with a New Mechanism for Bolling-Allerod Warming. *Science*, 325(5938): 310314, Jul 2009. ISSN 1095-9203. doi: 10.1126/science.1171041. URL `http://dx.doi.org/10.1126/science.1171041`.

E. Mantelli, M. Haseloff, and C. Schoof. Ice sheet flow with thermally activated sliding. Part 1: the role of advection. *Proceedings of the Royal Society A: Mathematical, Physical and Engineering Sciences*, 475(2230):20190410, 2019. ISSN 14712946. doi: 10.1098/rspa.2019.0410.

Shaun A. Marcott, Peter U. Clark, Laurie Padman, Gary P. Klinkhammer, Scott R. Springer, Zhengyu Liu, Bette L. Otto-Bliesner, Anders E. Carlson, Andy Ungerer, June Padman, Feng He, Jun Cheng, and Andreas Schmittner. Ice-shelf collapse from subsurface warming as a trigger for Heinrich events. *Proceedings of the National Academy of Sciences of the United States of America*, 108(33):13415–13419, 2011. ISSN 00278424. doi: 10.1073/pnas.1104772108.

C. McCarthy, H. Savage, and M. Nettles. Temperature dependence of ice-on-rock friction at realistic glacier conditions. *Philosophical Transactions of the Royal Society A: Mathematical, Physical and Engineering Sciences*, 375(2086):20150348, 2017. ISSN 1364503X. doi: 10.1098/rsta.2015.0348.

Met Office. *Cartopy: a cartographic python library with a Matplotlib interface.* Exeter, Devon, 2010 - 2015. URL https://scitools.org.uk/cartopy.

D. Pollard and R. M. DeConto. Description of a hybrid ice sheet-shelf model, and application to Antarctica. *Geoscientific Model Development*, 5(5):1273–1295, 2012. ISSN 1991959X. doi: 10.5194/gmd-5-1273-2012.

David Pollard and Robert M. Deconto. Improvements in one-dimensional grounding-line parameterizations in an ice-sheet model with lateral variations (PSUICE3D v2.1). *Geoscientific Model Development*, 13(12):6481–6500, 2020. ISSN 19919603. doi: 10.5194/gmd-13-6481-2020.

Ronja Reese, Ricarda Winkelmann, and G. Hilmar Gudmundsson. Grounding-line flux formula applied as a flux condition in numerical simulations fails for buttressed Antarctic ice streams. *Cryosphere*, 12(10):3229–3242, 2018. ISSN 19940424. doi: 10.5194/tc-12-3229-2018.

Christian Schoof. Marine ice-sheet dynamics. Part 1. The case of rapid sliding. *Journal of Fluid Mechanics*, 573:27–55, 2007. ISSN 00221120. doi: 10.1017/S0022112006003570.

R. L. Shreve. Glacier sliding at subfreezing temperatures. *Journal of Glaciology*, 30(106):341–347, 1984. ISSN 00221430. doi: 10.1017/S0022143000006195.

---

## Author Comment (AC3)

**Author's response to Jorge Alvarez-Solas's Comment 1**

June 28, 2024

[In this article, Hank and Tarasov, perform several simulations of the Laurentide ice sheet of the last glacial period in order to elucidate the most likely candidates to explain the origin of Heinrich events. The subject and the way it is addressed in the introduction and discussion are of very high relevance. The paper is potentially very well suited for publication in Climate of the Past. I see, however, two major deficiencies of the current manuscript: A questionable and irreproducible experimental setup and the lack of adequate conclusions. I expand these aspects in the following.]

We thank the referee Jorge Alvarez-Solas for their constructive comments. A point-by-point reply is reported below, with referee comments in orange and our replies in black. We address the claimed *two major deficiencies*, which was likely due to a misunderstanding of the experimental design in our detailed responses below.

**1 General comments regarding the experimental setup and the exploration of the parameter space:**

[Your experimental setup is firstly very hard to understand and secondly not reproducible at all. It is hardly understandable because your entire exploration of the parameter space relies on perturbing the values of the parameters of your table S1. However the majority of these parameters are not explained at all in the manuscript. The very pertinent questions you are addressing in this article are conveniently described at the end of the Introduction paragraph. They summarize the different mechanisms triggering Hudson Strait ice surges in the context of HEs. However, it is difficult to understand how exploring, for example, the values of the "northwestern desert-elevation cutoff" is relevant in this context. The equations on which these not-defined parameters apply are not present in the manuscript, so the reader can not even guess what influence they have on the scientific questions you are addressing. This is the case because the exploration of these parameters only helps to see whether a given realization of the model has fulfilled or not your sieves, but does not give any mechanistic picture of their influence on ice stream behavior. Nor these experiments can be reproduced with other models of similar physics. An article should be as shelf-contained as possible, so if you analyze an ensemble of simulations that have fulfilled a given sieve, the reader can understand why that is the case and the implications for your conclusions without having to open the experimental setup of another manuscript or guess that it is going to be more deeply explained in another article in preparation.] and [In my opinion, both these two problems would be solved by the following strategy: You could fix several of the parameters that have already allowed you to fulfill the sieves and that are mechanistically irrelevant for the ice stream behavior to a single documented value. The equations on which these parameters appear could be summarized in an appendix. Then, you could systematically explore the parameter space of the processes that are suspected to be very influential on the ice-stream dynamics (e.g. sliding laws, effective pressure, dependence on bed type and base thermal state...; see below). And finally you could describe how these different values affect the nature of the simulated Hudson Strait ice

 The reviewer apparently does not understand that our experimental design is precisely what they suggest: "fix several of the parameters"; however, we do this twenty times over our 20 base parameter vectors for all 52 parameter vector components. Upon re-reading our section 2.2, we can now see that the text would benefit from more clarity that we use the same 20 GSM parameter vectors for all experiments (to do otherwise would make no sense).

This clarification should address the reviewer's concerns. However, in case they were clear on the experimental design and still have issues with the use of an ensemble, we add the following points.

A flaw in the reviewer's reasoning is their presumption that most of the ensemble parameters are apriori "mechanistically irrelevant". Thermodynamic surge cycling will depend on energy balance, which depends on surface temperature, precipitation, and ice thickness. As such, most of the ensemble parameters related to climate forcing (the majority of our ensemble parameters) are potentially relevant. This also holds for the remaining ensemble parameters related to GIA, mass-balance processes, ice deformation, and basal drag. The key point of our ensemble design is to address parametric uncertainties. This is done by extracting a high-variance sample of non-implausible model configurations (each specified by a parameter vector) and then conducting sensitivity experiments (with respect to inputs or process suppression, not ensemble parameters) on this same sample of parameter vectors. Otherwise, it would remain unclear whether the results obtained through a set of sensitivity experiments are only valid for a single chosen single configuration (i.e., a single fixed parameter vector).

As to *"An article should be as self-contained as possible"*, this would entail every modelling paper describing every parameter in the model, every equation, and every discretization. We will add more overall description of the ensemble parameters, but it is unreasonable to expect a detailed description of each parameter (which would extremely bloat every GCM-based modelling paper). Furthermore, a GSM description will be submitted to GMD before submission of our revisions.

In summary, and as already indicated in our responses to the other referee comments, the goal of this study is not to explore the effect of ensemble parameters on HEs. Instead, the ensemble-based approach aims to (albeit incompletely) account for the uncertainties associated with these ensemble parameters (ranges in Table S1). In comparison, other modelling studies often set the values of model parameters comparable to the ones presented in Table S1 to a single value and, therefore, completely ignore the associated parametric uncertainties.

**2 Specific comments regarding the experimental setup and the exploration of the parameter space:**

[Have you explored different values of $n_{b,hard}$? I assume not (from what is shown in table S1). If not, why not simply use the same value as for $n_{b,soft}$? That will reduce one degree of freedom.]

The hard-bedded sliding exponent has not been explored in this context. Based on our reading of the literature, we judge the form of the soft bedded sliding as less constrained with more potential impact on results given the generally lower basal drag and higher fluxes.

[According to equations (2) and (3) your sliding coefficient, $C_b$ is a function of the bedrock type (soft sediments vs hard), the thermal character of the base and the effective pressure. Why do you need additionally to change the exponent of the sliding law over soft and hard bedrocks?]

Weertman's reasoning of controlling obstacle sizes places the hard bed exponent at 2 or 3. However, based on the results of a basal drag inversion for Greenland [Maier et al., 2021], a value of 4 appears to be more representative (at least for Greenland). Soft bedded sliding at small scales is likely Coulomb plastic, but the appropriate form at large scale has been subject to long ongoing debate. Therefore, there is no reason to assume that the exponent should be the same for both hard and soft beds.

[I deduce from table S1 that your maximum allowed value for $h_{\mathrm{wb,Crit}}$ is 1.0. Why limit it to that relatively low value? What happens if you significantly increase it?] and [Additionally, what does the model do if $h_{\mathrm{wb}}$ reaches $h_{\mathrm{wb,Crit}}$? Neff could not be reduced further (it would be purely 0!! From equation (4)), but what happens with the excess of heat? Can you create additional basal water? If not, could you diagnose the "free" values of $h_{\mathrm{wb}}$? How sensitive are the surging cycles to this basal water limit?] To clarify, $h_{\mathrm{wb,Crit}}$ is an estimated effective bed roughness scale, not the maximum basal water thickness. The maximum basal water thickness is $h_{\mathrm{wb,max}} = 10$ m. All additional sub-glacial meltwater ($h_{\mathrm{wb}} > 10$ m) leaves the ice sheet. We will re-do a set of simulations with the 10 m limit raised to 100 m to examine what impact this has.

While it is true that $N_{\mathrm{eff}}$ in Eq. 4 can reach 0 kPa, the addition of $N_{\mathrm{eff,min}} = 10$ kPa in the denominator of Eq. 3 enforces that the effective pressure used to determine the basal sliding coefficient $C_b$ never falls below $N_{\mathrm{eff,min}}$.

Eq. 4 and its exponent 3.5 are based on the work of Flowers [2000]. The parameter range for $h_{\mathrm{wb,Crit}}$ expands around the $h_{\mathrm{wb,Crit}} = 0.1$ m value used by Flowers and Clarke [2002].

The effect of the local basal hydrology model and its parameters (e.g., $h_{\mathrm{wb,max}}$, $h_{\mathrm{wb,Crit}}$, $N_{\mathrm{eff,min}}$) on surges has also been extensively examined in previous studies [Drew and Tarasov, 2023, Hank et al., 2023]. We will clarify all of the above in the revised manuscript.

[You introduce a Coulomb dragging law in equations (5a) and (5b). How is $C_c$ variable? In space or in time? According to what? (not deducible from the manuscript or table S1)] $C_c$ is a scalar (does not vary in space or time) ensemble parameter.

[The necessity of equation (6) is justified in the following manner: " To account for possible Weertman-type sliding when Coulomb drag is high . . ." A high Coulomb drag will potentially stabilize the flow and prevent the appearance of a limit cycle. Why convoluting a Weertman-type sliding law and a Coulomb one? What is the effect of using the "pure" Coulomb law (without limiting $\tau_b$ ad hoc) on the surges?] We will try to better convey the reasoning in the revised manuscript. To our knowledge, the use of the minimum of the two computed basal drags was first posed and motivated in a glaciological context by Tsai et al. [2015]. In part the motivation is: if Coulomb friction is high, Weertman-type enhanced deformation around controlling obstacles can still occur (especially given the physical separation of the Coulomb plastic deformation process within the till layer) and dominate the basal sliding. We do not follow the reviewer's reasoning on how a high Coulomb drag could stabilize the flow given the thermo-mechanical coupling and further non-linearity introduced by basal hydrology. Concretely, high Coulomb drag in Hudson Strait would not stay high as basal meltwater accumulates.

**3   GHF reconstructions:**

[Illustrating the dependence of your results to different GHF values seems completely adequate. However, as described above, spontaneous cycling ice stream behavior seems to appear only (given aside other questionable choices of your basal sliding laws) for the lower limit of available constraints.]   and [Blackwell and Richards, 2004 is just a map without any reference to a published peer-reviewed work. Shapiro and Ritzwoller, 2004 show values around 55 mW/m$^2$ with a standard deviation around 20 mW/m$^2$. The minimum value for the whole region in Pollack 1993 is higher than 45 mW/m$^2$ and the Hudson Bay and Strait regions would show a mean value closer to 60 mW/m$^2$. Goutorbe et al., 2011 showed a Hudson Bay around 40-45 mW/m$^2$ and higher values in Hudson Strait for their first method, and a regional mean around 50 mW/m$^2$ for the same region with some hot spots of more than 70 mW/m$^2$. Lucazeau, 2019 reconstructed values in Hudson Bay are around 50 mW/m$^2$ in their first method and around 40 mW/m$^2$ in the other two methods, while they show significantly higher values (around 60 mW/m$^2$) for the three methods in the Hudson Strait Area. Finally, as far as I could see, Cuesta-Valero et al., 2021 do not show any reconstruction of geothermal heat flow nor seems the intention of their paper] The map of Blackwell and Richards [2004] was published

by the American Association of Petroleum Geologists (AAPG) in 2004. While there is no peer-reviewed article associated with this map, other studies show a similarly low GHF in and around Hudson Strait and Hudson Bay with a negative northward trend [Jessop and Judge, 1971, Levy et al., 2010, Jaupart et al., 2014]. The GHF in Hudson Strait and Hudson Bay presented by Shapiro and Ritzwoller [2004] varies between 40 and 50 mW/m², not 55 mW/m². This, along with the approximate 20 mW/m² standard deviation, implies there is a 18% chance of the GHF being 20 mW/m². The mean Hudson Strait/Hudson Bay GHF in Pollack et al. [1993] is 56.1 mW/m². The values in Lucazeau [2019] are difficult to interpret towards the lower end because all values below 45 mW/m² have the same colour. The reference to Cuesta-Valero et al. [2021] was added to show the sparsity of geothermal borehole data in Hudson Strait and Hudson Bay. This can also be seen by browsing the IHFC Global Heat Flow Database [e.g., Fuchs et al., 2023]. The key point of this discussion is that the GHF for Hudson Bay is highly uncertain.

[Line 345 reads: "The exact transition point [to the binge-purge mode] depends on the parameter vector in question but generally requires a Hudson Strait/Hudson Bay GHF ave $\leq 37$ mW/m². And line 357 states: "Therefore both types of Hudson Strait ice stream surge cycling are consistent within available GHF constraints."] and [In light of the reconstructed values described in your referenced studies, this last sentence seems highly inaccurate or simply wrong. It should rather say something like: "The binge-purge mode is, under our experimental setup, only accessible if GHF¡37 mW/m² which represents the lower bound of available constraints".] The additional references above [Jessop and Judge, 1971, Levy et al., 2010, Jaupart et al., 2014] indicate that some GHF estimates in and around Hudson Strait and Hudson Bay are indeed as low as 20 mW/m² (as stated in the manuscript). We will add these references to the revised manuscript and, therefore, stand by our current statement.

**4    Conclusions of the manuscript in light of the experimental setup**

[What is the influence of $h_{\text{wb,Crit}}$ on the transition to binge-purge? Would this transition be possible if basal water is conserved?] As mentioned above, the influence of $h_{\text{wb,Crit}}$ has been examined in previous studies [Drew and Tarasov, 2023, Hank et al., 2023]. In general, the model results show only a small sensitivity to a change in $h_{\text{wb,Crit}}$.

[Your basal friction laws are designed in a way that effective pressure can be reduced several orders of magnitude (even to purely 0!) when there's enough basal water. Also, $C_b$ will accordingly reach extremely high values (equation (3)) when effective pressure drops. Do you think this is glaciologically realistic (as opposed to Leguy's parameterization, for example, or even the PISM treatment?] As outlined previously, we ensure that the effective pressure used to calculate $C_b$ never falls below $N_{\text{eff,min}} = 10$ kPa (Eq. 3). Eq. 3 further states that the multiplicative effective pressure term in the applied basal sliding coefficient is limited to a value of 10. Therefore, we consider the sliding law used within this study as glaciologically realistic or at least as realistic as other implementations.

[The regularized Coulomb law is thought to perform well, even capturing the heterogeneous character of the bed without needing to guess different friction coefficients in soft and hard beds (Joughin et al., 2019). In section 3.3 it is stated that the majority of the Coulomb runs crashed. And that it significantly increased the needed computational time for those who survived. Why is this? Could it be because of equation (6)? Or because of the additional complexity of the parameterizations over the friction coefficient and effective pressure introduced in equations (2), (3) and 4? Again, what would be the effect of letting the Coulomb law do its job, without limiting $\tau_b$?] Joughin et al. (2019) only examined Pine Island Glacier. Maier et al. [2021] also found major sectors of Greenland adhere to Weertman-type hard bed basal drag as compared to a hard bed with cavitation Mohr-Coulomb-like law. Therefore, we see no basis for the claim

*"The regularized Coulomb law is thought to perform well ..."* for our context. Since we have already partly addressed this issue in our response to the comment of referee 1, we will restate this reply here: *As the regularized Coulomb law negligibly increases basal drag beyond the order of the regularization threshold ($UV_{C,reg} = 20$ m/yr), we expect it to be much more unstable than the Weertman law according to CFL constraints. This is compounded by the schoofing grounding-line flux iteration in the SSA solution. It should also be noted that the GSM SSA solution imposes an upper bound of 40 km/yr on SSA ice velocities for this configuration. We suspect that this is higher than most other models. The imposition of this upper bound is itself another non-linearity in the solution that can contribute to both instability (as adding non-linearities will generally decrease convergence of iterative solutions) and stability (by limiting ice velocities).*

[Runs that are subjected to severe numeric problems can often illustrate that the physical problem is not well-posed. Can you discard this is what is happening here? How many of your reference ensemble runs crashed?] Ice stream surge cycling in itself is a highly non-linear physical mechanism. Therefore, we expect some runs to crash, especially since we are also probing a large parameter space. However, one of the criteria for the runs in the reference ensemble was a successful completion (no reference ensemble runs crashed).

Since we slightly vary the model configuration for each experiment, we can not guarantee a successful completion for all parameter vectors for all experiments. However, only runs that did not crash are included in the analysis. We will clarify this in the revised manuscript.

[Looking at figure 5, the reader might notice that the upper limit of the basal velocities bar is 40,000 m/yr. It is not known whether the simulated velocities ever approach that value, because the manuscript does not contain the time series of velocities over a whole ice stream cycle, but I imagine that they are not far from the upper limit of the depicted scale. From figure 5 it is clear that they reach up to 20,000 m/yr for several hundred of kilometers upstream of the grounding line. Current observed surface velocities in Antarctica do not go beyond 1,500 m/yr downstream of the grounding line. Do you think your simulated velocities are realistic?] The upper limit of the basal velocities colorbar is set to 40 km/yr because it is the upper bound on the SSA ice velocities imposed in the GSM. While observed velocities of surging glaciers can reach several hundreds of meters per day for short periods [K.M. Cuffey and W.S.B. Paterson., 2010, , e.g., 100 m/d = 36.5 km/yr], we do not have a clear present-day analog for Hudson Strait ice stream. Can the referee confidently ascert such a velocity was never reached in the past? We can not. However, we agree it is important to document maximum velocities in the experiments and will add that documentation to the revised manuscript.

[All in all, unless the authors show otherwise, their experimental setup is constructed in a way that the relevant-for-the-problem parameter space (and thus the phase space of the associated physical problem) is very narrow and situated in a very specific region.] We are unsure what the referee is referring to here. If the referee is arguing about the limited parameter space, given that all previous studies for this context outside of our own research group have used a much more limited parameter space, on what basis should they have been published? To reiterate, we sample 52 input parameters over wide parameter ranges based on extensive history-matching results. While others might not have such a large explicit parameter space, given system uncertainties, it is still implicitly there if one is actually trying to make meaningful inferences about Hudson Strait ice stream surge cycling. Additionally, we included the *bounding experiments* (or end member scenarios, Sec. 2.5.3) to bound the effects of the ocean forcing experiments and increase confidence in our model results. Furthermore, our conclusions generally hold for near-continuous ice streaming with occasional shutdowns and subsequent surge onset overshoot ($GHF_{ave} > 37\,\mathrm{mW\,m^{-2}}$) and the classic binge-purge surge cycling ($GHF_{ave} \leq 37\,\mathrm{mW\,m^{-2}}$). As we examine various sliding laws and as it has been previously shown that in a Hudson Strait ice stream context *sub-glacial hydrology matters but the process details mostly do not* [Hank et al., 2023, Drew and Tarasov, 2023], we do not see why the relevant-for-the-problem parameter space would be very narrow and situated in a very specific region, especially in comparison

to previous studies.

[In the absence of a rigorous illustration of the influence of the particular assumptions of the experimental setup (basal sliding laws, basal water, effective pressure, dependence of the friction coefficient on the nature and temperature of the bed, numerical issues. . . ) on the mechanisms favouring spontaneous ice stream cycling, the conclusions reached here are based on the analysis of a very particular experimental setup. (Which in my opinion is, by construction, prompt to oscillate in a questionable physical manner).] If the above is the standard required to credibly examine Hudson Strait ice surging, then no paper to date on the subject should have been published. Our aim is to bound the possible behaviour within the full system complexity of the North American ice sheet. We, therefore, intentionally use an ensemble approach to test the relative role and robustness of key surge cycling mechanisms/hypotheses given the uncertainties in climate forcing, basal drag, basal hydrology, and such. These are uncertainties that previous studies have largely ignored. Based on the reviewer's reasoning, it is fine to ignore uncertainties, but if you try to address them, then you have to isolate and analyze each contribution. We strongly disagree.

As mentioned previously, the effects of basal water, effective pressure, dependence of the friction coefficient on the nature and temperature of the bed, and model numerics have been extensively studied by Hank et al. [2023] and Drew and Tarasov [2022]. Although limited by the number of successful runs, the effect of different sliding laws is examined in this study.

[And this leads me to my next point: What happens if you run the Antarctic ice sheet under such a configuration?] and [To answer this question, you would need to make some assumptions concerning the bed type. One reasonable choice would be to assume the presence of soft sediments in every Antarctic marine sector. Do you expect that such a simulation would give reasonable results?] and [In my modelling experience, this is unlikely.] and [GRISLI and Yelmo can also show spontaneous ice stream oscillations under extreme conditions (e.g, limiting basal water and having very distinct spatial basal frictions). But when you apply such a physics to Antarctica, it becomes very difficult to approach observed velocity values, and furthermore some ice streams become very noisy and others dramatically oscillate in a manner that so far is not observed in Antarctica.] and [Therefore, I believe the current experimental setup extremely conditions the current conclusions of the paper.] An initial paper examining GSM history-matching results for Antarctica is currently under open review in egusphere [Lecavalier and Tarasov, 2024]. The ensemble parameters in the NROY (Not Ruled Out Yet) set for Antarctica bracket, on both ends, the soft and hard bed sliding parameters for our 20 base parameter vectors used here. Moreover, the Antarctic configuration assumes soft sediments in marine sectors (at least when marine after full unloading) with possible pinning points according to available DEMs. Before submitting revisions, we will check if the Antarctic simulations have the unstable ice streaming the referee describes. The caveat is that Antarctica was history-matched before the inclusion of basal hydrology. Future work with Antarctica will include the updated configuration with basal hydrology. What the reviewer may not be aware of is that the GSM configuration is informed by modelling of all last glacial cycle major ice sheets as well as major ice caps (or minor ice sheets) such as Icelandic and Patagonian.

[The introduction section nicely ends with the presentation of very relevant questions regarding the different mechanisms exciting the Hudson Strait ice stream variability during HEs. The discussion section also nicely re-addresses those questions (see some minor comments below) in the context of the new results shown in the manuscript. The conclusions, however, do not fairly summarize the findings and are biased towards the spontaneous ice-stream cycling mechanism.] and [After everything that is shown in the paper, and given all the limitations of the experimental setup pointed out in this review, concluding that "Based on our results, Hudson Strait ice stream surge cycling is the most likely Heinrich Event mechanism. . . " seems unjustified.] and [The questions Q2, Q3 and Q4 of your discussion are compatible with each other. When answering Q3 you state that your maximum simulated ice shelf (leaving aside the one with inhibited calving) is close to the minimum required by Hulbe 2004 to explain

the IRD signal. So, can you discard that a combination of Q2, Q3 and Q4 produces both the icebergs from breaking the ice shelves and the increase in the flux at the grounding line necessary for explaining HEs?] We thank the referee for raising this point and will discuss a combined mechanism of Q2, Q3, and Q4 (sub-surface ocean warming leads to the breakup of fringing ice shelves and, consequently, a sudden reduction of the buttressing effect) in the revised conclusions. Since the mean Labrador Sea ice shelf volume is significantly smaller than the maximum volume (Fig. 1; and, therefore, the minimum required by Hulbe et al. [2004]), and since the fringing ice shelves in front of Hudson Strait provide only minor buttressing, our results indicate that even when considering a combined mechanism, HE can not be explained without Hudson Strait ice stream surge cycling. However, as stated in the conclusions, even the small changes associated with sub-surface ocean warming *can affect the timing of surges and provide a means to synchronize HEs with the coldest phases of the Bond cycles.*

[Figure]

Figure 1: Labrador Sea ice shelf volume in the *Labrador Sea ice shelf area* outlined in Fig. 2. The thick line represents the mean of the 20 run ensemble. The shaded area marks the minimum and maximum of the ensemble.

[Under a questionable and very particular experimental setup, your simulations show that you have to go to the very low bound of Geothermal heat forcing for the ice stream surge cycling to emerge. But, even assuming that a binge-purge like oscillation is a good candidate to explain HEs, there is a remaining puzzling question that you completely ignored. Why are HEs happening at the middle of the cold NH phases, or stadials? Some synchronization mechanisms have been explored in the literature to potentially answer this question (.e.g. the work of Calov and Ganopolski and Shanwell et al., 2024) but you do not inform whether these synchronization mechanisms are captured in your simulations. So, under your experimental setup, the question of HEs occurrence during the surface NH cold phases is still of concern for your conclusions (see also Barker et al., 2015; Nature). Therefore, the authors seem to be evaluating certain hypotheses with a level of criticism and rigour not present in the case of their announced more likely mechanism.] While a low GHF (but as we have argued above, not extremally low given available literature) is essential for the classic binge-purge surge mechanism, it is not a requirement for the second ice stream surge cycling mode discussed in this study (near-continuous ice streaming with occasional shutdowns and subsequent surge onset overshoot). As stated in the manuscript: *ocean forcings can affect the timing of surges and provide a means to synchronize HEs with the coldest phases of the Bond cycles.* However, we will expand on this

in the revised draft, including a comparison with the recently published results of Schannwell et al. [2024, not published at the time of manuscript submission]. Additionally, we will clarify that the issue of HE synchronization with the coldest phases of the Bond cycles (though not always the case, e.g., HE1) is the least resolved issue from our experiments.

[You discard the ice-shelf breakup related hypothesis (Q2) because your simulated ice shelves are not big enough to significantly buttress the Hudson strait ice stream and because the paleo reconstructions of the Labrador Sea conditions during MIS3 do not seem compatible wit the existence of a big ice shelf in the Area. This is a fair criticism, but I would like to point out here that in GRISLI and Yelmo (both codes are available; the first upon request to the former developer, Catherine Ritz or myself, and the second here: https://github.com/palma-ice/yelmo) the emergence of a big Labrador Sea ice shelf is a pretty natural characteristic provided oceanic temperatures are low enough. Even with a relatively warm ocean, Yelmo simulates the existence of very developed fringing ice shelves around the mouth of Hudson Strait (Moreno_parada et al., 2023; The Cryosphere).] The paleoceanographic constraints and oceanic temperatures being low enough are the crux. If temperatures are low enough, it is not surprising to have Baffin Bay covered by an ice shelf along with the confined part of the Labrador Sea, especially if one considers the comparative area of say the inferred LGM Ronne-Filchner ice shelf. However, is it reasonable to assume that conditions over the LGM Labrador Sea were similar to those of LGM Ronne-Filchner? The paleoceanographic constraints appear to rule this out. Furthermore, the mouth of Hudson Strait is at the edge of Labrador Sea confinement. As stated in our submission, the base simulations do have fringing ice shelves (including at the mouth of Hudson Strait); the critical issue is whether these are large enough to exert enough back stress to affect Hudson Strait streaming.

For our revised submission, we will review/document the Labrador Sea temperature range (based on TraCE deglacial simulation run with the Community Climate System Model Version 3 [CCSM3, Liu et al., 2009]) used in the GSM for these experiments. Depending on this range, we may test further cooling to at least document what is needed to get a pan Northern Labrador Sea ice shelf.

[I believe that referencing our paper (Alvarez-Solas et al., 2013) in lines 95 and 548 in a technical and discussing context respectively, without having it cited in the introduction, when the different hypotheses are explained, is academically incorrect. Note, we also explored the conceptual idea of triggering HEs through changes in the oceanic temperature (Alvarez-Solas et al., 2010; NatGeo), and the effects of an ice-shelf breakup during H1 (Alvarez-Solas et al, 2011, ClimPast), coetaneous with Marcott et al., 2011] We thank the referee for raising this issue and will add the references to the introduction.

**References**

D.D. Blackwell and M. Richards. Geothermal map of north america. AAPG Map, scale 1:6,500,000, Product Code 423, 2004. URL https://www.smu.edu/-/media/Site/Dedman/Academics/Programs/Geothermal-Lab/Graphics/Geothermal_MapNA_7x10in.gif.

Francisco José Cuesta-Valero, Almudena García-García, Hugo Beltrami, J. Fidel González-Rouco, and Elena García-Bustamante. Long-term global ground heat flux and continental heat storage from geothermal data. *Climate of the Past*, 17(1):451–468, 2021. ISSN 18149332. doi: 10.5194/cp-17-451-2021.

M. Drew and L. Tarasov. Surging of a hudson strait scale ice stream: Subglacial hydrology matters but the process details don't. *The Cryosphere Discussions*, 2022:1–41, 2022. doi: 10.5194/tc-2022-226. URL https://tc.copernicus.org/preprints/tc-2022-226/.

M. Drew and L. Tarasov. Surging of a hudson strait-scale ice stream: subglacial hydrology matters but the process details mostly do not. *The Cryosphere*, 17(12):5391–5415, 2023. doi: 10.5194/tc-17-5391-2023. URL https://tc.copernicus.org/articles/17/5391/2023/.

Gwenn E. Flowers and Garry K. C. Clarke. A multicomponent coupled model of glacier hydrology 1. Theory and synthetic examples. *Journal of Geophysical Research: Solid Earth*, 107(B11), 2002. ISSN 0148-0227. doi: 10.1029/2001jb001122.

Gwenn Elizabeth Flowers. *A multicomponent coupled model of glacier hydrology*. PhD thesis, University of British Columbia, 2000. URL https://open.library.ubc.ca/collections/ubctheses/831/items/1.0053158.

Sven Fuchs, Ben Norden, Florian Neumann, Norbert Kaul, Akiko Tanaka, Ilmo T. Kukkonen, Christophe Pascal, Rodolfo Christiansen, Gianluca Gola, Jan Šafanda, Orlando Miguel Espinoza-Ojeda, Ignacio Marzan, Ladislaus Rybach, Elif Balkan-Pazvantoğlu, Elsa Cristina Ramalho, Petr Dĕdeček, Raquel Negrete-Aranda, Niels Balling, Jeffrey Poort, Yibo Wang, Argo Jõeleht, Dušan Rajver, Xiang Gao, Shaowen Liu, Robert Harris, Maria Richards, Sandra McLaren, Paolo Chiozzi, Jeffrey Nunn, Mazlan Madon, Graeme Beardsmore, Rob Funnell, Helmut Duerrast, Samuel Jennings, Kirsten Elger, Cristina Pauselli, and Massimo Verdoya. Quality-assurance of heat-flow data: The new structure and evaluation scheme of the IHFC Global Heat Flow Database. *Tectonophysics*, 863(June), 2023. ISSN 00401951. doi: 10.1016/j.tecto.2023.229976.

K. Hank, L. Tarasov, and E. Mantelli. Modeling sensitivities of thermally and hydraulically driven ice stream surge cycling. *Geoscientific Model Development*, 16(19):5627–5652, 2023. doi: 10.5194/gmd-16-5627-2023. URL https://gmd.copernicus.org/articles/16/5627/2023/.

Christina L. Hulbe, Douglas R. MacAyeal, George H. Denton, Johan Kleman, and Thomas V. Lowell. Catastrophic ice shelf breakup as the source of Heinrich event icebergs. *Paleoceanography*, 19(1):n/a–n/a, 2004. ISSN 0883-8305. doi: 10.1029/2003pa000890.

C. Jaupart, J. C. Mareschal, H. Bouquerel, and C. Phaneuf. The building and stabilization of an Archean Craton in the Superior Province, Canada, from a heat flow perspective. *Journal of Geophysical Research: Solid Earth*, 119(12):9130–9155, 2014. ISSN 21699356. doi: 10.1002/2014JB011018.

Alan M. Jessop and Alan S. Judge. Five Measurements of Heat Flow in Southern Canada. *Canadian Journal of Earth Sciences*, 8 (6):711–716, 1971. ISSN 0008-4077. doi: 10.1139/e71-069.

K.M. Cuffey and W.S.B. Paterson. *The Physics of Glaciers*. Butterworth-Heinemann/Elsevier, Burlington, MA, 4th edition, 2010. ISBN 9780123694614.

B. S. Lecavalier and L. Tarasov. A history-matching analysis of the antarctic ice sheet since the last interglacial – part 1: Ice sheet evolution. *EGUsphere*, 2024:1–38, 2024. doi: 10.5194/egusphere-2024-1291. URL https://egusphere.copernicus.org/preprints/2024/egusphere-2024-1291/.

F. Levy, C. Jaupart, J. C. Mareschal, G. Bienfait, and A. Limare. Low heat flux and large variations of lithospheric thickness in the Canadian Shield. *Journal of Geophysical Research: Solid Earth*, 115(6):1–23, 2010. ISSN 21699356. doi: 10.1029/2009JB006470.

Z. Liu, B. L. Otto-Bliesner, F. He, E. C. Brady, R. Tomas, P. U. Clark, A. E. Carlson, J. Lynch-Stieglitz, W. Curry, E. Brook, and et al. Transient Simulation of Last Deglaciation with a New Mechanism for Bolling-Allerod Warming. *Science*, 325(5938): 310314, Jul 2009. ISSN 1095-9203. doi: 10.1126/science.1171041. URL http://dx.doi.org/10.1126/science.1171041.

F. Lucazeau. Analysis and Mapping of an Updated Terrestrial Heat Flow Data Set. *Geochemistry, Geophysics, Geosystems*, 20 (8):4001–4024, 2019. ISSN 15252027. doi: 10.1029/2019GC008389.

Nathan Maier, Florent Gimbert, Fabien Gillet-Chaulet, and Adrien Gilbert. Basal traction mainly dictated by hard-bed physics over grounded regions of Greenland. *Cryosphere*, 15(3):1435–1451, 2021. ISSN 19940424. doi: 10.5194/tc-15-1435-2021.

Henry N Pollack, Suzanne J Hurter, and Jeffrey R Johnson. Heat flow from the Earth's interior: Analysis of the global data set. *Reviews of Geophysics*, 31(3):267–280, 1993. doi: https://doi.org/10.1029/93RG01249. URL https://agupubs.onlinelibrary.wiley.com/doi/abs/10.1029/93RG01249.

Clemens Schannwell, Uwe Mikolajewicz, Marie Luise Kapsch, and Florian Ziemen. A mechanism for reconciling the synchronisation of Heinrich events and Dansgaard-Oeschger cycles. *Nature Communications*, 15(1):1–8, 2024. ISSN 20411723. doi: 10.1038/s41467-024-47141-7.

Nikolai M. Shapiro and Michael H. Ritzwoller. Inferring surface heat flux distributions guided by a global seismic model: Particular application to Antarctica. *Earth and Planetary Science Letters*, 223(1-2):213–224, 2004. ISSN 0012821X. doi: 10.1016/j.epsl.2004.04.011.

Victor C. Tsai, Andrew L. Stewart, and Andrew F. Thompson. Marine ice-sheet profiles and stability under Coulomb basal conditions. *Journal of Glaciology*, 61(226):205–215, 2015. ISSN 00221430. doi: 10.3189/2015JoG14J221.

---

## Author Response (AR1)

**Author's response to Anonymous Referee 1 Comment 1**

September 22, 2024

[Kevin Hank and Lev Tarasov present a comprehensive study on the potential causes of Heinrich events. Their aim is to investigate several hypotheses which have been proposed in the literature, mainly: the effect of geothermal heat flux and basal drag, buttressing effect, subsurface melt and GIA. For this, they use the glacial system model (GSM) and for a set of experiments (20) they conduct a test of sensitivity experiments]

[While I believe the manuscript holds significance and aligns with the focus of Climate of the Past, I suggest some restructuring prior to publication. While the introduction effectively outlines the study's goals, the subsequent sections lack cohesion, making it challenging for readers to follow. There are too many figures which are referenced several times without following a chronological order necessarily which makes it easy to lose the guiding thread. Additionally, the interpretation of the experiments may be difficult to follow.]

[My biggest concern is related to the investigated ensemble parameters. I think the choice of parameter space feels a bit arbitrary and unnecessarily extensive for the paper's aims. I want to emphasize that I recognize the value of this manuscript but stress the importance of enhancing its readability for a broader audience. Below you can find my main concerns.]

We thank the referee for their constructive comments. A point-by-point reply is reported below, with referee comments in orange and our replies in black. We agree with the specific referee comments not listed here and have revised the manuscript accordingly. For revisions too substantial to be explicitly stated here (e.g., restructuring of the methods section), we added a brief description and refer the interested reader to the Author's track changes for the detailed changes.

To clarify, we use the GSM with a set of 20 parameter vectors (not experiments) for the base ensemble and conduct an ensemble-based sensitivity analysis for 60 experiments. As such, each experiment is carried out with a 20-member ensemble.

As suggested by the referee, we merged plots with similar content. Where possible, the figures were rearranged to follow a chronological reference pattern. However, breaking this pattern was required in some instances (e.g., when outlining all sensitivity experiments in the new Tab. 1 or when referring to the end-member scenarios in the supplement).

The choice of ensemble parameters and parameter ranges within this study is based on extensive model development and testing [Tarasov and Peltier, 1999, 2007, Pollard and DeConto, 2012, Drew and Tarasov, 2023, Hank et al., 2023, Tarasov et al., 2024]. While the parameter space is more extensive than in previous Heinrich Event modeling studies, the authors argue that it is still insufficient to fully capture the uncertainties involved. However, computational expenses restrict our ability to further increase the parameter space.

**1 General Comment**

**1.1 Main manuscript**

[Parameter choice - I do not understand how you choose your 20 parameter vectors. First you run ~15000 simulations, from which you consider ~200 simulations as realistic based on

your sieves. Then you redo these 200 simulations at higher spatial resolution and apply new sieves based on IRD layers. From those you hand pick 20 simulations. Have I understood it correctly?] That is generally correct. However, we refer to the 200 simulations as *Not Ruled Out Yet (NROY)* rather than *realistic*. The latter is imprecise, and the former better communicates the contingent nature of our inferences about past earth system evolution, as well as a reminder that explicit criteria are being used for this designation. Furthermore, as stated in the text, the 20 runs were hand-picked to provide two high-variance (in parameter values and run metrics) subgroups: one subgroup of 10 for which all runs have more than 2 surges (at the higher resolution), and one for which this is not the case for the higher resolution run but is the case for the corresponding lower resolution run (i.e., with the same parameter vector).

**Changes:** *The remaining* $\sim 200$ *runs are re-submitted ...* **to** *The* $\sim 200$ *not ruled out yet (NROY) runs are re-submitted ...*

[Parameter choice - I am a bit skeptical about some of your parameters. A lot of your parameters are related to climatologies (for example: global LGM temperature scale factor, desert-elevation exponent, temporal Empirical Orthogonal Function weight 1), but you are not assessing the role of different climatologies in your study. Since this is not the focus of your manuscript I think you need to apply the same climatologies to all of your experiments. If not, the occurrence of surges in your experiment could be caused by different climatologies rather than ice dynamics or other forcings. This would be also very beneficial for the readers since you introduce a lot of parameters (52 in Table 1) which you do not explain and are very technical.] Relying on only one climatology neglects the uncertainties associated with the climatic forcings during the last glacial cycle. Otherwise, how does one justify arbitrarily chosen climate forcing given the uncertainty in glacial cycle climate? And how does one know if their results have any validity for a different climate history? To (albeit partly) account for these uncertainties, our analysis is based on ensemble results. The key idea behind this approach is to identify physical processes that show a significant effect across simulations with different ensemble parameters. This both increases confidence in our results and reduces the possibility that an observed modeling response is only due to, e.g., the chosen climate forcing.

**Changes:** We restructured and revised the Methods section to clarify that we run all experiments with 20 different climatologies (20 parameter vectors). We now clearly outline that we directly compare the model results of, e.g., parameter vector 0 for experiment A and B but never compare the results of, e.g., parameter vector 0 experiment A to parameter vector 1 experiment B.

[Parameter choice - I get the feeling that many of your selected parameters are the same as those of Tarasov et al. in preparation describing the model GSM. Though investigating such a large parameter space makes sense for a description paper, I do not think it is intended for this manuscript.] The reviewer does not seem to appreciate that the ensemble parameters in the GSM are effectively in all paleo ice sheet models not coupled to an EMIC more advanced than the EBM (energy balance climate model) in the GSM. Most models implicitly set most of the parameters in the model to a fixed value, but that does not make the uncertainty associated with that parameter go away. For those ISMs that are coupled to a moderate to advanced complexity EMIC, there will be plenty of new EMIC parameters to replace the climate-forcing parameters in the GSM.

While investigating the parameter space is not the main aim of this paper, Table S1 intends to show the parameter ranges used within this study.

**Changes:** The revised Methods section clarifies the benefits of using a parameter ensemble over a single parameter vector and outlines the purpose of Table S1.

[Reference state - This point goes a bit in line with my previous comment. Many of your plots only show one vector parameter but it's not always the same vector parameter. To me this feels confusing, I would prefer to have a reference vector over which you change conditions rather than different states. If you are trying to assess the effect of different boundary conditions, friction laws or oceanic forcings it would be more useful to have one reference state over which

you change conditions.] While we agree that using the same parameter vector might be less confusing, it is not straightforward to determine one base vector. Different parameter vectors respond differently to changes in model configuration. Generally, we plot the parameter vector that best resembles the ensemble response for a specific change in model configuration. However, for every time series (single vector) plot, we also show an ensemble mean plot. The single vector plots are intended to give a physically self-consistent example (which, e.g., an ensemble mean does not provide). On the other hand, the ensemble means and standard deviations provide more robust statistical results for the ensemble response to the change in model configuration. Therefore, we prefer to keep the different parameter vectors in the plots.

[Hydrology model - Based on Figure 4 and Figure 7 it seems that basal melt below grounded ice plays a major role in your surges. You need to explain how you compute basal melt for grounded ice points and your hydrology model. Since it is a local hydrology model and you state that you saturate your water thickness at 10 meters i assume that you do not conserve mass, right? How does your water thickness affect your surges? I guess it will play a major role in your surges and I think it is necessary to investigate that parameter.] We agree that the inclusion of a brief description of the basal melt rate (which is simply from applying conservation of energy to the ice sheet) is essential. However, we are confused by the reviewer's question. The basal meltwater from the grounded ice sheet is fed directly into the basal hydrology model, and we cannot envision what else could be done. Furthermore, the GSM is configured as all ice sheet models not fully coupled to a dynamical ocean model: all meltwater that is not refrozen and not stored subglacially leaves the ice sheet, and therefore mass is not conserved. The GSM internally does conserve mass: integrated: accumulation − melt + refreezing − calving = change in ice mass.

**Changes:** Added *The local basal hydrology model nominally sets the time derivative of* $h_{wb}$ *to the difference between the basal melt rate* $M_b$ *(from conservation of energy at the ice sheet base) and a constant bed drainage rate* $R_{b,drain}$. Also added *Ice thermodynamics and basal melt for grounded ice are computed via an energy-conserving finite-volume solver.*

[Additional - What does your LGM ice sheet look like? How does it compare to other studies? I am also missing your forcing index. You could add it in your time series plots.] The LGM ice sheet configuration differs between parameter vectors. **Changes:** A plot showing the ice volume of all parameter vectors when run in the reference configuration (Fig. S5), along with MIS 3 (Fig. S6) and LGM example timeslice map plots (Fig. S7), were added to the revised draft. However, a detailed comparison of the LGM ice sheet configuration to other studies and the details of the glacial forcing index are not the primary goal of this study and will be addressed in manuscripts currently in preparation.

**1.2 Supplementary Material**

[41 figures in the Supplementary Material is way too much and makes it difficult to follow the paper. These figures are referenced too many times and cuts the flow for the reader. Please consider reducing the amount of figures with those which are strictly necessary for your article. For example, you could merge figures S2-S6.] While we agree that the Supplementary Material is quite long, it provides valuable information for readers interested in specific details of the study.

**Changes:** The old Fig. S2-S6 have been merged into the new Fig. S4.

Added *The supplementary material referenced throughout this text (indicated by a capital S) provides additional information to support the corresponding claim but is not essential to the understanding of this study.*

**2 Technical comments**

[Section 2.3 - Please describe the equation of your Weertman-type power law. You state that your parameter C_warm depends on other parameters such as C_rmu and C_fslid but you do not give further detail. Either you remove that sentence or you explain how C_warm depends on those parameters.] **Changes:** Added the value for $C_{\text{froz}} = 2 \cdot 10^{-4} \text{ m yr}^{-1} \left(5 \cdot 10^{-6} \text{ Pa}^{-1}\right)^4$ and the following description of $C_{\text{warm}}$: *$C_{\text{warm}}$ is the fully warm-based sliding coefficient with dependence on bed properties:*

$$C_{\text{warm,soft}} = C_{\text{rmu}} \cdot \min\left[1; \max\left[0.2; \frac{F_{\text{sub,till}}}{0.01 \ \sigma_{\text{hb}}}\right]\right] \tag{1a}$$

$$C_{\text{warm,hard}} = C_{\text{slid}} \cdot \min\left[1; \max\left[0.1; \frac{F_{\text{sub,slid}}}{0.01 \ \sigma_{\text{hb}}}\right]\right] \cdot (1 + 20 \ F_{\text{sed}}), \tag{1b}$$

*Here $C_{\text{rmu}}$ and $C_{\text{slid}}$ are the ensemble parameters for the Weertman soft- and hard-bed sliding coefficients, respectively (Table S1). The ensemble parameters $F_{\text{sub,till}}$ and $F_{\text{sub,slid}}$ (Table S1) impose the Weertman basal drag dependencies on the subgrid standard deviation of the bed elevation $\sigma_{\text{hb}}$ and $F_{\text{sed}}$ is the subgrid fraction of soft bed cover.*

[Section 2.3 - Do you apply any basal-stress scaling at the grounding line? If you are using a coarse resolution, scaling basal stress at the grounding line has shown to help to simulate grounding line migration in agreement with high resolution. Actually, this could help to simulate more surges potentially. Do you apply any melt at the grounding line?] The GSM uses the Schoof [2007] grounding line flux condition as implemented in [Pollard and DeConto, 2012]. The authors only recently became aware of issues around this approach for complex 2D geometries likely of most consequence for Antarctica [Reese et al., 2018], and the revised validated treatment [Pollard and Deconto, 2020] has subsequently been implemented in the GSM.

Submarine basal melt is not applied to the grounding line grid cell. Sub-marine face melt is applied at the grounding line if it is a tidewater outlet with an exposed (i.e., no ice shelf) calving face.

**Changes:** We tested sensitivities for different grounding line treatments on Hudson Strait surge characteristics and documented this in the revised draft (new Fig. S2 and S3). Using the revised validated treatment of Pollard and Deconto [2020] instead of the Schoof [2007] grounding line flux condition has no significant effect on the surge characteristics.

[Section 3.3 - I am surprised that using a (regularized) Coulomb law leads to model crashes and a model run time much slower than a Weertman law. Do you have an explanation for this?] As the regularized Coulomb law negligibly increases basal drag beyond the order of the regularization threshold ($UV_{\text{C,reg}} = 20$ m/yr), the regularized Coulomb law lacks the increase in basal drag with basal velocity intrinsic to Weertman sliding and therefore will be more numerically unstable. This is compounded by the schoofing grounding-line flux iteration in the SSA solution.

It should also be noted that the GSM SSA solution imposes an upper bound of 40 km/yr on SSA ice velocities for this configuration. We suspect that this is higher than most other models. The imposition of this upper bound is itself another non-linearity in the solution that can contribute to both instability (as adding non-linearities will generally decrease convergence of iterative solutions) and stability (by limiting ice velocities).

**Changes:** Added *According to CFL constraints, we expect even the regularized Coulomb law (Eq. 6b) to be more unstable than the Weertman law, because it negligibly increases basal drag for basal velocities beyond the order of the regularization threshold $UV_{\text{C,reg}} = 20$ m yr$^{-1}$. This is compounded by the grounding-line flux iteration in the SSA solution.*

[Figure 4 - Please use another naming for your "surges" since notation S9 can be confused between you Figure S9 or your surge S9. How is it possible that you obtain the highest buttressing value when your ice shelf is small?] We thank the referee for highlighting this issue.

**Changes:** The naming of the surges was adjusted from *S1, S2, ...* to *P1, P2, ...* .

The plotted buttressing time series is given as a fraction of the grounding line longitudinal stress. Therefore, a small ice shelf can lead to large buttressing when the grounding line

longitudinal stress is small.

**Changes:** The y-label in the plot was updated from *Mean buttressing along the Hudson Strait grounding line [%]* to *Buttressed fraction of the Hudson Strait* $GL_{lon}$ *stress [%]*.

[Figure 9 - You do not reference Figure 9 in your manuscript. Do you need that figure?] Fig. 9 is already referenced in *While a change in surge mode also occurs for a lower GHF, using the regularized Coulomb sliding law leads to more frequent binge-purge surges (Table 2, Fig. 8 and 9).*

[Figure 10 - You do not need Figure 10 since you are already showing your reference state in Figure 12.] While we are showing the reference state in Fig. 12, it is not exactly the same as in Fig. 10 (new Fig. 11). As indicated in the captions of these figures, Fig. 12 shows the mean and min/max range of 18 parameter vectors for all setups shown (parameter vectors 8 and 15 crashed (automatically terminated once the time step size is reduced below the set minimum of 0.015625 yr) in the comparison setups), whereas Fig. 10 shows the mean and min/max range of all 20 parameter vectors. Due to the missing 2 parameter vectors in Fig. 12 and the larger and clearer visualization of the reference setup statistics in Fig. 10, we prefer to keep both figures. However, to clarify the differences, we adjusted the caption of Fig. 12.

**Changes:** *The thick line represents the mean of* 18 *runs (the runs for parameter vectors* 8 *and* 15 *crashed in both comparison setups and were not included).* **to** *The thick line represents the mean of* 18 *runs (the runs for parameter vectors* 8 *and* 15 *crashed in both comparison setups and were not included for all setups shown).*

**References**

M. Drew and L. Tarasov. Surging of a hudson strait-scale ice stream: subglacial hydrology matters but the process details mostly do not. *The Cryosphere*, 17(12):5391–5415, 2023. doi: 10.5194/tc-17-5391-2023. URL `https://tc.copernicus.org/articles/17/5391/2023/`.

K. Hank, L. Tarasov, and E. Mantelli. Modeling sensitivities of thermally and hydraulically driven ice stream surge cycling. *Geoscientific Model Development*, 16(19):5627–5652, 2023. doi: 10.5194/gmd-16-5627-2023. URL `https://gmd.copernicus.org/articles/16/5627/2023/`.

D. Pollard and R. M. DeConto. Description of a hybrid ice sheet-shelf model, and application to Antarctica. *Geoscientific Model Development*, 5(5):1273–1295, 2012. ISSN 1991959X. doi: 10.5194/gmd-5-1273-2012.

David Pollard and Robert M. Deconto. Improvements in one-dimensional grounding-line parameterizations in an ice-sheet model with lateral variations (PSUICE3D v2.1). *Geoscientific Model Development*, 13(12):6481–6500, 2020. ISSN 19919603. doi: 10.5194/gmd-13-6481-2020.

Ronja Reese, Ricarda Winkelmann, and G. Hilmar Gudmundsson. Grounding-line flux formula applied as a flux condition in numerical simulations fails for buttressed Antarctic ice streams. *Cryosphere*, 12(10):3229–3242, 2018. ISSN 19940424. doi: 10.5194/tc-12-3229-2018.

Christian Schoof. Marine ice-sheet dynamics. Part 1. The case of rapid sliding. *Journal of Fluid Mechanics*, 573:27–55, 2007. ISSN 00221120. doi: 10.1017/S0022112006003570.

L. Tarasov and W. Richard Peltier. Impact of thermomechanical ice sheet coupling on a model of the 100 kyr ice age cycle. *Journal of Geophysical Research Atmospheres*, 104(D8):9517–9545, 1999. ISSN 01480227. doi: 10.1029/1998JD200120.

Lev Tarasov and W. R. Peltier. Coevolution of continental ice cover and permafrost extent over the last glacial-interglacial cycle in North America. *Journal of Geophysical Research: Earth Surface*, 112(2):1–13, 2007. ISSN 21699011. doi: 10.1029/2006JF000661.

Lev Tarasov, Benoit S. Lecavalier, Kevin Hank, and David Pollard. The glacial systems model (GSM). 2024. URL `https://www.physics.mun.ca/~lev/GSM.pdf`.

**Author's response to Clemens Schannwell's Comment 1**

September 22, 2024

**1 General Comment**

[The manuscript by Hank & Tarasov presents a suite of simulations that aims to investigate the sensitivity of the Glacial Systems Model (GSM) to processes that have previously been shown to have an influence on Heinrich Event characteristics in other models. The novelty in comparison to previous studies is that the authors use transient forcing and present an ensemble that covers a much larger parameter space. Based on this, they confirm earlier findings from e.g. Mann et al. [2021], Schannwell et al. [2023] that GSM also exhibits two different states - a streaming and cycling surging state. Moreover, they show that in their model, geothermal heat is a major controlling factor while ice shelves and their buttressing matter very little.]

[Overall, the topic of the paper is interesting and I believe that the science behind it is sound. However, in its present form the reader is drowned in the sheer amount of simulations that are presented in a rather unstructured fashion. The lack of structure in the paper and particularly in the methods sections leaves the reader confused as to what exactly the authors did and makes it challenging to find a common thread through the manuscript as well as evaluate the scientific results. Therefore, I do not think the paper is ready for publication in its current form and a substantial restructuring and part-rewrite is required. I recommend the authors take into account my comments listed below. I hope the authors find my comments helpful.]

We thank the referee Clemens Schannwell for their constructive comments. A point-by-point reply is reported below, with referee comments in orange and our replies in black. We agree with the specific referee comments not listed here and revised the manuscript accordingly. For revisions too substantial to be explicitly stated here (e.g., restructuring of the methods section), we added a brief description and refer the interested reader to the Author's track changes for the detailed changes.

We restructured the manuscript according to the referee's helpful suggestions, including a short roadmap at the end of the introduction and a table summarizing the reference setup and all experiments.

**2 Specific comments**

[The entire methods sections felt really incoherent almost as if sections were being added as new ideas for sensitivity simulations were designed. By the time I got to the results, there were so many simulations and sub-ensembles that I was utterly confused what was done in the end and what the different simulations were actually referring to. The absence of an experiment table or a flowchart of how the different sub-ensembles were generated just confounded my confusion. My suggestion is to remove section 2.1. "Modelling approach" and really start with the model description (your current section 2.3). In there, I would like to see what type of model GSM is! I believe it is an ice-sheet model but this is never really spelled out. This section should also incorporate all of the coupling that is included e.g. subglacial hydrology etc. as well as time-steps, resolution and the like. Then, I would follow that by all the boundary conditions e.g. your current sections 2.4 to 2.5.2. Then move on to the section that you call

"Ensemble parameter vector" which I suggest to rename to "Creation of sub-ensembles" as I believe it makes it more accessible. In this subsection, you absolutely need a table with the different sub-ensembles as well as a flowchart illustrating how each of these sub-ensembles was generated, e.g Baseline ensemble -> sieve 1 -> sieve 2 etc. Also, I find the section title "Bounding experiments" somewhat confusing. Do you mean end member scenarios? If so, why not name it like that?] The methods section in the revised draft was restructured following some of the above suggestions, including a table summarizing all experiments, merging the *Modeling approach* and *Model description* sections, and re-naming of sub-sections. Furthermore, a description of the GSM was submitted to GMD for open review and is already available as a preprint on Lev Tarasov's web page (https://www.physics.mun.ca/~lev/).

[In your results section you often speak about specific parameter vectors, for example in the caption of Figure 4 you mention "parameter vector 1". It is unclear to me to what sub-ensemble this refers to as parameter vector 1 is never really defined in the manuscript. So, I strongly recommend to define these parameter vectors somewhere in the methods section (e.g. in a table), so that the reader can go back and check what this refers to.] and [L310: As mentioned in my main points above, I simply do not know what parameter vector 1 refers to? Is this a single simulation, a composite, or a sub-ensemble?] and [L344: Again an explanation, definition what your refer to as parameter vector would be highly appreciated.] and [Fig. 4 caption: Again, it is unclear what parameter vector 1 is. I think one way of helping the reader other than a clear definition is a more intuitive name such as GIA sub-ensemble.] and [Table 1: Again it is unclear to me what the reference setup really is?]

**Changes:** Added *The GSM configuration in this study uses* 52 *model input parameters (ensemble parameters). The ensemble parameter space covered within this study is summarized in detail in Tab. S1. A parameter vector holds one value for each of these ensemble parameters. As such, each parameter vector fully specifies how the configuration of the GSM varies between ensemble members. To partly address potential non-linear dependencies of model results on the ensemble parameters, we use a* 20 *member (*20 *parameter vectors) high-variance reference ensemble (with respect to ensemble parameters and ice sheet configuration) instead of just a single reference run (*1 *parameter vector). The creation of the reference ensemble is described in detail in Sec. 2.4.1. The ice volume time series and the ice sheet surface elevation over the entire model domain at* 60 *and* 24 kyr BP *are shown for all* 20 *parameter vectors in Fig. S5 to S7.*

*For every sensitivity experiment (e.g., different basal sliding law), we then re-run the GSM for all* 20 *parameter vectors. A list of all sensitivity experiments conducted within this study is shown in Tab. 1. The analysis of each sensitivity experiment in comparison to the reference ensemble is described in detail in Sec. 2.4.2.*

[This issue was already raised by the other reviewer, but I believe that the effect of the transient climate forcing certainly warrants a discussion. First of all, it is never mentioned what climate forcing is used to drive the model. This goes back to my previous point that I am not sure whether GSM is an ice-sheet model or a coupled climate-ice sheet model. In any case, it remains unclear to me whether the differences you report here are due to your parameter changes or due to the changing transient forcing and it seems quite likely that there is at least some signal from the transient climate forcing in your results. This ought to be acknowledged and discussed.] and [On a related note, is it correct that your reference setup (e.g. Fig 7) does not show any Heinrich events, but simply a steady ice stream? I believe it is never mentioned, but your algorithm does not detect any surges in that time series right? But then it is unclear to me how you calculate the period and number of surges presented in Table 1.] To answer these comments, we expand on the logic described in the comment directly above. Fig. 7, for example, shows the effect when applying different GHFs for parameter vector 18. Therefore, the input parameters and the applied transient climate forcing are the same for all runs shown in Fig. 7. The differences are, therefore, not caused by a different climate forcing.

The climate forcing, however, varies between different parameter vectors depending on the

values of the climate input parameters in Table S1. Or to rephrase, each experiment is carried out with the same set of 20 different transient climate forcings (20 parameter vectors). As indicated in our response to the first referee comment, *The key idea behind this approach is to identify physical processes that show a significant effect across simulations with different ensemble parameters. This both increases confidence in our results and reduces the possibility that an observed modeling response is only due to, e.g., the chosen climate forcing.*

It is correct that the algorithm does not detect any surges (or, more precisely, detects less than 3 surges) when parameter vector 18 is run in the reference configuration. As described in the text and our response to the first referee comment, the reference ensemble (20 runs) can be divided into two high-variance (in parameter values and run metrics) subgroups: one subgroup of 10 for which all runs have more than 2 surges (at the higher resolution), and one for which this is not the case for the higher resolution run but is the case for the corresponding lower resolution run (i.e., with the same parameter vector). Parameter vector 18 belongs to the second subgroup, which is not included in the results presented in, e.g., Table 1. However, Fig. 7 clearly shows that surges occur when a smaller GHF is applied.

**Changes:** We updated the ensemble description, including a plot showing the ice volume of all parameter vectors when run in the reference configuration (new Fig. S5), and MIS 3 and LGM example time-slice map plots (new Fig. S6 and S7).

[You introduce the term "Minimum Numerical Error Estimate (MNEE)" which at least to me is a new concept. I think it should be clearly stated whether this is a common concept from the field of numerical analysis or something that you have introduced (I see that you used a the same approach in your previous paper). If you introduced it, it would be helpful to briefly(!!) motivate why this is a useful quantity. Because from what you mention in between lines 274 – 279, this seems a rather arbitrary choice (increasing your Picard iteration by one). I am also confused that your only parameter to measure MNEE is the "number of iterations". At the very least, I was expecting a combination of "number of iterations" and "residuals". The reason for this is that I expect the residual to be higher after 4 iterations if you are in a rapidly changing state than after 2 iterations in a relatively stable state. In my view, the way you have defined it is inconsistent because it does not tell anything about whether your solver has actually converged or not. For example, you can increase your number of iterations to 50, but that does not say anything about the quality of your solution. Also since GSM is run in serial, why not compare it to the solution from a direct solver like UMFPACK?] Yes, the concept of MNEEs was introduced in Hank et al. [2023]. They are specified thresholds for determining whether a difference in a sensitivity experiment is within a minimal estimate for the numerical uncertainty of the ice sheet dynamics solution. Given the non-linearity of surge onset and termination, this plays a potentially important part in our analysis. More specifically, as described in the text, MNEEs for the GSM are the maximum model difference across two different model simulations: 1) imposing stricter (than default) numerical convergence (what the referee refers to as smaller residual) and 2) imposing stricter numerical convergence with increased maximum iterations for the outer Picard loop (from 2 to 3, solving for the ice thickness) and the non-linear elliptic SSA (Shallow-Shelf Approximation) equation (from 2 to 4, solving for horizontal ice velocities). Therefore, the MNEEs are based on simulations with 1) smaller residual and 2) the combination of increased maximum number of iterations and smaller residual, but never just on an increased number of maximum iterations.

**Changes:** We updated the description of MNEEs according to the above. Refer to the Author's track changes for the detailed changes.

[At the end of the introduction you list five research questions that your paper aims to answer. This is admittedly a bit of a subjective matter, but my impression was that these questions are very generic and could be summarized with something like are our Hudson surges sensitive to perturbations in geothermal heat flux, GIA, ocean melting, etc. While certainly worth exploring, to me, this type of question layout is more suited to a thesis format but not necessarily for a paper. I also could not help, but get the impression that you do a bit of

everything which results in a lack of focus what you are really trying do address here. For example, I do not think that your paper actually addresses Q3 as you solely focus on the Hudson ice stream. As an add-on, I am also in favour of a short paper roadmap at the end of the introduction especially when it is as complicated and long as this paper.] We added a short roadmap to the end of the introduction.

Yes, what we are addressing is broad, but we do not see an alternative given our stated intention: *However, an extensive study simultaneously investigating the relative role of each proposed HE hypothesis is still missing. Furthermore, previous model-based tests of HE-related Hudson Strait surge cycling have largely ignored uncertainties in key potentially relevant processes and inputs. These include the deep geothermal heat flux under Hudson Strait, glacio-isostatic adjustment, and the form of the basal drag law employed.* In short, instead of tidy, idealized experiments addressing single hypotheses and ignoring the role of other possible mechanisms, we aim to address the lack of a comprehensive assessment of the relative role of proposed mechanisms while addressing key relevant uncertainties. Ignoring the latter negates the interpretative value of the former. We believe the stated research questions provide a clear structure for this assessment, and the question presentation succinctly summarizes the motivation for each question.

Furthermore, we suspect that not every reader will be interested in every detail of the results. Organizing the discussion section according to the research questions outlined in the introduction provides an easy way to get the most relevant information and allows the reader to then jump to individual results for more details.

As the majority of HE IRD in the North Atlantic is attributed to Hudson Strait, examining the ice shelf volume around it allows us to, at least partly, address Q3, especially considering that proxy records indicate an seasonally ice-free Labrador Sea during MIS3 [Hillaire-Marcel et al., 1994, Hesse et al., 1999, De Vernal et al., 2000, Gibb et al., 2014].

**Changes:** We included a discussion of the Labrador Sea and Baffin Bay ice shelf volume. Added *Even when considering fringing ice shelves along the Canadian coast (*52.5 *to* 75.0°N*), the maximum ice shelf volume between* 100 *and* 10 kyr BP *across all reference runs is only* $6.2 \cdot 10^4$ km$^3$ *and is below the minimum estimate of Hulbe et al. [2004] for* 12 *(out of* 20*) runs.*

**3 Technical corrections**

[L2: I think instead of using the term Glacial Systems Model, it would be better to refer to the type of model like coupled ice sheet-subglacial hydrology model. Of course, if you'd like to keep GSM in there, you could combine these two.] Glacial Systems Model (GSM) is simply the name of our model (e.g., comparable to Parallel Ice Sheet Model (PISM)). Since this name has been used in several publications [e.g., Drew and Tarasov, 2023, Hank et al., 2023], we prefer to keep it.

**Changes:** Additional details about the exact model type were added to the revised draft.

[L38: Not to be too picky, but I would argue that we did test the sensitivity of mPISM to the geothermal heat flux in our Schannwell et al. [2023] paper] We thank the referee for pointing this out.

**Changes:** Furthermore, previous model-based tests of HE-related Hudson Strait surge cycling have *not addressed uncertainties* in key potentially relevant processes and inputs. **to** Furthermore, previous model-based tests of HE-related Hudson Strait surge cycling have *largely ignored* uncertainties in key potentially relevant processes and inputs.

[L89: What is the physical mechanism for the choice to allow sliding at sub-freezing temperatures?] Observational and experimental evidence suggests that basal sliding starts below the pressure melting point and ramps up as the pressure melting point is approached [e.g., Barnes et al., 1971, Shreve, 1984, Echelmeyer and Zhongxiang, 1987, Cuffey et al., 1999, McCarthy et al., 2017, Mantelli et al., 2019]. Furthermore, Mantelli et al. [2019] show that an abrupt sliding onset at the transition from a cold-based ice sheet to an ice sheet bed at the

pressure-melting point causes refreezing on the warm-based side and, therefore, cannot exist. An additional numerical argument can be made on numerical grounds for coarse horizontal grid resolutions. It is unlikely that an entire grid cell reaches the pressure-melting point within one time step (e.g., 25×25km in 1 year). All of these aspects are described in detail in Hank et al. [2023].

**Changes:** We added [Hank et al., 2023] to the corresponding sentence.

[L93: Be precise here! What components are included in the current setup? Also asynchronous coupling mean different things to different communities. Does that mean you run your GIA model accelerated? Moreover, for your GIA model what kind of ice thickness distribution do you prescribe in the southern hemisphere or the other northern hemispheric ice sheets? This confusion originates from the fact that you never specify what your modelling domain is. In addition, it would be helpful to provide more detail about the GIA model, because out of the blue in the results section, you start talking about local and non-local GIA.] There is no acceleration. Visco-elastic GIA models are generally run on spherical harmonic grids and, therefore, must be global. We use the ice sheet chronologies from recently completed history matching for the non-NA inputs into the GIA calculation. As stated in section 2.1 of our submission: *The topography and sediment cover of the entire model domain are shown in Fig. S1.* As an aside, we have never seen "asynchronous" alone imply acceleration in modeling.

**Changes:** The revised draft includes more information about the GSM, the asynchronous coupling, and the different GIA models.

[L121: What is that resolution in kms for the Hudson ice stream. How does this coarse resolution affect your ability to model surging. We saw quite dramatic changes when we increased resolution from 50 km down to 20 km some years ago.] The grid cell size in Hudson Strait is roughly 25x25 km. As stated in the text *Due to the inclusion of a resolution-dependent basal temperature ramp [Hank et al., 2023], the differences in surge characteristics between the coarse resolution runs (horizontal grid resolution of $\Delta_{\mathrm{lon}} = 1.0°$, $\Delta_{\mathrm{lat}} = 0.5°$) and the reference runs ($\Delta_{\mathrm{lon}} = 0.5°$, $\Delta_{\mathrm{lat}} = 0.25°$) are generally within the MNEEs (Fig. 6). While finer (than the reference setup) horizontal grid resolutions are currently unfeasible in the context of this study, given the results of resolution response testing of surge cycling down to $3.125$ km horizontal grid resolution in Hank et al. [2023], the differences in surge characteristics for finer resolutions are also expected to be within the MNEEs.*

**Changes:** Added ... ($\Delta_{\mathrm{lon}} = 0.5°$ and $\Delta_{\mathrm{lat}} = 0.25°$, *corresponding to* $\sim 25x25$ km *in Hudson Strait*)

[L167: From your description it is unclear whether or not you do "Schoofing", meaning whether you prescribe the Schoof flux as an internal boundary condition at the grounding line. Please clarify.] As stated in our response to the first referee comment: *The GSM uses the Schoof [2007] grounding line flux condition as implemented in [Pollard and DeConto, 2012]. The authors only recently became aware of issues around this approach for complex 2D geometries likely of most consequence for Antarctica [Reese et al., 2018], and the revised validated treatment [Pollard and Deconto, 2020] has subsequently been implemented in the GSM.*

**Changes:** We tested sensitivities for different grounding line treatments on Hudson Strait surge characteristics and documented this in the revised draft (new Fig. S2 and S3). Using the revised validated treatment of Pollard and Deconto [2020] instead of the Schoof [2007] grounding line flux condition has no significant effect on the surge characteristics.

[L197: What do you do if you do not have any ocean points under your floating ice shelf? Do such situations arise?] We are not clear on what context the referee is referring to. However, for each of sub-shelf melt, GIA within the ice sheet grid, and ice dynamics, the GSM handles a changing ocean mask, and sub-marine temperatures are spatially extrapolated as needed for the subshelf melt calculation. This is detailed in the GSM description preprint [Tarasov et al., 2024].

[L199: Do you mean you tune your parameters for present-day ice sheets, because there is no Laurentide today.]

**Changes:** *... computed melt brackets present-day observations [e.g., Depoorter et al., 2013, Enderlin and Howat, 2013, Alley et al., 2015].* **to** *... computed melt brackets present-day observations for major Antarctic ice shelves [e.g., Depoorter et al., 2013, Enderlin and Howat, 2013, Alley et al., 2015].*

[L223: This makes it sound as if GSM has an ocean component coupled to it, where in fact you are using ocean temps. from a GCM simulations, right? If so, please rephrase.] Yes, the ocean temperatures are derived from the TraCE deglacial simulation run with the Community Climate System Model Version 3 [CCSM3, Liu et al., 2009]. We also already state in our submission *Using a glacial index approach, the ocean temperature chronology is interpolated between full glacial (last glacial maximum) and present day conditions for all other time slices.*

**Changes:** To better avoid the confusion the reviewer is indicating, we changed the phrase to *GSM ocean temperature forcing.*

[L230: Even for me as an ice-sheet modeller, this is getting quite hard to follow here. I am not sure, but is your ice shelf removal simply a very high basal melt rate that melts your ice shelf away? If so, what is the time it takes for the ice shelf to disappear and how might this potentially rather gradual removal affect the response in comparison to a sudden removal of the ice shelf?] The ice shelf removal experiments are simply based on increased ocean temperatures (as shown in Fig. 3). This experimental design is in line with proxy records indicating ocean temperature increases over a 1 to 2 kyr interval prior to HEs [e.g., Marcott et al., 2011]. The mechanism is also explicitly motivated in our initial presentation of $Q_2$: *A $2°C$ increase in the sub-surface ocean temperature has been shown to cause a 6 fold increase in the ice shelf basal melt rates in front of Hudson Strait ($\sim 6 \frac{m}{yr}$ to 35-40 $\frac{m}{yr}$) in simulations with an ocean/ice-shelf model [Marcott et al., 2011].* Since the GSM incorporates the relevant physics for rapid ice shelf disintegration, we consider this approach physically more defensible than artificially removing the entire ice shelf.

As stated in the text *The ocean temperature at the relevant depth is then used to calculate the sub-shelf melt $M_{\text{SSM}}$ and terminus face melt $M_{\text{face}}$.* Since both $M_{\text{SSM}}$ and $M_{\text{face}}$ depend on ensemble parameters and the ocean temperature of the current experiment, the time it takes for the ice shelf to disappear varies between different parameter vectors and experiments. The ice shelf response to a change in ocean temperature and calving varies from no significant change to rapid ($< 100$ yr) ice shelf disintegration (e.g., Fig. 13).

[L349: Again, this is pretty much what we showed in Schannwell et al. [2023].]

**Changes:** Added *This shift in surge mode is in agreement with the results of a previous study examining the effect of GHF on HEs [Schannwell et al., 2023].* to the discussion.

[L365: What does crashing mean here? Solvers did not converge anymore?] Yes, the solvers did not converge with the specified minimum model time step of 0.015625 yr. This threshold could be lowered, but given how far this is already below nominal CFL for our grid resolution, such small time-steps raise concerns about non-linear numerical instabilities that will distort our analysis.

**Changes:** Added *Model runs automatically terminate once the time step size is reduced below the set minimum of 0.015625 yr. Such ("crashed") runs are not considered for any analysis conducted within this study.*

[L366: I think, this certainly needs some discussion why you think the switch from Weertman to Coulomb makes such a big difference in the run time.] As mentioned in our response to the first referee comment: *As the regularized Coulomb law negligibly increases basal drag beyond the order of the regularization threshold ($UV_{\text{C,reg}} = 20$ m/yr), the regularized Coulomb law lacks the increase in basal drag with basal velocity intrinsic to Weertman sliding and therefore will be more numerically unstable. This is compounded by the schoofing grounding-line flux iteration in the SSA solution.*

*It should also be noted that the GSM SSA solution imposes an upper bound of 40 km/yr on SSA ice velocities for this configuration. We suspect that this is higher than most other models. The imposition of this upper bound is itself another non-linearity in the solution that*

*can contribute to both instability (as adding non-linearities will generally decrease convergence of iterative solutions) and stability (by limiting ice velocities).* A numerical challenge with Coulomb plastic (regularized or not) is that, unlike Weertman, there is a constant basal drag term in the SSA equation that does not automatically/implicitly have the correct sign.

**Changes:** Added *According to CFL constraints, we expect even the regularized Coulomb law (Eq. 6b) to be more unstable than the Weertman law, because it negligibly increases basal drag for basal velocities beyond the order of the regularization threshold $UV_{C,reg} = 20$ m yr$^{-1}$. This is compounded by the grounding-line flux iteration in the SSA solution.*

[L392: In the interest of shortening the paper, consider removing everything regarding the regularized Coulomb and simply state that run times were too long to make these runs feasible.] We respectfully decline this suggestion. A major motivation of this paper was to comprehensively address key uncertainties not (or at least mostly not) addressed in previous tests of hypotheses for explaining Heinrich events. Uncertainties in basal drag law are one such uncertainty; thus, we consider it important to retain this subsection.

[L457: Are you sure your mean "increase" here? It is possible, but I would argue an elevation decrease is more often than not associated with an accumulation decrease.]

Yes, increase is correct here, but the statement should have been *precipitation (and therefore accumulation over most of the ice sheet) generally increases ....* Assuming everything else is the same, a lower ice sheet surface elevation will have a higher precipitation rate (relation between atmospheric lapse rate and the Clausius–Clapeyron formula for saturation vapour pressure) though this is complicated by slope orographic forcing dependencies of precipitation in the GSM.

**Changes:** *... accumulation generally increases ...* **to** *... precipitation (and therefore accumulation over most of the ice sheet) generally increases ...*

[L569: What are pseudo-Hudson Strait surges? Bassis et al. use an idealized setup that is based on the geometry of the Hudson Strait. Does GSM have the marine-ice-cliff instability mechanism implemented? This is an integral part of the Bassis et al. mechanism and renders the comparison pretty far-fetched if it doesn't.] "pseudo"-Hudson Strait surges is an earlier notation.

Yes, the GSM does have the MICI, but only for iced grid cells adjacent to neighbouring open ocean. However, as stated in the text: *Due to the numerous differences in the model setup (e.g., model domain considered, grid discretization near the grounding line, GIA model, calving and sub-shelf melt implementations, and the lack of ice thermodynamics in Bassis et al. [2017]), we do not aim to directly replicate the experiments in Bassis et al. [2017]. Instead, we examine the role of SSOW in a HE context by applying a sub-surface ocean temperature increase for every DO event.*

**Changes:** Removed *"pseudo"-*.

Added *The GSM includes the dynamics of Marine Ice Cliff Instability [MICI, DeConto and Pollard, 2016], but only for ice-covered grid cells adjacent to neighbouring open ocean.*

[L574: I find these melt-rates unreasonably high. I mean maximum present-day melt rates are around 100 m/yr and you have four times the melt during the glacial? That seems very hard to believe.] 400 m/yr are the highest melt rates occurring in all of our experiments, including the end-member scenarios. Since some end-member scenarios were specifically designed to increase the melt rate, it is unsurprising that the maximum modeled melt rates exceed the maximum observed present-day melt rates. The melt rates in the reference setup are generally below 100 m/yr.

**Changes:** sub-marine melt can reach up to $400 \, \text{m yr}^{-1}$ in our simulations **to** sub-marine melt can reach up to $400 \, \text{m yr}^{-1}$ in our *end-member scenario* simulations *(specifically designed to increase the melt rate; melt rates in the reference setup are generally below $100 \, \text{m yr}^{-1}$)*

[L606: I am pretty sure that you did not show synchronization of your HEs with the Dansgaard-Oeschger cycles.] We agree that we did not show a synchronization of HEs with the coldest phases of the Bond cycles. However, the timing of surges is affected by the applied

additional ocean forcing, indicating the possibility of synchronization.

**Changes:** *Based on our results, Hudson Strait ice stream surge cycling is the most likely Heinrich Event mechanism, but ocean forcings can affect the timing of surges and provide a means to synchronize HEs with the coldest phases of the Bond cycles.* **to** *Overall, our experiments indicate that Hudson Strait ice stream surge cycling is the most likely Heinrich Event mechanism, but ocean forcings can indirectly affect the timing of surges through a change in ice sheet evolution. The key HE characteristic our experiments have not resolved is HE synchronization with the coldest phases of the Bond cycles (though not always the case, e.g., HE1). While ice shelf collapse and ocean forcing are insufficient to synchronize Hudson Strait ice stream surge cycling in the experiments presented here, a synchronization mechanism without the need for a trigger event during the stadial [Schannwell et al., 2024] could potentially provide the missing link and should be explored in future studies.*

**4 Figures**

[The Figures are overall of good quality. What I am missing is a Figure of the modelling domain. If it is a global setup, this is not needed, but then this needs to be stated clearly in the text.] As stated in section 2.1 of our submission: *The topography and sediment cover of the entire model domain are shown in Fig. S1.* Additionally, as indicated above, we included some whole NAIS (showing whole model domain) example time slices.

**Changes:** Addition of Fig. S6 and S7.

[Fig. 2: The contours are very hard to see in the left but primarily in the right panel.] **Changes:** Adjusted contours in Fig. 2.

[Fig. 4: and throughout. I am not a big fan of pythons default option to have the scientific notation on top of each subplot in pretty small font. For a second I thought that your flux was as high as 2 Sv. You could try using mSv instead or work that exponent into your axis titles.]

**Changes:** For all figures in the main manuscript, we changed the units where possible and otherwise used, e.g., $10^5$ km$^2$ in the axis label instead of the python scientific notation. We will hold off on updating all the figures in the supplement in case further figure adjustments are required.

[Fig. 5: Consider removing repetitive colourbars and axis labels for the benefit of larger panels.]

**Changes:** All repetitive legends, labels, and colorbars were removed.

[Fig. 7: Judging from this, your reference simulations has no Heinrich events? But why is there an ice volume increase at the same time as the ice flux increases? What is the origin of this?] As already addressed in our response above, parameter vector 18 (base vector shown in Fig. 7) is part of the ensemble with < 3 surges detected. In the Fig. 7 plots, the reference run is shown in orange. The most significant growth in Hudson Strait ice flux occurs between 70 and 60 kyr BP. This is also the time of the most significant whole ice sheet volume growth in the model. The increased ice volume eventually increases ice flux to and, consequently, through Hudson Strait (Fig. 1 in this response). As this is peripheral to the focus of our submission, we see no need to add this brief analysis to it.

[Supplementary Figures: I can also only reiterate what the other reviewer mentioned, 40 Figures in the supplement is a lot and you should really make sure that each of these Figures serves a purpose. For example, what does the Figure S8 add? In the text it is referenced to show a more gradual increase in ice flux, but I have no idea how a snapshot of the velocity field could convey such a message.] We have verified that every supplemental figure is referenced at least once in the main text. We are aware this is on the long side, but we are generally going by our rule of thumb that if more than one reader is likely to want to see the plot and less than a dozen are likely to, then stick in the supplement. Most readers are free to ignore the supplement and just note that there is graphical documentation of the relevant claim.

[Figure]

Figure 1: The ice sheet surface elevation and basal ice velocity of parameter vector 18 are shown at 70.0 kyr BP and 60 kyr BP. The black contour is the present-day coastline provided by *cartopy* [Met Office, 2010 - 2015].

Fig. S8 (new Fig. S10) is used in the following sentence: *During these surges, ice transport from Hudson Bay and Foxe Basin through Hudson Strait (and other outlets) towards the ice sheet margin increases (e.g., Fig. S8).* The confusion may stem from the abbreviation S8 for surge 8 (appearing in *"gradual increases in Hudson Strait ice flux (S8 and S9)"*), which was rectified in the revised draft.

**Changes:** We merged plots with similar content and changed the abbreviation S for surge to P (surge peak). Added *The supplementary material referenced throughout this text (indicated by a capital S) provides additional information to support the corresponding claim but is not essential to the understanding of this study.*

**References**

Richard B. Alley, Sridhar Anandakrishnan, Knut Christianson, Huw J. Horgan, Atsu Muto, Byron R. Parizek, David Pollard, and Ryan T. Walker. Oceanic forcing of ice-sheet retreat: West Antarctica and more. *Annual Review of Earth and Planetary Sciences*, 43:207–231, 2015. ISSN 00846597. doi: 10.1146/annurev-earth-060614-105344.

P. Barnes, D. Tabor, and J. C. F. Walker. The friction and creep of polycrystalline ice. *Proceedings of the Royal Society of London. Series A, Mathematical and Physical Sciences*, 324(1557):127–155, 1971. ISSN 00804630. URL http://www.jstor.org/stable/77933.

Jeremy N Bassis, Sierra V Petersen, and L Mac Cathles. Heinrich events triggered by ocean forcing and modulated by isostatic adjustment. *Nature*, 542(7641):332—334, February 2017. ISSN 0028-0836. doi: 10.1038/nature21069. URL https://doi.org/10.1038/nature21069.

K. M. Cuffey, H. Conway, B. Hallet, A. M. Gades, and C. F. Raymond. Interfacial water in polar glaciers and glacier sliding at -17 °C. *Geophysical Research Letters*, 26(6):751–754, 1999. ISSN 00948276. doi: 10.1029/1999GL900096.

Anne De Vernal, Claude Hillaire-Marcel, Jean Louis Turon, and Jens Matthiessen. Reconstruction of sea-surface temperature, salinity, and sea-ice cover in the northern North Atlantic during the last glacial maximum based on dinocyst assemblages. *Canadian Journal of Earth Sciences*, 37(5):725–750, 2000. ISSN 00084077. doi: 10.1139/cjes-37-5-725.

Robert M. DeConto and David Pollard. Contribution of Antarctica to past and future sea-level rise. *Nature*, 531(7596):591–597, 2016. ISSN 14764687. doi: 10.1038/nature17145. URL http://dx.doi.org/10.1038/nature17145.

M. A. Depoorter, J. L. Bamber, J. A. Griggs, J. T.M. Lenaerts, S. R.M. Ligtenberg, M. R. Van Den Broeke, and G. Moholdt. Calving fluxes and basal melt rates of Antarctic ice shelves. *Nature*, 502(7469):89–92, 2013. ISSN 14764687. doi: 10.1038/nature12567.

M. Drew and L. Tarasov. Surging of a hudson strait-scale ice stream: subglacial hydrology matters but the process details mostly do not. *The Cryosphere*, 17(12):5391–5415, 2023. doi: 10.5194/tc-17-5391-2023. URL https://tc.copernicus.org/articles/17/5391/2023/.

Keith Echelmeyer and Wang Zhongxiang. Direct Observation of Basal Sliding and Deformation of Basal Drift at Sub-Freezing Temperatures. *Journal of Glaciology*, 33(113):83–98, 1987. ISSN 0022-1430. doi: 10.3189/s0022143000005396.

Ellyn M. Enderlin and Ian M. Howat. Submarine melt rate estimates for floating termini of Greenland outlet glaciers (2000-2010). *Journal of Glaciology*, 59(213):67–75, 2013. ISSN 00221430. doi: 10.3189/2013JoG12J049.

Olivia T. Gibb, Claude Hillaire-Marcel, and Anne de Vernal. Oceanographic regimes in the northwest Labrador Sea since Marine Isotope Stage 3 based on dinocyst and stable isotope proxy records. *Quaternary Science Reviews*, 92:269–279, 2014. ISSN 02773791. doi: 10.1016/j.quascirev.2013.12.010. URL http://dx.doi.org/10.1016/j.quascirev.2013.12.010.

K. Hank, L. Tarasov, and E. Mantelli. Modeling sensitivities of thermally and hydraulically driven ice stream surge cycling. *Geoscientific Model Development*, 16(19):5627–5652, 2023. doi: 10.5194/gmd-16-5627-2023. URL https://gmd.copernicus.org/articles/16/5627/2023/.

Reinhard Hesse, Ingo Klauck, Saeed Khodabakhsh, and David Piper. Continental slope sedimentation adjacent to an ice margin. III. The upper Labrador Slope. *Marine Geology*, 155(3-4):249–276, 1999. ISSN 00253227. doi: 10.1016/S0025-3227(98)00054-1.

C. Hillaire-Marcel, A. De Vernal, G. Bilodeau, and G. Wu. Isotope stratigraphy, sedimentation rates, deep circulation, and carbonate events in the Labrador Sea during the last ~200 ka. *Canadian Journal of Earth Sciences*, 31(1):63–89, 1994. ISSN 00084077. doi: 10.1139/e94-007.

Christina L. Hulbe, Douglas R. MacAyeal, George H. Denton, Johan Kleman, and Thomas V. Lowell. Catastrophic ice shelf breakup as the source of Heinrich event icebergs. *Paleoceanography*, 19(1):n/a–n/a, 2004. ISSN 0883-8305. doi: 10.1029/2003pa000890.

Z. Liu, B. L. Otto-Bliesner, F. He, E. C. Brady, R. Tomas, P. U. Clark, A. E. Carlson, J. Lynch-Stieglitz, W. Curry, E. Brook, and et al. Transient Simulation of Last Deglaciation with a New Mechanism for Bolling-Allerod Warming. *Science*, 325(5938):310314, Jul 2009. ISSN 1095-9203. doi: 10.1126/science.1171041. URL http://dx.doi.org/10.1126/science.1171041.

E. Mantelli, M. Haseloff, and C. Schoof. Ice sheet flow with thermally activated sliding. Part 1: the role of advection. *Proceedings of the Royal Society A: Mathematical, Physical and Engineering Sciences*, 475(2230):20190410, 2019. ISSN 14712946. doi: 10.1098/rspa.2019.0410.

Shaun A. Marcott, Peter U. Clark, Laurie Padman, Gary P. Klinkhammer, Scott R. Springer, Zhengyu Liu, Bette L. Otto-Bliesner, Anders E. Carlson, Andy Ungerer, June Padman, Feng He, Jun Cheng, and Andreas Schmittner. Ice-shelf collapse from subsurface warming as a trigger for Heinrich events. *Proceedings of the National Academy of Sciences of the United States of America*, 108(33):13415–13419, 2011. ISSN 00278424. doi: 10.1073/pnas.1104772108.

C. McCarthy, H. Savage, and M. Nettles. Temperature dependence of ice-on-rock friction at realistic glacier conditions. *Philosophical Transactions of the Royal Society A: Mathematical, Physical and Engineering Sciences*, 375(2086):20150348, 2017. ISSN 1364503X. doi: 10.1098/rsta.2015.0348.

Met Office. *Cartopy: a cartographic python library with a Matplotlib interface*. Exeter, Devon, 2010 - 2015. URL https://scitools.org.uk/cartopy.

D. Pollard and R. M. DeConto. Description of a hybrid ice sheet-shelf model, and application to Antarctica. *Geoscientific Model Development*, 5(5):1273–1295, 2012. ISSN 1991959X. doi: 10.5194/gmd-5-1273-2012.

David Pollard and Robert M. Deconto. Improvements in one-dimensional grounding-line parameterizations in an ice-sheet model with lateral variations (PSUICE3D v2.1). *Geoscientific Model Development*, 13(12):6481–6500, 2020. ISSN 19919603. doi: 10.5194/gmd-13-6481-2020.

Ronja Reese, Ricarda Winkelmann, and G. Hilmar Gudmundsson. Grounding-line flux formula applied as a flux condition in numerical simulations fails for buttressed Antarctic ice streams. *Cryosphere*, 12(10):3229–3242, 2018. ISSN 19940424. doi: 10.5194/tc-12-3229-2018.

C. Schannwell, U. Mikolajewicz, F. Ziemen, and M.-L. Kapsch. Sensitivity of heinrich-type ice-sheet surge characteristics to boundary forcing perturbations. *Climate of the Past*, 19(1):179–198, 2023. doi: 10.5194/cp-19-179-2023. URL https://cp.copernicus.org/articles/19/179/2023/.

Clemens Schannwell, Uwe Mikolajewicz, Marie Luise Kapsch, and Florian Ziemen. A mechanism for reconciling the synchronisation of Heinrich events and Dansgaard-Oeschger cycles. *Nature Communications*, 15(1):1–8, 2024. ISSN 20411723. doi: 10.1038/s41467-024-47141-7.

Christian Schoof. Marine ice-sheet dynamics. Part 1. The case of rapid sliding. *Journal of Fluid Mechanics*, 573:27–55, 2007. ISSN 00221120. doi: 10.1017/S0022112006003570.

R. L. Shreve. Glacier sliding at subfreezing temperatures. *Journal of Glaciology*, 30(106):341–347, 1984. ISSN 00221430. doi: 10.1017/S0022143000006195.

Lev Tarasov, Benoit S. Lecavalier, Kevin Hank, and David Pollard. The glacial systems model (GSM). 2024. URL https://www.physics.mun.ca/~lev/GSM.pdf.

**Author's response to Jorge Alvarez-Solas's Comment 1**

September 22, 2024

[In this article, Hank and Tarasov, perform several simulations of the Laurentide ice sheet of the last glacial period in order to elucidate the most likely candidates to explain the origin of Heinrich events. The subject and the way it is addressed in the introduction and discussion are of very high relevance. The paper is potentially very well suited for publication in Climate of the Past. I see, however, two major deficiencies of the current manuscript: A questionable and irreproducible experimental setup and the lack of adequate conclusions. I expand these aspects in the following.]

We thank the referee Jorge Alvarez-Solas for their constructive comments. A point-by-point reply is reported below, with referee comments in orange and our replies in black. We address the claimed *two major deficiencies*, which was likely due to a misunderstanding of the experimental design in our detailed responses below. We agree with the specific referee comments not listed here and revised the manuscript accordingly. For revisions too substantial to be explicitly stated here (e.g., restructuring of the methods section), we added a brief description and refer the interested reader to the Author's track changes for the detailed changes.

**1 General comments regarding the experimental setup and the exploration of the parameter space:**

[Your experimental setup is firstly very hard to understand and secondly not reproducible at all. It is hardly understandable because your entire exploration of the parameter space relies on perturbing the values of the parameters of your table S1. However the majority of these parameters are not explained at all in the manuscript. The very pertinent questions you are addressing in this article are conveniently described at the end of the Introduction paragraph. They summarize the different mechanisms triggering Hudson Strait ice surges in the context of HEs. However, it is difficult to understand how exploring, for example, the values of the "northwestern desert-elevation cutoff" is relevant in this context. The equations on which these not-defined parameters apply are not present in the manuscript, so the reader can not even guess what influence they have on the scientific questions you are addressing. This is the case because the exploration of these parameters only helps to see whether a given realization of the model has fulfilled or not your sieves, but does not give any mechanistic picture of their influence on ice stream behavior. Nor these experiments can be reproduced with other models of similar physics. An article should be as shelf-contained as possible, so if you analyze an ensemble of simulations that have fulfilled a given sieve, the reader can understand why that is the case and the implications for your conclusions without having to open the experimental setup of another manuscript or guess that it is going to be more deeply explained in another article in preparation.] and [In my opinion, both these two problems would be solved by the following strategy: You could fix several of the parameters that have already allowed you to fulfill the sieves and that are mechanistically irrelevant for the ice stream behavior to a single documented value. The equations on which these parameters appear could be summarized in an appendix. Then, you could systematically explore the parameter space of the processes that

are suspected to be very influential on the ice-stream dynamics (e.g. sliding laws, effective pressure, dependence on bed type and base thermal state...; see below). And finally you could describe how these different values affect the nature of the simulated Hudson Strait ice stream variability] The reviewer apparently does not understand that our experimental design is precisely what they suggest: "fix several of the parameters"; however, we do this twenty times over our 20 base parameter vectors for all 52 parameter vector components. Upon re-reading our section 2.2, we can now see that the text would benefit from more clarity that we use the same 20 GSM parameter vectors for all experiments (to do otherwise would make no sense).

This clarification should address the reviewer's concerns. However, in case they were clear on the experimental design and still have issues with the use of an ensemble, we add the following points.

A flaw in the reviewer's reasoning is their presumption that most of the ensemble parameters are apriori "mechanistically irrelevant". Thermodynamic surge cycling will depend on energy balance, which depends on surface temperature, precipitation, and ice thickness. As such, most of the ensemble parameters related to climate forcing (the majority of our ensemble parameters) are potentially relevant. This also holds for the remaining ensemble parameters related to GIA, mass-balance processes, ice deformation, and basal drag. The key point of our ensemble design is to address parametric uncertainties. This is done by extracting a high-variance sample of non-implausible model configurations (each specified by a parameter vector) and then conducting sensitivity experiments (with respect to inputs or process suppression, not ensemble parameters) on this same sample of parameter vectors. Otherwise, it would remain unclear whether the results obtained through a set of sensitivity experiments are only valid for a single chosen configuration (i.e., a single fixed parameter vector).

As to *"An article should be as self-contained as possible"*, this would entail every modelling paper describing every parameter in the model, every equation, and every discretization. Obviously, this is always a balancing act for non-trivial geophysical models.

**Changes:** We added more overall description of the ensemble parameters, but it is unreasonable to expect a detailed description of each parameter (which would extremely bloat every GCM-based modelling paper). Furthermore, a GSM description has been submitted to GMD and is already available as a preprint on Lev Tarasov's web page (https://www.physics.mun.ca/~lev/).

In summary, and as already indicated in our responses to the other referee comments, the goal of this study is not to explore the effect of ensemble parameters on HEs. Instead, the ensemble-based approach aims to (albeit incompletely) account for the uncertainties associated with these ensemble parameters (ranges in Table S1). In comparison, other modelling studies often set the values of model parameters comparable to the ones presented in Table S1 to a single value and, therefore, completely ignore the associated parametric uncertainties.

**2 Specific comments regarding the experimental setup and the exploration of the parameter space:**

[What are the values of $C_{\text{warm}}$ and $C_{\text{froz}}$? (absent in table S1)]

**Changes:** Added the value for $C_{\text{froz}} = 2 \cdot 10^{-4}$ m yr$^{-1}$ $\left(5 \cdot 10^{-6} \text{ Pa}^{-1}\right)^4$ and the following description of $C_{\text{warm}}$: $C_{\text{warm}}$ *is the fully warm-based sliding coefficient with dependence on bed properties:*

$$C_{\text{warm,soft}} = C_{\text{rmu}} \cdot \min\left[1; \max\left[0.2; \frac{F_{\text{sub,till}}}{0.01 \ \sigma_{\text{hb}}}\right]\right] \tag{1a}$$

$$C_{\text{warm,hard}} = C_{\text{slid}} \cdot \min\left[1; \max\left[0.1; \frac{F_{\text{sub,slid}}}{0.01 \ \sigma_{\text{hb}}}\right]\right] \cdot \left(1 + 20 \ F_{\text{sed}}\right), \tag{1b}$$

*Here $C_{\text{rmu}}$ and $C_{\text{slid}}$ are the ensemble parameters for the Weertman soft- and hard-bed sliding coefficients, respectively (Table S1). The ensemble parameters $F_{\text{sub,till}}$ and $F_{\text{sub,slid}}$ (Table S1)*

*impose the Weertman basal drag dependencies on the subgrid standard deviation of the bed elevation $\sigma_{hb}$ and $F_{sed}$ is the subgrid fraction of soft bed cover.*

[What is the value of $n_b$ for hard bedrock? (I could guess it is a 3 from a dimensional analysis based on the units of table S1, but it is not present in that table)] and [Have you explored different values of $n_{b,hard}$? I assume not (from what is shown in table S1). If not, why not simply use the same value as for $n_{b,soft}$? That will reduce one degree of freedom.] and [According to equations (2) and (3) your sliding coefficient, $C_b$ is a function of the bedrock type (soft sediments vs hard), the thermal character of the base and the effective pressure. Why do you need additionally to change the exponent of the sliding law over soft and hard bedrocks?]

The hard-bedded sliding exponent has not been explored in this context. Based on our reading of the literature, we judge the form of the soft bedded sliding as less constrained with more potential impact on results given the generally lower basal drag and higher fluxes.

Weertman's reasoning of controlling obstacle sizes places the hard bed exponent at 2 or 3. However, based on the results of a basal drag inversion for Greenland [Maier et al., 2021], a value of 4 appears to be more representative (at least for Greenland). Soft bedded sliding at small scales is likely Coulomb plastic, but the appropriate form at large scale has been subject to long ongoing debate. Therefore, there is no reason to assume that the exponent should be the same for both hard and soft beds.

**Changes:** see $n_{b,soft}$ in Table S1 **to** $n_{b,hard} = 4$ [Maier et al., 2021] and ensemble parameter $n_{b,soft}$ in Table S1

[Please justify the 3.5 exponent in equation (4)] and [I deduce from table S1 that your maximum allowed value for $h_{wb,Crit}$ is 1.0. Why limit it to that relatively low value? What happens if you significantly increase it?] and [Additionally, what does the model do if $h_{wb}$ reaches $h_{wb,Crit}$? Neff could not be reduced further (it would be purely 0!! From equation (4)), but what happens with the excess of heat? Can you create additional basal water? If not, could you diagnose the "free" values of $h_{wb}$? How sensitive are the surging cycles to this basal water limit?] To clarify, $h_{wb,Crit}$ is an estimated effective bed roughness scale, not the maximum basal water thickness. The maximum basal water thickness is $h_{wb,max} = 10$ m. All additional sub-glacial meltwater ($h_{wb} > 10$ m) leaves the ice sheet. We re-did a set of simulations with the 10 m limit raised to 100 m. This leads to slightly fewer, shorter, and stronger surges, but the differences are generally smaller than the MNEEs and are, therefore, numerically insignificant (new Fig. S2 and S3).

While it is true that $N_{eff}$ in Eq. 4 can reach 0 kPa, the addition of $N_{eff,min} = 10$ kPa in the denominator of Eq. 3 enforces that the effective pressure used to determine the basal sliding coefficient $C_b$ never falls below $N_{eff,min}$.

Eq. 4 and its exponent 3.5 are empirically derived [Flowers, 2000]. The parameter range for $h_{wb,Crit}$ expands around the $h_{wb,Crit} = 0.1$ m value used by Flowers and Clarke [2002].

The effect of the local basal hydrology model and its parameters (e.g., $h_{wb,max}$, $h_{wb,Crit}$, $N_{eff,min}$) on surges has also been extensively examined in previous studies [Drew and Tarasov, 2023, Hank et al., 2023].

**Changes:** *The effective pressure itself is calculated according to* **to** *The effective pressure is given by an empirically-derived dependence [Flowers, 2000] on basal water thickness $h_{wb}$:*

$h_{wb,Crit}$ *an estimated effective bed roughness scale [ensemble parameter in Table S1, see also Drew and Tarasov, 2023]* **to** $h_{wb,Crit}$ *an estimated effective bed roughness scale [ensemble parameter in Table S1, see also Flowers and Clarke, 2002, Drew and Tarasov, 2023]*

**Added** *The local basal hydrology model nominally sets the time derivative of $h_{wb}$ to the difference between the basal melt rate $M_b$ (from conservation of energy at the ice sheet base) and a constant bed drainage rate $R_{b,drain}$. The basal water thickness is limited to $h_{wb,max} = 10$ m and is set to $h_{wb} = 0$ m where the ice thickness is less than $h_{hyd,lim} = 10$ m and where the temperature with respect to the pressure melting point ($T_{bp}$) is below $T_{bp,lim} = -0.1°C$. Previous experiments showed changes in $h_{wb,max}$, $h_{hyd,lim}$, and $T_{bp,lim}$ do not significantly affect surge characteristics [Hank et al., 2023]. Additional experiments using $h_{hyd,lim} = 100$ m conducted*

*within this study support this finding (Fig. S2 and S3).*

 $C_c$ is a scalar (does not vary in space or time) ensemble parameter.

**Changes:** where $C_c$ is a variable drag coefficient (ensemble parameter, ... **to** where $C_c$ is a variable drag coefficient (*scalar* ensemble parameter, ...

 To our knowledge, the use of the minimum of the two computed basal drags was first posed and motivated in a glaciological context by Tsai et al. [2015]. In part the motivation is: if Coulomb friction is high, Weertman-type enhanced deformation around controlling obstacles can still occur (especially given the physical separation of the Coulomb plastic deformation process within the till layer) and dominate the basal sliding. We do not follow the reviewer's reasoning on how a high Coulomb drag could stabilize the flow given the thermo-mechanical coupling and further non-linearity introduced by basal hydrology. Concretely, high Coulomb drag in Hudson Strait would not stay high as basal meltwater accumulates.

**Changes:** *To account for possible Weertman-type sliding when Coulomb drag is high, the basal shear stress used in the GSM when the Coulomb drag option is activated is set to* **to** *When Coulomb drag is high, Weertman-type enhanced deformation around controlling obstacles can still occur (especially given the physical separation of the Coulomb plastic deformation process within the till layer) and dominate the basal sliding. Therefore, the basal shear stress used in the GSM when the Coulomb drag option is activated follows Tsai et al. [2015]:* $\boldsymbol{\tau_b} = \min[\boldsymbol{\tau_{b,W}}; \boldsymbol{\tau_{b,C}}]$.

**3    GHF reconstructions:**

 and  The map of Blackwell and Richards [2004] was published by the American Association of Petroleum Geologists (AAPG) in 2004. While there is no peer-reviewed article associated with this map, other studies show a similarly low GHF in and around Hudson Strait and Hudson Bay with a negative northward trend [Jessop and Judge, 1971, Levy et al., 2010, Jaupart et al., 2014]. The GHF in Hudson Strait and Hudson Bay presented by Shapiro and Ritzwoller [2004] varies between 40 and 50 mW/m², not 55 mW/m². This, along with the approximate 20 mW/m² standard deviation, implies there is a 18% chance of the GHF being 20 mW/m². The mean Hudson Strait/Hudson Bay GHF in Pollack et al. [1993] is 56.1 mW/m². The values in Lucazeau [2019] are difficult to interpret towards the lower

end because all values below 45 mW/m$^2$ have the same colour. The reference to Cuesta-Valero et al. [2021] was added to show the sparsity of geothermal borehole data in Hudson Strait and Hudson Bay. This can also be seen by browsing the IHFC Global Heat Flow Database [e.g., Fuchs et al., 2023]. The key point of this discussion is that the GHF for Hudson Bay is highly uncertain.

[Line 345 reads: "The exact transition point [to the binge-purge mode] depends on the parameter vector in question but generally requires a Hudson Strait/Hudson Bay GHF ave $\leq 37$ mW/m$^2$. And line 357 states: "Therefore both types of Hudson Strait ice stream surge cycling are consistent within available GHF constraints."] and [In light of the reconstructed values described in your referenced studies, this last sentence seems highly inaccurate or simply wrong. It should rather say something like: "The binge-purge mode is, under our experimental setup, only accessible if GHF¡37 mW/m$^2$ which represents the lower bound of available constraints".] The additional references above [Jessop and Judge, 1971, Levy et al., 2010, Jaupart et al., 2014] indicate that some GHF estimates in and around Hudson Strait and Hudson Bay are indeed as low as 20 mW/m$^2$ (as stated in the manuscript). Therefore, we stand by our current statement. **Changes:** We add the references Jessop and Judge [1971], Levy et al. [2010], Jaupart et al. [2014] to the revised manuscript.

**4 Conclusions of the manuscript in light of the experimental setup**

[What is the influence of $h_{\mathrm{wb,Crit}}$ on the transition to binge-purge? Would this transition be possible if basal water is conserved?] As mentioned above, $h_{\mathrm{wb,Crit}}$ is an estimated effective bed roughness scale, not the maximum basal water thickness $h_{\mathrm{wb,max}}$. As the exact value of $h_{\mathrm{wb,Crit}}$ is uncertain, it is one of the ensemble parameters, and our reference ensemble covers the range 0.017 to 0.912 m. The influence of $h_{\mathrm{wb,max}}$ has been examined in previous studies [Drew and Tarasov, 2023, Hank et al., 2023]. In general, the model results show only a small sensitivity to a change in $h_{\mathrm{wb,max}}$. See also the new Fig. S2 and S3 for an experiment with $h_{\mathrm{wb,max}} = 100$ m.

[Your basal friction laws are designed in a way that effective pressure can be reduced several orders of magnitude (even to purely 0!) when there's enough basal water. Also, $C_b$ will accordingly reach extremely high values (equation (3)) when effective pressure drops. Do you think this is glaciologically realistic (as opposed to Leguy's parameterization, for example, or even the PISM treatment?] As outlined previously, we ensure that the effective pressure used to calculate $C_b$ never falls below $N_{\mathrm{eff,min}} = 10$ kPa (Eq. 3). Eq. 3 further states that the multiplicative effective pressure term in the applied basal sliding coefficient is limited to a value of 10. Therefore, we consider the sliding law used within this study as glaciologically realistic as other implementations.

[The regularized Coulomb law is thought to perform well, even capturing the heterogeneous character of the bed without needing to guess different friction coefficients in soft and hard beds (Joughin et al., 2019). In section 3.3 it is stated that the majority of the Coulomb runs crashed. And that it significantly increased the needed computational time for those who survived. Why is this? Could it be because of equation (6)? Or because of the additional complexity of the parameterizations over the friction coefficient and effective pressure introduced in equations (2), (3) and 4? Again, what would be the effect of letting the Coulomb law do its job, without limiting $\tau_b$?] Joughin et al. (2019) only examined Pine Island Glacier. Maier et al. [2021] also found major sectors of Greenland adhere to Weertman-type hard bed basal drag as compared to a hard bed with cavitation Mohr-Coulomb-like law. Therefore, we see no basis for the claim *"The regularized Coulomb law is thought to perform well ..."* for our context. Since we have already partly addressed this issue in our response to the comment of referee 1, we will restate this reply here: *As the regularized Coulomb law negligibly increases basal drag beyond the order of the regularization threshold ($UV_{\mathrm{C,reg}} = 20$ m/yr), the regularized Coulomb law lacks*

*the increase in basal drag with basal velocity intrinsic to Weertman sliding and therefore will be more numerically unstable. This is compounded by the schoofing grounding-line flux iteration in the SSA solution. It should also be noted that the GSM SSA solution imposes an upper bound of 40 km/yr on SSA ice velocities for this configuration. We suspect that this is higher than most other models. The imposition of this upper bound is itself another non-linearity in the solution that can contribute to both instability (as adding non-linearities will generally decrease convergence of iterative solutions) and stability (by limiting ice velocities).*

**Changes:** Added *According to CFL constraints, we expect even the regularized Coulomb law (Eq. 6b) to be more unstable than the Weertman law, because it negligibly increases basal drag for basal velocities beyond the order of the regularization threshold $UV_{\text{C,reg}} = 20$ m yr$^{-1}$. This is compounded by the grounding-line flux iteration in the SSA solution.*

[Runs that are subjected to severe numeric problems can often illustrate that the physical problem is not well-posed. Can you discard this is what is happening here? How many of your reference ensemble runs crashed?] Ice stream surge cycling in itself is a highly non-linear physical process. Therefore, we are not surprised if some runs crash (more accurately, automatically terminate when the required time step size is reduced below the set minimum of 0.015625 yr as such short timesteps do raise concerns about numerical stability), especially since we are probing a large parameter space. One of the criteria for the runs in the reference ensemble was a successful completion, so by definition, no reference ensemble runs crashed. Furthermore, only runs that did not crash are included in the analysis of each sensitivity ensemble.

**Changes:** Added *Crashed runs are not considered in any of the above steps. Therefore, the reference ensemble only contains parameter vectors with a successful run completion at the fine horizontal grid resolution.*

[Looking at figure 5, the reader might notice that the upper limit of the basal velocities bar is 40,000 m/yr. It is not known whether the simulated velocities ever approach that value, because the manuscript does not contain the time series of velocities over a whole ice stream cycle, but I imagine that they are not far from the upper limit of the depicted scale. From figure 5 it is clear that they reach up to 20,000 m/yr for several hundred of kilometers upstream of the grounding line. Current observed surface velocities in Antarctica do not go beyond 1,500 m/yr downstream of the grounding line. Do you think your simulated velocities are realistic?] The upper limit of the basal velocities colorbar is set to 40 km/yr because it is the upper bound on the SSA ice velocities imposed in the GSM. While observed velocities of surging glaciers can reach several hundreds of meters per day for short periods [K.M. Cuffey and W.S.B. Paterson., 2010, , e.g., 100 m/d = 36.5 km/yr], we do not have a clear present-day analog for Hudson Strait ice stream. Can the referee confidently ascert such a velocity was never reached in the past? We can not. However, we agree it is important to document maximum velocities in the experiments. **Changes:** Added ... $\boldsymbol{u_b}$ is the basal sliding velocity *(imposed upper limit of 40 km yr$^{-1}$)*

[All in all, unless the authors show otherwise, their experimental setup is constructed in a way that the relevant-for-the-problem parameter space (and thus the phase space of the associated physical problem) is very narrow and situated in a very specific region.] We are unsure what the referee is referring to here. If the referee is arguing about the limited parameter space, given that all previous studies for this context outside of our own research group have used a much more limited parameter space, on what basis should they have been published? To reiterate, we sample 52 input parameters over wide parameter ranges based on extensive history-matching results. While others might not have such a large explicit parameter space, given system uncertainties, it is still implicitly there if one is actually trying to make meaningful inferences about Hudson Strait ice stream surge cycling. Additionally, we included the *bounding experiments* (or end member scenarios, Sec. 2.5.3) to bound the effects of the ocean forcing experiments and increase confidence in our model results.

Furthermore, our conclusions about the most likely HE mechanism generally hold for nearcontinuous ice streaming with occasional shutdowns and subsequent surge onset overshoot ($GHF_{ave} > 37\,mW\,m^{-2}$) and the classic binge-purge surge cycling ($GHF_{ave} \leq 37\,mW\,m^{-2}$). These conclusions are based on a high-variance sub-ensemble (with respect to ensemble parameters and ice sheet configuration). The choice of high variance sub-ensemble and the documentation of sub-ensemble variance of results quantifies the extent to which results are parameter vector dependent. As we also examine various sliding laws and as it has been previously shown that in a Hudson Strait ice stream context *sub-glacial hydrology matters but the process details mostly do not* [Hank et al., 2023, Drew and Tarasov, 2023], we do not see why the relevant-for-the-problem parameter space would be very narrow and situated in a very specific region, especially in comparison to previous studies.

[In the absence of a rigorous illustration of the influence of the particular assumptions of the experimental setup (basal sliding laws, basal water, effective pressure, dependence of the friction coefficient on the nature and temperature of the bed, numerical issues...) on the mechanisms favouring spontaneous ice stream cycling, the conclusions reached here are based on the analysis of a very particular experimental setup. (Which in my opinion is, by construction, prompt to oscillate in a questionable physical manner).] If the above is the standard required to credibly examine Hudson Strait ice surging, then no paper to date on the subject should have been published. Our aim is to bound the possible behaviour within the full system complexity of the North American ice sheet. We, therefore, intentionally use an ensemble approach to test the relative role and robustness of key surge cycling mechanisms/hypotheses given the uncertainties in climate forcing, basal drag, basal hydrology, and such. These are uncertainties that previous studies have largely ignored. Based on the reviewer's reasoning, it is fine to ignore uncertainties, but if you try to address them, then you have to isolate and analyze each contribution separately. We strongly disagree.

As mentioned previously, the effects of basal water, effective pressure, dependence of the friction coefficient on the nature and temperature of the bed, and model numerics have been extensively studied by Hank et al. [2023] and Drew and Tarasov [2022]. Although limited by the number of successful runs, the effect of different sliding laws is examined in this study.

[And this leads me to my next point: What happens if you run the Antarctic ice sheet under such a configuration?] and [To answer this question, you would need to make some assumptions concerning the bed type. One reasonable choice would be to assume the presence of soft sediments in every Antarctic marine sector. Do you expect that such a simulation would give reasonable results?] and [In my modelling experience, this is unlikely.] and [GRISLI and Yelmo can also show spontaneous ice stream oscillations under extreme conditions (e.g, limiting basal water and having very distinct spatial basal frictions). But when you apply such a physics to Antarctica, it becomes very difficult to approach observed velocity values, and furthermore some ice streams become very noisy and others dramatically oscillate in a manner that so far is not observed in Antarctica.] and [Therefore, I believe the current experimental setup extremely conditions the current conclusions of the paper.] Given the reviewer's concerns, we re-ran a few North American simulations extracting yearly ice stream velocities for a few diagnostic grid cells and found no significant high-frequency instability (annual ice velocity step), as described by the referee for Antarctica. Furthermore, the use of MNEEs partly delineates what is numerically significant (as any model response below the MNEE threshold is considered to be within numerical uncertainties).

**Changes:** Added *To further increase confidence in model numerics, we extracted yearly ice stream velocities (default time series output is* 100 yr*) for a few diagnostic grid cells (including Hudson Strait and Hudson Bay) from the reference ensemble and found no significant high-frequency instability ("noisy" ice streams).*

[The introduction section nicely ends with the presentation of very relevant questions regarding the different mechanisms exciting the Hudson Strait ice stream variability during HEs. The discussion section also nicely re-addresses those questions (see some minor comments below) in the context of the new results shown in the manuscript. The conclusions, however, do

not fairly summarize the findings and are biased towards the spontaneous ice-stream cycling mechanism.] and [After everything that is shown in the paper, and given all the limitations of the experimental setup pointed out in this review, concluding that "Based on our results, Hudson Strait ice stream surge cycling is the most likely Heinrich Event mechanism..." seems unjustified.] and [The questions Q2, Q3 and Q4 of your discussion are compatible with each other. When answering Q3 you state that your maximum simulated ice shelf (leaving aside the one with inhibited calving) is close to the minimum required by Hulbe 2004 to explain the IRD signal. So, can you discard that a combination of Q2, Q3 and Q4 produces both the icebergs from breaking the ice shelves and the increase in the flux at the grounding line necessary for explaining HEs?] This is a good point to consider, but given our experimental design (which keeps most processes active) and results, combining Q2/Q3/Q4 without Hudson Strait ice stream cycling can not explain (at least for the GSM) HEs. We have now added a discussion of a combined mechanism of Q2, Q3, and Q4 (sub-surface ocean warming leads to the breakup of fringing ice shelves and, consequently, a sudden reduction of the buttressing effect).

**Changes:** Added *Even when considering fringing ice shelves along the Canadian coast (*52.5 *to* 75.0°N*), the maximum ice shelf volume between* 100 *and* 10 kyr BP *across all reference runs is only* $6.2 \cdot 10^4$ km$^3$ *and is below the minimum estimate of Hulbe et al. [2004] for* 12 *(out of* 20*) runs.* to the discussion and *Since the ice shelf volume along the Canadian coast is generally smaller than the estimated minimum required for HEs, and since the fringing ice shelves in front of Hudson Strait provide only minor buttressing, our results indicate that even when considering a combined mechanism of sub-surface ocean warming, breakup of fringing ice shelves, and consequential sudden reduction of the buttressing effect, HEs can not be explained without Hudson Strait ice stream surge cycling.* to the conclusions.

However, as discussed in the conclusions (see response to the comment directly below), our model does not resolve the issue of HEs generally occurring during the coldest phases of Bond cycles. As such, these other mechanisms, if not something else, may, to varying degrees, contribute more of a timing control than is resolved in the experiments herein.

[Figure]

Figure 1: Labrador Sea ice shelf volume in the *Labrador Sea ice shelf area* outlined in Fig. 2. The thick line represents the mean of the 20 run ensemble. The shaded area marks the minimum and maximum of the ensemble.

[Under a questionable and very particular experimental setup, your simulations show that

you have to go to the very low bound of Geothermal heat forcing for the ice stream surge cycling to emerge. But, even assuming that a binge-purge like oscillation is a good candidate to explain HEs, there is a remaining puzzling question that you completely ignored. Why are HEs happening at the middle of the cold NH phases, or stadials? Some synchronization mechanisms have been explored in the literature to potentially answer this question (.e.g. the work of Calov and Ganopolski and Shanwell et al., 2024) but you do not inform whether these synchronization mechanisms are captured in your simulations. So, under your experimental setup, the question of HEs occurrence during the surface NH cold phases is still of concern for your conclusions (see also Barker et al., 2015; Nature). Therefore, the authors seem to be evaluating certain hypotheses with a level of criticism and rigour not present in the case of their announced more likely mechanism.] While a low GHF (but as we have argued above, not extremally low given available literature) is essential for the classic binge-purge surge mechanism, it is not a requirement for the second ice stream surge cycling mode discussed in this study (near-continuous ice streaming with occasional shutdowns and subsequent surge onset overshoot). As stated in the manuscript: *ocean forcings can affect the timing of surges and provide a means to synchronize HEs with the coldest phases of the Bond cycles.* However, we expand on this in the revised draft, including the recently published results of Schannwell et al. [2024, not published at the time of manuscript submission].

**Changes:** *Based on our results, Hudson Strait ice stream surge cycling is the most likely Heinrich Event mechanism, but ocean forcings can affect the timing of surges and provide a means to synchronize HEs with the coldest phases of the Bond cycles.* **to** *Overall, our experiments indicate that Hudson Strait ice stream surge cycling is the most likely Heinrich Event mechanism, but ocean forcings can indirectly affect the timing of surges through a change in ice sheet evolution. The key HE characteristic our experiments have not resolved is HE synchronization with the coldest phases of the Bond cycles (though not always the case, e.g., HE1). While ice shelf collapse and ocean forcing are insufficient to synchronize Hudson Strait ice stream surge cycling in the experiments presented here, a synchronization mechanism without the need for a trigger event during the stadial [Schannwell et al., 2024] could potentially provide the missing link and should be explored in future studies.*

[You discard the ice-shelf breakup related hypothesis (Q2) because your simulated ice shelves are not big enough to significantly buttress the Hudson strait ice stream and because the paleo reconstructions of the Labrador Sea conditions during MIS3 do not seem compatible wit the existence of a big ice shelf in the Area. This is a fair criticism, but I would like to point out here that in GRISLI and Yelmo (both codes are available; the first upon request to the former developer, Catherine Ritz or myself, and the second here: https://github.com/palma-ice/yelmo) the emergence of a big Labrador Sea ice shelf is a pretty natural characteristic provided oceanic temperatures are low enough. Even with a relatively warm ocean, Yelmo simulates the existence of very developed fringing ice shelves around the mouth of Hudson Strait (Moreno_parada et al., 2023; The Cryosphere).] The paleoceanographic constraints and oceanic temperatures being low enough are the crux. If temperatures are low enough, it is not surprising to have Baffin Bay covered by an ice shelf along with the confined part of the Labrador Sea, especially if one considers the comparative area of say the inferred LGM Ronne-Filchner ice shelf. However, is it reasonable to assume that conditions over the LGM Labrador Sea were similar to those of LGM Ronne-Filchner? The paleoceanographic constraints appear to rule this out. Furthermore, the mouth of Hudson Strait is at the edge of Labrador Sea confinement. As stated in our submission, the base simulations do have fringing ice shelves (including at the mouth of Hudson Strait); the critical issue is whether these are large enough to exert enough back stress to affect Hudson Strait streaming. Reducing the ocean temperature in the Labrador Sea ice shelf area by 2°C (end member scenario, maximum ocean temperature within 1°C of freezing point for 50% of the runs) slightly increases the ice shelf size. However, calving remains a restriction factor for the ice shelf size in the GSM. The end member scenario without calving in the Labrador Sea ice shelf area demonstrates that the GSM can generally

grow large ice shelves. As this is an extreme scenario and, more critically, large ice shelves are inconsistent with the paleo reconstructions, we conclude that the ice-shelf breakup hypothesis is unlikely.

**Changes:** Added *Nonetheless, as the maximum ocean temperature in the Labrador Sea ice shelf area when applying $T_{\mathrm{add}} = -2°C$ (EMS$_3$) is within $1°C$ of the freezing point for 10 (out of 20) runs, calving is the main restricting factor for the growth of large ice shelves.*

**and changed** Reducing the ocean temperature by $-2°C$ leads to minor ice shelf growth. **to** Reducing the ocean temperature by $-2°C$ leads to minor ice shelf growth*, as calving remains a restricting factor.*

[I believe that referencing our paper (Alvarez-Solas et al., 2013) in lines 95 and 548 in a technical and discussing context respectively, without having it cited in the introduction, when the different hypotheses are explained, is academically incorrect. Note, we also explored the conceptual idea of triggering HEs through changes in the oceanic temperature (Alvarez-Solas et al., 2010; NatGeo), and the effects of an ice-shelf breakup during H1 (Alvarez-Solas et al, 2011, ClimPast), coetaneous with Marcott et al., 2011] We added the references to the introduction.

**Changes:** Such an increase can significantly degrade the buttressing effect of a confined ice shelf and, thereby, potentially trigger ice stream activation or surging *[e.g., Álvarez-Solas et al., 2011]* **and** ... applying a sub-surface ocean temperature increase for every DO event *(similar to the approach used by, e.g., Alvarez-Solas et al. [2010, 2013])*

**References**

J. Álvarez-Solas, M. Montoya, C. Ritz, G. Ramstein, S. Charbit, C. Dumas, K. Nisancioglu, T. Dokken, and A. Ganopolski. Heinrich event 1: An example of dynamical ice-sheet reaction to oceanic changes. *Climate of the Past*, 7(4):1297–1306, 2011. ISSN 18149324. doi: 10.5194/cp-7-1297-2011.

Jorge Alvarez-Solas, Sylvie Charbit, Catherine Ritz, Didier Paillard, Gilles Ramstein, and Christophe Dumas. Links between ocean temperature and iceberg discharge during Heinrich events. *Nature Geoscience*, 3(2):122–126, 2010. ISSN 17520894. doi: 10.1038/ngeo752.

Jorge Alvarez-Solas, Alexander Robinson, Marisa Montoya, and Catherine Ritz. Iceberg discharges of the last glacial period driven by oceanic circulation changes. *Proceedings of the National Academy of Sciences of the United States of America*, 110(41): 16350–16354, 2013. ISSN 00278424. doi: 10.1073/pnas.1306622110.

D.D. Blackwell and M. Richards. Geothermal map of north america. AAPG Map, scale 1:6,500,000, Product Code 423, 2004. URL https://www.smu.edu/~/media/Site/Dedman/Academics/Programs/Geothermal-Lab/Graphics/Geothermal_MapNA_7x10in.gif.

Francisco José Cuesta-Valero, Almudena García-García, Hugo Beltrami, J. Fidel González-Rouco, and Elena García-Bustamante. Long-term global ground heat flux and continental heat storage from geothermal data. *Climate of the Past*, 17(1):451–468, 2021. ISSN 18149332. doi: 10.5194/cp-17-451-2021.

M. Drew and L. Tarasov. Surging of a hudson strait scale ice stream: Subglacial hydrology matters but the process details don't. *The Cryosphere Discussions*, 2022:1–41, 2022. doi: 10.5194/tc-2022-226. URL https://tc.copernicus.org/preprints/tc-2022-226/.

M. Drew and L. Tarasov. Surging of a hudson strait-scale ice stream: subglacial hydrology matters but the process details mostly do not. *The Cryosphere*, 17(12):5391–5415, 2023. doi: 10.5194/tc-17-5391-2023. URL https://tc.copernicus.org/articles/17/5391/2023/.

Gwenn E. Flowers and Garry K. C. Clarke. A multicomponent coupled model of glacier hydrology 1. Theory and synthetic examples. *Journal of Geophysical Research: Solid Earth*, 107(B11), 2002. ISSN 0148-0227. doi: 10.1029/2001jb001122.

Gwenn Elizabeth Flowers. *A multicomponent coupled model of glacier hydrology*. PhD thesis, University of British Columbia, 2000. URL https://open.library.ubc.ca/collections/ubctheses/831/items/1.0053158.

Sven Fuchs, Ben Norden, Florian Neumann, Norbert Kaul, Akiko Tanaka, Ilmo T. Kukkonen, Christophe Pascal, Rodolfo Christiansen, Gianluca Gola, Jan Šafanda, Orlando Miguel Espinoza-Ojeda, Ignacio Marzan, Ladislaus Rybach, Elif Balkan-Pazvantoğlu, Elsa Cristina Ramalho, Petr Dědeček, Raquel Negrete-Aranda, Niels Balling, Jeffrey Poort, Yibo Wang, Argo Jõeleht, Dušan Rajver, Xiang Gao, Shaowen Liu, Robert Harris, Maria Richards, Sandra McLaren, Paolo Chiozzi, Jeffrey Nunn, Mazlan Madon, Graeme Beardsmore, Rob Funnell, Helmut Duerrast, Samuel Jennings, Kirsten Elger, Cristina Pauselli, and Massimo Verdoya. Quality-assurance of heat-flow data: The new structure and evaluation scheme of the IHFC Global Heat Flow Database. *Tectonophysics*, 863(June), 2023. ISSN 00401951. doi: 10.1016/j.tecto.2023.229976.

K. Hank, L. Tarasov, and E. Mantelli. Modeling sensitivities of thermally and hydraulically driven ice stream surge cycling. *Geoscientific Model Development*, 16(19):5627–5652, 2023. doi: 10.5194/gmd-16-5627-2023. URL https://gmd.copernicus.org/articles/16/5627/2023/.

Christina L. Hulbe, Douglas R. MacAyeal, George H. Denton, Johan Kleman, and Thomas V. Lowell. Catastrophic ice shelf breakup as the source of Heinrich event icebergs. *Paleoceanography*, 19(1):n/a–n/a, 2004. ISSN 0883-8305. doi: 10.1029/2003pa000890.

C. Jaupart, J. C. Mareschal, H. Bouquerel, and C. Phaneuf. The building and stabilization of an Archean Craton in the Superior Province, Canada, from a heat flow perspective. *Journal of Geophysical Research: Solid Earth*, 119(12):9130–9155, 2014. ISSN 21699356. doi: 10.1002/2014JB011018.

Alan M. Jessop and Alan S. Judge. Five Measurements of Heat Flow in Southern Canada. *Canadian Journal of Earth Sciences*, 8 (6):711–716, 1971. ISSN 0008-4077. doi: 10.1139/e71-069.

K.M. Cuffey and W.S.B. Paterson. *The Physics of Glaciers*. Butterworth-Heinemann/Elsevier, Burlington, MA, 4th edition, 2010. ISBN 9780123694614.

F. Levy, C. Jaupart, J. C. Mareschal, G. Bienfait, and A. Limare. Low heat flux and large variations of lithospheric thickness in the Canadian Shield. *Journal of Geophysical Research: Solid Earth*, 115(6):1–23, 2010. ISSN 21699356. doi: 10.1029/2009JB006470.

F. Lucazeau. Analysis and Mapping of an Updated Terrestrial Heat Flow Data Set. *Geochemistry, Geophysics, Geosystems*, 20 (8):4001–4024, 2019. ISSN 15252027. doi: 10.1029/2019GC008389.

Nathan Maier, Florent Gimbert, Fabien Gillet-Chaulet, and Adrien Gilbert. Basal traction mainly dictated by hard-bed physics over grounded regions of Greenland. *Cryosphere*, 15(3):1435–1451, 2021. ISSN 19940424. doi: 10.5194/tc-15-1435-2021.

Henry N Pollack, Suzanne J Hurter, and Jeffrey R Johnson. Heat flow from the Earth's interior: Analysis of the global data set. *Reviews of Geophysics*, 31(3):267–280, 1993. doi: https://doi.org/10.1029/93RG01249. URL `https://agupubs.onlinelibrary.wiley.com/doi/abs/10.1029/93RG01249`.

Clemens Schannwell, Uwe Mikolajewicz, Marie Luise Kapsch, and Florian Ziemen. A mechanism for reconciling the synchronisation of Heinrich events and Dansgaard-Oeschger cycles. *Nature Communications*, 15(1):1–8, 2024. ISSN 20411723. doi: 10.1038/s41467-024-47141-7.

Nikolai M. Shapiro and Michael H. Ritzwoller. Inferring surface heat flux distributions guided by a global seismic model: Particular application to Antarctica. *Earth and Planetary Science Letters*, 223(1-2):213–224, 2004. ISSN 0012821X. doi: 10.1016/j.epsl.2004.04.011.

Victor C. Tsai, Andrew L. Stewart, and Andrew F. Thompson. Marine ice-sheet profiles and stability under Coulomb basal conditions. *Journal of Glaciology*, 61(226):205–215, 2015. ISSN 00221430. doi: 10.3189/2015JoG14J221.